# Understanding snow saltation parameterizations: lessons from theory, experiments and numerical simulations

Daniela Brito Melo[1,2], Armin Sigmund[1], and Michael Lehning[1,2]

[1]School of Architecture, Civil and Environmental Engineering, Ecole Polytechnique Fédérale de Lausanne, Lausanne, Switzerland
[2]WSL Institute of Snow and Avalanche Research SLF, Davos, Switzerland

**Correspondence:** Daniela Brito Melo (daniela.britomelo@epfl.ch)

**Abstract.** Drifting and blowing snow are important features in polar and high mountain regions. They control the surface mass balance in windy conditions and influence sublimation of snow and ice surfaces. Despite their importance, model representations in weather and climate assessments have high uncertainties because the associated physical processes are complex and highly variable in space and time. This contribution investigates the saltation system, which is the lower boundary condition for drifting and blowing snow models. Using a combination of (previous) measurements and new physics-based modeling with Large Eddy Simulation (LES), we show that the prevailing parameterizations that describe the saltation system in atmospheric models are based on contradictory assumptions: while some scaling laws are typical of a saltation system dominated by aerodynamic entrainment, others represent a saltation system controlled by splash. We show that both regimes can exist, depending on the friction velocity. Contrary to sand saltation, aerodynamic entrainment of surface particles is not negligible. It is important at low wind speeds, leading to a saltation height and near-surface particle velocity which increase with the friction velocity. In a splash dominated saltation regime at higher friction velocities, the saltation height and near-surface particle velocity become invariant with the friction velocity and closer to what is observed with sand. These findings are accompanied by a detailed description of the theoretical, experimental and numerical arguments behind snow saltation parameterizations. This work offers a comprehensive understanding of the snow saltation system and its scaling laws, useful for both modelers and experimentalists.

## 1 Introduction

The aeolian or wind-driven transport of snow is common in snow-covered regions, in particular in high mountain areas, the Arctic and the Antarctic. This phenomenon, generally named drifting or blowing snow, occurs when the wind velocity is sufficiently high to lift the snow particles from the surface. During snow transport events, the wind erodes the snowpack in the most exposed regions and leads to snow re-deposition in sheltered areas. Therefore, the formation of bedforms, such as dunes, ripples, sastrugi (Filhol and Sturm, 2015), as well as cornices at the leeward side of mountain crests (Hancock et al., 2020; Yu et al., 2022), is often observed. Snow sublimation is enhanced because the snow particles aloft are more exposed to the dry air flow (Dyunin, 1967; Liston and Sturm, 2004; Dai and Huang, 2014). In addition, some regions of the snow cover become harder and denser due to the deposition of snow particles that were previously aloft (Sommer et al., 2018). The

influence of the aeolian transport of snow on the mass and energy balances of the snow cover is therefore evident: it leads to snow redistribution, enhances the mass loss due to sublimation and modifies the near-surface flow, temperature and moisture fields (Bintanja, 1998; Déry et al., 1998; Mott et al., 2018).

The terms drifting and blowing snow are commonly used to distinguish between weak and strong snow transport events. During drifting snow events, it is generally assumed that the snow particles do not rise above 2 m height above the ground, while during blowing snow events the snow grains are expected to reach higher regions of the atmosphere (e.g., Li and Pomeroy, 1997; Lenaerts et al., 2012). Nevertheless, in order to simplify the nomenclature, the term drifting snow is preferred throughout the text to designate snow transport events. These events are very frequent in the Antarctic continent, specifically in the coastal areas, which are subjected to strong katabatic winds throughout the year (Lenaerts et al., 2012; Palm et al., 2017). Drifting snow sublimation is also maximum along the coast due to the higher temperatures and lower values of relative humidity in comparison with the Antarctic plateau (Gallée, 1998; Palm et al., 2017; Gerber et al., 2023). On average, the Antarctic surface mass balance is controlled by snowfall, but snow sublimation is the main mass ablation process (Lenaerts et al., 2012; Palm et al., 2017; Gerber et al., 2023). In some regions of East Antarctica, like the Neumayer station and Adélie Land, drifting snow sublimation is expected to remove 17 % and 19 % of the snowfall, respectively (van den Broeke et al., 2010; Lenaerts and van den Broeke, 2012). Similar figures are obtained in the low Arctic, such as in Trail Valley Creek, Canada, where drifting snow sublimation is expected to remove 19.5 % of the snowfall (Pomeroy et al., 1997). In some regions of the Canadian prairies, estimates based on a simplified model indicate that sublimation losses due to snow transport may amount to 44 % or 74 % of the annual snowfall over a 4 km fetch depending on the meteorological conditions (Pomeroy et al., 1993). In the Alpine region, drifting snow sublimation is highly variable in space and is, on average, of the same order of magnitude of surface sublimation (Groot Zwaaftink et al., 2011). In mountain areas worldwide, snow transport and redistribution influence snow height variability, which plays a significant role on the occurrence of avalanches (Lehning et al., 2000; Das et al., 2012) and hinders the assessment of snowfall from direct snow height observations (Liston and Sturm, 2002; Mott and Lehning, 2010).

The current estimates of snow transport and sublimation are based on several decades of experimental measurements that supported the development of theoretical models and empirical correlations for the mass flow rate of particles aloft and their respective sublimation (e.g., Budd et al., 1966; Kobayashi, 1972; Schmidt, 1982; Pomeroy and Gray, 1990). Nevertheless, the overall contribution of drifting snow transport and sublimation to the surface mass balance of snow-covered regions on Earth is still uncertain. For example, while the regional atmospheric model RACMO (van Wessem et al., 2018) estimates a total of $102 \pm 5$ Gt y$^{-1}$ of drifting snow sublimation over the Antarctic continent (except the Antarctic Peninsula), the amount of drifting snow sublimation derived from satellite observations (Palm et al., 2017) over the Antarctic continent (except the region south of the 82°S parallel) yielded $393 \pm 196$ Gt y$^{-1}$. The disagreement between model and measurements of drifting snow sublimation brings into question the accuracy of large-scale predictions and highlights the need to revise the drifting snow formulations implemented in atmospheric models.

The motion of drifting snow particles is generally conceptualized in three modes: creep, saltation and suspension (Bagnold, 1941). Creep corresponds to the rolling and sliding of snow particles, or to small hops with a maximum height of approximately

the particle diameter. Saltation defines the ballistic motion of snow particles close to the surface, with a maximum height of no more than 10 to 15 cm (Gordon et al., 2009). These particles are either lifted by the turbulent flow (aerodynamic entrainment) or ejected by an impacting snow particle (splash). After impacting the surface, the saltating particles might deposit or bounce-off into a new trajectory (rebound). This region close to the surface where most particles are transported in saltation is named the saltation layer. Suspension is the motion of small snow particles that travel without regular contact with the surface. These particles are advected by the wind for long distances and up to high regions of the boundary layer.

The general approach in atmospheric and snow models (Lehning et al., 2008; Vionnet et al., 2014; van Wessem et al., 2018; Amory et al., 2021; Sharma et al., 2023) is to parameterize the snow particle concentration and mass flux in the saltation layer and to compute the spatial and temporal distribution of snow particles in suspension by taking into account horizontal particle advection, particle settling and the turbulent vertical mixing of particles aloft (Dyunin and Kotlyakov, 1980; Pomeroy and Male, 1992; Déry et al., 1998). The bottom boundary of the snow suspension scheme is defined at the top of the saltation layer, where particle concentration is specified as a function of the near-surface wind velocity and some snow surface characteristics. The contribution of creep to the mass flux of snow particles is either neglected or assumed to be included in the snow saltation parameterization. Snow sublimation is usually estimated with the model of Thorpe and Mason (1966) (Groot Zwaaftink et al., 2011; Lenaerts et al., 2012; Vionnet et al., 2014; Sharma et al., 2023). The process of snow sublimation extracts heat and adds moisture from/to the near-surface flow, which is taken into account in the energy and moisture budgets of the near-surface atmosphere.

Shortcomings in the modeled drifting snow mass flux and sublimation can be related to uncertainties in the input parameters, such as the near-surface wind velocity, which greatly depends on the description of the surface topography (Gauer, 1998; Raderschall et al., 2008; Bernhardt et al., 2009; Lenaerts et al., 2012), or to inaccuracies in the drifting snow scheme itself. In fact, and concerning the latter, it seems that the resultant quantities of interest are significantly sensitive to the snow saltation parameterizations applied. For instance, using the atmospheric model MAR, Amory et al. (2021) obtained an improved agreement between the simulated and measured drifting snow mass fluxes at D47 (Adélie Land, Antarctica) mainly by adjusting the minimum shear stress exerted by the flow that leads to the onset of snow drift. Another example can be given, this time with the atmospheric model RACMO: van Wessem et al. (2018) reduced the computed drifting snow sublimation over the Antarctic continent (except the Antarctic Peninsula) from $181 \pm 9 \, \text{Gt y}^{-1}$ to $102 \pm 5 \, \text{Gt y}^{-1}$ by halving the snow particle concentration at the top of the saltation layer. While the first parameter has the ability to vary the frequency of drifting snow events, the particle concentration at the top of the saltation layer mentioned in the second example affects the total mass of particles in suspension and, therefore, the rate of drifting snow sublimation.

As discussed above, a rigorous description of snow saltation in atmospheric models is of the utmost importance. Unfortunately, too little attention has been given to its development. The early model of Pomeroy and Gray (1990) has been used in the drifting snow schemes of RACMO (Lenaerts et al., 2012; van Wessem et al., 2018) and MAR (Amory et al., 2015, 2021). However, this saltation model is based on poorly constrained parameters and the highly simplified assumption that the vertically-averaged mass flux in saltation is well approximated by a point measurement inside the saltation layer. Vionnet et al. (2014) considered a different set of parameterizations in the atmospheric model Meso-NH. In particular, the vertical profile of particle

mass flux in saltation is approximated by an exponential equation, as a function of the transport rate and a decay parameter, which we denote decay height. This formulation was also adopted by Sharma et al. (2023) when coupling the Weather Research and Forecasting model (WRF) (Skamarock et al., 2019) with a drifting snow scheme and the snow model SNOWPACK (Bartelt and Lehning, 2002). The exponential decay of the mass flux in saltation with increasing height above the surface is a well known feature of the saltation system (e.g., Sugiura et al., 1998; Nishimura and Nemoto, 2005; Kok and Renno, 2009; Martin and Kok, 2017). However, the calculation of the transport rate and the decay height is still a challenge. Several expressions have been proposed, but there is still no full consensus on how these quantities vary with the wind field and the snow cover characteristics.

In this work, we revisit the different snow saltation parameterizations available in the literature and their use in atmospheric models. In particular, we focus on the theoretical, experimental and numerical arguments that support them (Sect. 2). We highlight where consensus and disagreement exist in order to help the reader develop a critical view on the currently used snow saltation parameterizations. By pinpointing where most of the uncertainty lies, this exposition also seeks to motivate further studies in the field. In addition, we use a numerical model to investigate some aspects of the saltation system that are not yet fully understood (Sect. 3). The numerical model computes the particle trajectories in a turbulent flow by means of a Large Eddy Simulation flow solver coupled with a Lagrangian stochastic model (LES-LSM) (Sharma et al., 2018; Comola et al., 2019b; Sigmund et al., 2022; Melo et al., 2022). The transport rate and the vertical profiles of particle mass flux and velocity are investigated, as well as the particle hop height and length during saltation. Special focus is given to the decay height and to its relation to the saltation dynamics. In Sect. 4, we summarize the main results and give suggestions for future work.

## 2 Snow saltation parameterizations

In this section, the different snow saltation parameterizations available in the literature and their use in atmospheric models are presented and discussed. We focus on the most common approximations within the steady-state framework and highlight the evidence that support them. After an introduction about the steady-state assumption (Sect. 2.1), the parameterizations used to compute the transport rate (Sect. 2.2) and the vertical profiles of particle concentration, streamwise velocity and mass flux (Sect. 2.3) are analyzed. These quantities are needed to compute the particle concentration at the top of the saltation layer - the lower boundary condition for the advection and diffusion of suspended particles (Lehning et al., 2008; Vionnet et al., 2014; van Wessem et al., 2018; Amory et al., 2021; Sharma et al., 2023). The effect of the wind velocity on the different quantities of interest is particularly discussed. The saltation dynamics is also highly dependent on the granular bed characteristics, which is of particular importance in the study of snow saltation (Comola and Lehning, 2017). However, considerably less literature is available on this topic.

### 2.1 The steady-state assumption

Let us denote by $x$ [m], $y$ [m] and $z$ [m] the streamwise, crosswise and vertical directions, respectively, in a Cartesian coordinate system. In this reference frame, we consider a turbulent air flow driven by a constant streamwise pressure gradient over a flat,

homogeneous and erodible surface. It is assumed that the wind field is statistically steady and that the time-averaged wind velocity does not vary along the streamwise and crosswise directions. Under these conditions, the saltation system is considered statistically steady and invariant along the horizontal directions (steady-state saltation).

Close to the surface and in the absence of saltating particles, the time-averaged fluid shear stress can be assumed constant in height (Prandtl, 1935). This region is generally named the inner layer. In the atmospheric boundary layer, this constant shear stress assumption is approximately valid in the first 10 % of the boundary layer depth. In geophysical flows, this region is called the surface layer and has a thickness of 10-100 m (Stull, 1988). When the surface is aerodynamically smooth, this near-surface region is conceptualized in three sublayers: a viscous sublayer in the vicinity of the surface, a logarithmic sublayer in the

upper region, and a buffer sublayer in between. When the surface is fully rough (which is frequently the case in atmospheric boundary layer flows), the viscous sublayer is disrupted by the roughness features (Monin, 1970). In this case, the constant shear stress region is better described by two sublayers: the roughness sublayer and the logarithmic sublayer (Jiménez, 2004). In the latter, the turbulent shear stresses (often denoted Reynolds stresses) are dominant in comparison to those of viscous origin. In addition, under neutral stability conditions, the average streamwise wind velocity is expected to follow a logarithmic profile,

which is a function of the friction velocity, $u_*$ [m s$^{-1}$], and of the roughness length, $z_o$ [m], which characterizes the roughness of the surface. In this way, the time-averaged fluid shear stress in the logarithmic sublayer, $\tau = \rho_a u_*^2$ [Pa], where $\rho_a$ [kg m$^{-3}$] is the air density, can be computed from the covariance of the velocity fluctuations or estimated by fitting the vertical profile of average streamwise wind velocity to a logarithmic function. In atmospheric models, the streamwise wind velocity is assumed to follow a logarithmic function between the surface and the first grid point (Gallée et al., 2001; White, 2003; Skamarock et al.,

2019). In agreement with Monin-Obukhov similarity theory (Monin and Obukhov, 1954), stability corrections are added to the standard logarithmic profile, which changes slightly the relationship between $u_*$, $z_o$ and the streamwise wind velocity at the first grid point above the surface. Under the constant shear stress assumption, the time-averaged fluid shear stress in the logarithmic sublayer is equal to the time-averaged surface shear stress, $\tau_s = \rho_a u_{*,s}^2$ [Pa] (henceforth referred to as surface shear stress), where $u_{*,s}$ [m s$^{-1}$] is the surface friction velocity.

When surface particles are entrained by the fluid flow and advected by it, there is an exchange of momentum between the fluid and the particles aloft. This leads to a decrease in the fluid shear stress in the saltation layer (Owen, 1964; Raupach, 1991). Above the saltation layer and within the logarithmic sublayer, the time-averaged fluid shear stress is still well approximated by $\tau$ and the average streamwise wind velocity profile is still expected to follow a logarithmic trend. However, this logarithmic profile is no longer a function of $z_o$, but of an equivalent roughness length, greater than $z_o$, which takes into account the momentum deficit in the saltation layer (Bagnold, 1941; Owen, 1964). At the surface, the fluid shear stress, $\tau_s$, reaches a value

smaller than $\tau$.

During steady-state saltation, the net erosion and deposition of particles from/on the granular bed is equal to zero. In this way, for a given friction velocity, $u_*$ (characteristic of the logarithmic region outside the saltation layer), the rate at which particles are transported by the wind stays at an equilibrium value. The saltation system is therefore characterized by a self-regulatory behaviour: a local increase in the particle flow rate above its equilibrium value locally extracts more momentum from the wind,

reducing its velocity; this decreases the ability of the air flow to accelerate the particles in saltation and, consequently, decreases the particle flow rate back to its equilibrium value (e.g., Bagnold, 1941; Owen, 1964).

The premises behind the steady-state saltation assumption contrast with the reality of snow-covered regions, which are rarely flat or homogeneous. In addition, the wind field in a mountainous terrain can seldom be characterized as statistically steady. Nevertheless, steady-state saltation is frequently assumed, based on the notion that the saltation system can adapt very rapidly to changes in the wind velocity (e.g., Anderson et al., 1991). Wind tunnel and field measurements have revealed a strong coupling between the saltation system and the wind velocity fluctuations (Paterna et al., 2016; Aksamit and Pomeroy, 2018). However, atmospheric models require simple and computationally light models that cannot capture the full complexity of the wind-particle interaction. In addition, the time scales of atmospheric models do not encompass the full wind field variability. In this context, when deriving snow saltation parameterizations for large-scale models, the assumption of steady-state saltation is regarded as a reasonable approximation.

## 2.2 The transport rate

The transport rate of saltating particles, $Q$ [kg m$^{-1}$ s$^{-1}$], corresponds to the mass flow rate of particles moving in saltation per unit width. From its definition, $Q$ can be computed by multiplying the mass of particles in saltation per unit surface area by the average particle streamwise velocity.

The transport rate can also be computed as a function of the wind field characteristics by performing a balance of linear momentum in the streamwise direction to the saltating particles and to the fluid flow. Let us assume that the only force applied to a given saltating particle in the streamwise direction is the aerodynamic force (neglecting, for instance, electrostatic forces and mid-air collisions). In this case, the average aerodynamic force in the streamwise direction applied to a particle along a hop equals the average rate of change of the particle streamwise momentum, $m(v_x^i - v_x^e)/t_p$, where $m$ [kg] is the particle mass, $v_x^i$ [m s$^{-1}$] and $v_x^e$ [m s$^{-1}$] are the streamwise components of the particle velocity at impact and ejection from the surface, respectively, and $t_p$ [s] is the duration of the particle hop (Fig. 1a). The balance of linear momentum in the streamwise direction applied to the fluid flow is simplified by assuming steady-state saltation conditions and by neglecting the pressure gradient. According to Newton's third law, the aerodynamic force applied to a particle is equal in magnitude and opposite to the force applied to the fluid. Therefore, the sum of aerodynamic forces in the streamwise direction applied to all particles in saltation per unit surface area equals the difference in shear stresses, $\tau - \tau_s$ (Fig. 1b). From these considerations and assuming that the variety of particle hops can be approximated by an average trajectory (characteristic path) (e.g., Bagnold, 1941; Ungar and Haff, 1987; Durán et al., 2011; Kok et al., 2012), the following expression for the transport rate is obtained:

$$Q = (\tau - \tau_s) \frac{\langle v_p \rangle \langle t_p \rangle}{\langle \Delta v_{x,s} \rangle} = (\tau - \tau_s) \frac{\langle l_p \rangle}{\langle \Delta v_{x,s} \rangle} \tag{1}$$

where $v_p$ [m s$^{-1}$] is the hop-averaged particle streamwise velocity, $\Delta v_{x,s} = v_x^i - v_x^e$ [m s$^{-1}$] is the difference between the impact and ejection streamwise velocities, and $l_p = v_p t_p$ [m] is the particle hop length in the streamwise direction (henceforth, hop length). The angle brackets, $\langle \rangle$, represent averages over the ensemble of particles in saltation. Therefore, $\langle v_p \rangle$ denotes

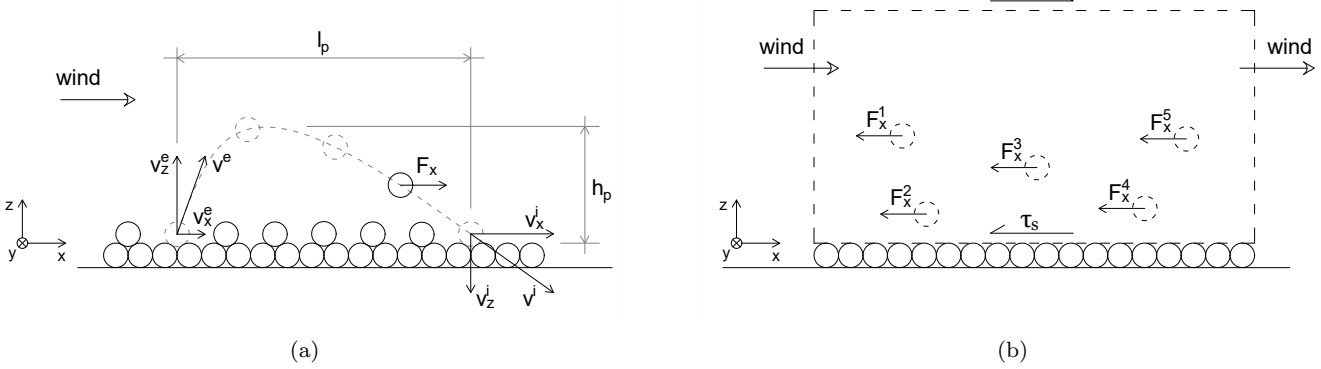

**Figure 1.** (a) Schematic of a particle hop: representation of the particle hop height and length, the instantaneous aerodynamic force in the streamwise direction applied to a particle, $F_x$ [N], and the particle ejection and impact velocities, $v^e$ [m s$^{-1}$] and $v^i$ [m s$^{-1}$] (please note that the ejection and impact velocities often have a crosswise component which is not represented in the figure). (b) Forces and stresses applied to the fluid flow in the streamwise direction in a given control volume spanning from the surface to a height above the saltation layer (only five particles are illustrated as an example).

the average particle streamwise velocity. When deducing Eq. 1, the snow surface was assumed homogeneous and erodible. However, the presence of non-erodible roughness elements can be taken into account by adding a parameterization for shear
stress partition (Pomeroy and Gray, 1990; Raupach et al., 1993).

The use of average quantities to describe the various particle trajectories introduces an error in the calculation of the transport rate. This assumption is not sufficiently discussed in the literature (e.g., Shao and Mikami, 2005). Nevertheless, Eq. 1 is the basis for several simple parameterizations (e.g., Bagnold, 1941; Ungar and Haff, 1987; Pomeroy and Gray, 1990; Sørensen, 1991; Kok et al., 2012) and is expected to yield a reasonable approximation for the scaling of $Q$ with $u_*$. The variety of
parameterizations proposed for $Q$ arise from different expressions for $\tau_s$, $\langle \Delta v_{x,s} \rangle$ and $\langle l_p \rangle$. These quantities are discussed in the next subsections.

### 2.2.1 The surface shear stress

The onset of saltation occurs when the surface shear stress grows above a given threshold, which allows the aerodynamic entrainment of surface particles to take place (Bagnold, 1941). This threshold shear stress is named the fluid threshold, $\tau_{ft}$ [Pa].
By definition, $\tau_{ft} = \rho_a u_{*,ft}^2$, where $u_{*,ft}$ [m s$^{-1}$] is the fluid threshold friction velocity. According to Owen (1964), during steady-state saltation, the particles are continuously set in motion by aerodynamic entrainment and the particle bombardment at the surface is only responsible for dislodging or loosening the grains. Naturally, he claims that saltation can only be sustained if $\tau_s$ is equal or greater than a minimum value that allows the entrainment of loose grains, previously dislodged by impacting particles. Owen (1964) goes one step further and hypothesizes that the surface shear stress stays at this minimum value at
which saltation is sustained (Owen's hypothesis): if $\tau_s$ is lower than this threshold no aerodynamic entrainment of previously

dislodged particles occurs; and if it is larger than that threshold more particles are entrained which ultimately restores $\tau_s$ to its original value. According to Owen, the minimum surface shear stress required to sustain saltation is equal to the impact threshold, $\tau_{it} = \rho_a u_{*,it}^2$ [Pa]: the lower fluid shear stress outside the saltation layer for which saltating particles maintain a steady-state motion (Bagnold, 1941). $u_{*,it}$ [m s$^{-1}$] is the impact threshold friction velocity. Due to the contribution of the

impacting particles to dislodging the grains, Owen (1964) expects the impact threshold to be smaller than the fluid threshold. Indeed, Bagnold (1941) showed with wind tunnel experiments developed with sand that the shear stress needed to keep saltating particles in motion is smaller than that required to initiate saltation from rest. In his experiments, the shear stress was computed from fitting the wind velocity outside the saltation layer to a logarithmic function. Therefore, the reported values of shear stress are characteristic of the logarithmic sublayer. Nevertheless, at the fluid and impact thresholds, the surface shear stress and the

shear stress outside the saltation layer should assume similar figures due to the low number of particles aloft and the consequent small momentum exchange between the fluid and the particles.

The experimental assessment of Owen's hypothesis is challenging because it requires high frequency measurements of the wind velocity close to the surface within a saltation layer (e.g., Li and McKenna Neuman, 2012). The measurements of Walter et al. (2014) of the surface shear stress during saltation revealed a slight variation of $\tau_s$ as $u_*$ increases. In addition, some

numerical models that include a statistical description of rebound and splash obtained a significant variation of the surface shear stress with respect to $u_*$ (Werner, 1990; Nemoto and Nishimura, 2004; Kok and Renno, 2009), while others predicted a negligible change (Niiya and Nishimura, 2017; Melo et al., 2022). Doorschot and Lehning (2002), who modeled steady-state saltation with a simplified numerical model, have also obtained a slight variation of $\tau_s$ with the rise of $u_*$. From the different trends obtained by measurements and models, it is clear that the self-regulatory behaviour of steady-state saltation is

not necessarily characterized by a constant surface shear stress, invariant with respect to $u_*$, but that different equilibrium states can emerge depending on the wind and particle characteristics. In this way, Owen's hypothesis is expected to oversimplify the saltation dynamics (Kok et al., 2012). Nonetheless, the assumption that $\tau_s$ does not vary with $u_*$, but depends solely on the bed characteristics, is regarded as an acceptable first order approximation (Walter et al., 2014; Niiya and Nishimura, 2022). Therefore, it might be appropriate for simple models and parameterizations, regardless of aerodynamic entrainment being

considered the main entrainment process (e.g., Pomeroy and Gray, 1990; Sørensen, 1991, 2004).

The numerical model of Kok and Renno (2009) has revealed an increase of $\tau_s$ with the increase of the particle diameter in the granular bed (see Fig. 13 in Kok et al., 2012). The same is found by Melo et al. (2022) when considering uniformly-sized particles. In the latter, it is seen that other characteristics of the granular bed, as the particle size distribution and the existence of cohesive bonds between the particles, also influence the value of $\tau_s$ during steady-state saltation.

In the drifting snow schemes of the atmospheric models RACMO, MAR, Meso-NH and CRYOWRF (Lenaerts et al., 2012; Amory et al., 2021; Vionnet et al., 2014; Sharma et al., 2023), $\tau_s$ is assumed invariant with respect to $u_*$ and approximated by the fluid threshold, $\tau_{ft}$. In these models, the fluid threshold varies with the snow surface characteristics, which change over time and space according to the meteorological conditions (Guyomarc'h and Mérindol, 1998; Gallée et al., 2001; Lehning et al., 2000). Therefore, the equality between $\tau_{ft}$ and $\tau_s$ guarantees that $\tau_s$ varies with the snow surface characteristics as

proposed by Owen (1964). However, the assumption that $\tau_s = \tau_{ft}$ neglects the effect of particle impacts in dislodging the

**Table 1.** The different shear stresses, their definitions and particular cases within the steady-state assumption.

| Concept | Symbol | Particular cases |
|---|---|---|
| Shear stress in the logarithmic sublayer | $\tau = \rho u_*^2$ | |
| Surface shear stress | $\tau_s = \rho u_{*,s}^2$ | No particles aloft: $\tau_s = \tau$ |
| | | Owen's hypothesis: $\tau_s = \tau_{it}$ |
| | | Common in atmospheric models: $\tau_s = \tau_{ft}$ |
| Fluid threshold: lowest shear stress at which saltation starts | $\tau_{ft} = \rho u_{*,ft}^2$ | At the fluid threshold: $\tau = \tau_s = \tau_{ft}$ |
| Impact threshold: lowest shear stress at which saltation is maintained [1] | $\tau_{it} = \rho u_{*,it}^2$ | At the impact threshold: $\tau = \tau_s = \tau_{it}$ |

[1]Some authors define the impact threshold as the minimum surface shear stress for which saltation is sustained by splash (Comola et al., 2022). This definition of $\tau_{it}$ will be considered in Sect. 3.3.

grains, which justifies Owen's claim that $\tau_s$ must be smaller than $\tau_{ft}$. The assumption that $\tau_s = \tau_{ft}$ also implies that the fluid shear stresses for saltation initiation and cessation are equal, which is a useful assumption (note that Eq. 1 reduces to zero when $\tau$ equals $\tau_s$). However, it is not in agreement with wind tunnel measurements nor saltation models developed for sand (e.g., Bagnold, 1941; Kok et al., 2012). For the case of snow saltation, the relationship between $\tau_s$ and $\tau_{ft}$ is still not clear. While some numerical models predict $\tau_s > \tau_{ft}$ (Nemoto and Nishimura, 2004; Melo et al., 2022), other predicts $\tau_s < \tau_{ft}$ (Niiya and Nishimura, 2017). Both trends are found with the model of Doorschot and Lehning (2002) depending on the characteristics of the snow cover. In this way, further investigation is needed to better assess the value of $\tau_s$ during snow saltation and the impact of the currently used assumption on the accuracy of the transport rate. A summary of the different shear stresses defined in this work and the main assumptions regarding the surface shear stress are presented in Table 1.

### 2.2.2 The ejection and impact velocities

When assessing the transport rate, Bagnold (1941) assumed that the ratio $\langle l_p \rangle / \langle \Delta v_{x,s} \rangle$ scales with $\langle v_z^e \rangle / g$, where $v_z^e$ $[\mathrm{m\,s^{-1}}]$ is the vertical component of the ejection velocity and $g$ $[\mathrm{m\,s^{-2}}]$ is the acceleration of gravity. One arrives to this result by solving the ballistic motion of a particle in saltation subjected to some simplifying assumptions (see for instance, Sørensen, 1991). In addition, Bagnold (1941) assumed that $\langle v_z^e \rangle$ is proportional to $u_*$, based on the idea that the particle velocity at impact must scale with $u_*$ (please note that $u_*$ is the friction velocity outside the saltation layer and not the surface friction velocity, which is defined by $u_{*,s}$).

Numerical models, like the one proposed by Ungar and Haff (1987), generally suggest that $\langle \Delta v_{x,s} \rangle$ is invariant with respect to $u_*$. Ungar and Haff (1987) arrived to this result by assuming that in steady-state saltation each particle impacting the surface is replaced by a new one, equal in size, splashed from the surface upon impact. Indeed, during steady-state saltation, zero net erosion and deposition of particles implies an equality between the mass flow rate of particles that fail to rebound and the mass flow rate of particles that successfully enter saltation via splash or aerodynamic entrainment (Kok et al., 2012). Contrary to Owen (1964), most numerical models assume that the surface processes of rebound and splash play an important role in steady-state saltation (e.g., Kok and Renno, 2009; Durán et al., 2011). These models compute the probability of rebound and

the number and initial velocity of splashed grains as a function of the bed characteristics and the impacting particle mass and velocity. As a result, they predict that $\langle \Delta v_{x,s} \rangle$ does not vary with $u_*$. This assumption is present in the analytic model of Sørensen (2004), which is used in Meso-NH (Vionnet et al., 2014) and CRYOWRF (Sharma et al., 2023) to compute the transport rate. In addition, numerical models of snow saltation that represent the surface processes of aerodynamic entrainment, rebound and splash have also revealed ejection and impact velocities that are mainly invariant with respect to $u_*$ (Melo et al., 2022; Niiya and Nishimura, 2022).

Field and wind tunnel measurements conducted with sand show a negligible influence of $u_*$ on the near-surface particle velocity. For instance, Namikas (2003) found a better agreement between model results and field measurements of particle mass flux when assuming an averaged ejection velocity invariant with respect to $u_*$ for friction velocities ranging from 0.27 to 0.63 $\mathrm{m\,s^{-1}}$. Moreover, wind tunnel experiments developed by Creyssels et al. (2009) and Ho et al. (2011) over an erodible sand surface show a negligible variation of the particle streamwise velocity close to the surface with respect to $u_*$, considering friction velocities ranging from 0.24 to 0.67 $\mathrm{m\,s^{-1}}$ and from 0.42 to 1 $\mathrm{m\,s^{-1}}$, respectively. During the experiments, the particles were illuminated by a laser sheet and recorded with a high-speed camera. The particle velocities were calculated with the Particle Tracking Velocimetry (PTV) technique and the particle streamwise velocity close to the surface was extrapolated from a linear regression of the average streamwise velocity profiles in the first 5 and 3 cm above the surface, respectively. The particle streamwise velocity close to the surface is named slip velocity and represents the average between the ejection and impact streamwise velocities.

The streamwise velocity of saltating snow was investigated by Aksamit and Pomeroy (2016) during field experiments in Alberta, Canada, using a similar experimental setup as the one used by Creyssels et al. (2009) and Ho et al. (2011). The streamwise velocity of ascending particles in the first 2.5 cm above the surface was reported for friction velocities ranging from 0.21 to 0.54 $\mathrm{m\,s^{-1}}$ for three different snow conditions (old snow, fresh snow and fine decomposing grains). The average particle ejection velocity in the streamwise direction, $\langle v_x^e \rangle$, was extrapolated from a linear regression of the average streamwise velocity of ascending particles along the first 1 cm above the surface. Even though steady-state conditions were not attained during the experiments, different trends were obtained for different snow surface characteristics. For the old and fresh snow covers, the streamwise ejection velocity was not found to vary monotonically with the friction velocity, while for the one composed of fine decomposing grains, a slight increase with $u_*$ was observed. Among the three snow surfaces, the latter was characterized by the highest hand hardness index. The increase of the streamwise ejection velocity with $u_*$, obtained over the surface of fine decomposing grains, is in agreement with the increase of the particle slip velocity as $u_*$ increases, obtained by Ho et al. (2011) after performing wind tunnel measurements of saltating sand over a rigid (non-erodible) surface, considering friction velocities ranging from 0.3 to 0.45 $\mathrm{m\,s^{-1}}$. Aksamit and Pomeroy (2016) obtained higher values of $\langle v_x^e \rangle$ over the fresh snow cover than over the fine decomposing grains and old snow covers. Differently, Ho et al. (2011) obtained higher values for the particle slip velocity over the rigid surface in comparison to the erodible surface.

In summary, most of the available literature suggests that the average ejection and impact velocities of particles in saltation are invariant with respect to $u_*$. However, the few experiments developed with snow reveal different trends depending on the snow surface characteristics. In order to understand the particular case of snow saltation, more experimental measurements

developed with snow are required, which must include a detailed characterization of the snow cover, the wind field and the kinematics of the particles in saltation.

### 2.2.3 The hop length

The particle hop length, $l_p$, is equal to the duration of the particle hop, $t_p$, times the hop-averaged particle streamwise velocity, $v_p$. If we approximate the variety of particle trajectories by an average trajectory, the average hop length, $\langle l_p \rangle$, can be estimated from the product $\langle t_p \rangle \langle v_p \rangle$. $\langle t_p \rangle$ is expected to deviate from the theoretical value of $2\langle v_z^e \rangle / g$ (obtained when neglecting the aerodynamic forces in the particle motion), but it is nonetheless expected to scale with $\langle v_z^e \rangle$ (Sørensen, 1991). In addition, as discussed in the previous section, the particle ejection velocity during steady-state saltation is generally regarded as invariant with respect to $u_*$. Under these conditions, the scaling of $\langle l_p \rangle$ with $u_*$ is mainly dependent on the scaling of $\langle v_p \rangle$ with $u_*$. Assuming that the particle ejection velocity does not vary with $u_*$, the average particle streamwise velocity, $\langle v_p \rangle$, depends solely on the aerodynamic forces applied on the particles during their trajectories and, therefore, on the wind velocity inside the saltation layer.

Ungar and Haff (1987) concluded that $\langle l_p \rangle$ is invariant with respect to $u_*$ and proportional to the particle diameter, $d$ [m]. This result is obtained by assuming that the ejection and impact velocities scale with $(dg)^{\frac{1}{2}}$ and that the wind velocity in the saltation layer is invariant with respect to $u_*$. Differently, the analytic model of Sørensen (2004), used in Meso-NH (Vionnet et al., 2014) and CRYOWRF (Sharma et al., 2023), arrived to an average particle hop length proportional to $u_*$. In the model, the particle ejection velocity is also assumed invariant with respect to $u_*$ and the logarithmic (particle free) wind profile is modified to take into account the reduction in fluid shear stress induced by the particles in saltation. Therefore, the wind velocity decreases in comparison to the logarithmic profile, but it is nonetheless proportional to $u_*$ inside the saltation layer.

Wind tunnel measurements show that the wind velocity during saltation is expected to be invariant with respect to $u_*$ in the first centimeter above the surface, where most saltating particles are (Bagnold, 1941; Li and McKenna Neuman, 2012). However, at higher elevations, the wind velocity is expected to scale with $u_*$ (e.g., Nishimura et al., 2014; Aksamit and Pomeroy, 2016). Therefore, if a significant number of particles travel above 1 cm height during a significant extent of their trajectories, $\langle v_p \rangle$ might increase as $u_*$ increases. This is in agreement with the field experiments of Namikas (2003) developed with sand, which revealed a slight increase of the average hop length with $u_*$.

The numerical model of Durán et al. (2011) suggests that the scaling of the hop length with the friction velocity differs according to the saltation regime. At a low-velocity regime (for, approximately, $u_*/u_{*,it} < 4$), most saltating particles are within this first centimeter above the surface where the wind velocity is invariant with respect to $u_*$. Therefore, the average particle velocity and hop length are expected to be invariant with respect to $u_*$. However, at a high-velocity regime ($u_*/u_{*,it} > 4$), a considerable number of saltating particles reach higher elevations above the surface, where the wind velocity increases with $u_*$. Under these conditions, both the average particle streamwise velocity and hop length are expected to increase linearly with $u_*$. Taking into account that $u_*/u_{*,it} > 4$ rarely occurs in nature for the case of saltating sand, Kok et al. (2012) argued that neglecting the effect of $u_*$ on the hop length can be regarded as a reasonable approximation. If both $\langle \Delta v_{x,s} \rangle$ and $\langle l_p \rangle$ are

assumed invariant with respect to $u_*$, the transport rate scales with $u_*^2$ (note that $\tau = \rho u_*^2$) (Ungar and Haff, 1987; Durán et al., 2011). Conversely, if $\langle l_p \rangle$ increases linearly with $u_*$, one arrives to $Q \propto u_*^3$ (Sørensen, 1991, 2004).

Further studies are needed in order to fully understand the scaling of $\langle l_p \rangle$ with the friction velocity during snow saltation. In particular, it would be interesting to unveil if the two regimes proposed by Durán et al. (2011) are also representative of saltating snow. The full vertical profile of the particle streamwise velocity is discussed in Sect. 2.3.2.

### 2.2.4 The average particle streamwise acceleration

The hop-averaged particle streamwise acceleration, $a_p$ [m s$^{-2}$], is given by the ratio $\Delta v_{x,s}/t_p$. When assuming a characteristic particle hop for all particles in saltation, the average particle streamwise acceleration, $\langle a_p \rangle$, is approximated by $\langle \Delta v_{x,s} \rangle / \langle t_p \rangle$. In this way, the transport rate in Eq. (1) can be expressed by

$$Q = (\tau - \tau_s) \frac{\langle v_p \rangle}{\langle a_p \rangle}. \tag{2}$$

According to the balance of linear momentum applied to the particles in saltation described in Sect. 2.2, the ratio between the total aerodynamic force along the streamwise direction applied to all particles in saltation and their weight is given by $\langle a_p \rangle / g$. Bagnold (1973) argued that this ratio between the applied horizontal force and the weight of the mass in motion represents a mean friction coefficient. This friction coefficient is expect to be of a similar nature as the coefficient of static friction obtained between two solid surfaces in contact. Taking into account that the real (microscopic) contact between two surfaces is actually discontinuous, Bagnold (1973) claims that this analogy holds for the case of saltating particles, in which the contact between the particles and the surface is intermittent. Therefore, similar to a coefficient of static friction, the ratio $\langle a_p \rangle / g$ is expected to be a function of the particle-bed interaction only and, consequently, invariant with respect to $u_*$.

In the saltation model proposed by Pomeroy and Gray (1990), an expression for the transport rate of saltating snow is also obtained by estimating the ratio $\langle a_p \rangle / g$. However, differently from Bagnold (1973), $\langle a_p \rangle$ was found to increase linearly with $u_*$. In addition, it was found to be highly variable with the snow surface characteristics. In the model, $\langle a_p \rangle$ was not estimated from the direct measurements of $\langle \Delta v_{x,s} \rangle$ and $\langle t_p \rangle$, but it was deduced from fitting experimental measurements of the particle mass flux in saltation with Eq. (2). Therefore, the resultant scaling of $\langle a_p \rangle$ with $u_*$ is a function of the parameterizations considered for $\tau_s$ and $\langle v_p \rangle$, as well as of the accuracy of the mass flux measurements and the respective vertically integrated value, $Q$.

The mass flux measurements considered in the model were performed at Saskatoon, Canada, using an optoelectronic snow particle detector placed at approximately 2 cm height above the snow surface. This device counts the number of particles crossing an infrared laser beam. The mass flux is then estimated by assuming a particle size distribution given by a two-parameters Gamma distribution, as suggested by Schmidt (1981), and a particle density equal to the density of ice ($\sim 917$ kg m$^{-3}$). The mass flux in saltation is assumed to be uniform in height. Therefore, the transport rate is estimated from the product of the measured mass flux and the height of the saltation layer, $h_{salt}$ [m]. As addressed in Sect. 2.3.3, the vertical profile of the particle mass flux exhibits strong gradients close to the surface. In this way, the saltation mass flux is hardly defined by a single

point measurement and the assumption of a constant mass flux along the saltation layer might lead to errors in the assessment of the transport rate.

In summary, Pomeroy and Gray (1990) arrived to the conclusion that $\langle a_p \rangle$ scales with $u_*$ by assuming that the mass flux is uniform in height, that $h_{salt}$ scales with $u_*^2$ (Owen, 1964) and that $\langle v_p \rangle$ is invariant with respect to $u_*$. As a consequence, they concluded that $Q$ scales linearly with $u_*$. This result is not in agreement with most saltation models and was shown to highly underestimate the transport rate in comparison to other expressions proposed in the literature (Melo et al., 2022). Notwithstanding, this model is still widely used, specifically in the drifting snow schemes of RACMO (van Wessem et al., 2018) and MAR (Amory et al., 2021). The model of Pomeroy and Gray (1990) is one of the few parameterizatons for the transport rate that take into account field measurements of snow saltation. However, it is highly dependent on the assumptions made for the particle mass flux and streamwise velocity profiles. These will be discussed in Sect. 2.3.

If both $\langle \Delta v_{x,s} \rangle$ and $\langle t_p \rangle$ are invariant with respect to $u_*$, as suggested by several models and measurements (see Sect. 2.2.2 and 2.2.3), $\langle a_p \rangle$ is mainly expected to be invariant with respect to $u_*$, as early hypothesized by Bagnold (1973). Taking into account that the direct measurement of $\langle a_p \rangle$ is challenging (e.g., Araoka and Maeno, 1981), the variation of $Q$ with $u_*$ and the snow surface characteristics might be better understood from the direct study of $\langle \Delta v_{x,s} \rangle$, $\langle t_p \rangle$ and $\langle v_p \rangle$.

## 2.3 The vertical profiles of particle concentration, streamwise velocity and mass flux

The particle mass flux in saltation, $q$ [$\mathrm{kg\,m^{-2}\,s^{-1}}$], corresponds to the mass flow rate of particles moving in saltation per unit cross section area. The particle concentration, $c$ [$\mathrm{kg\,m^{-3}}$], is the mass of particles in saltation per unit volume. By definition, these two quantities are related by $q = c\bar{v}_x$, where $\bar{v}_x$ [$\mathrm{m\,s^{-1}}$] is the volume-averaged particle streamwise velocity (henceforth, particle streamwise velocity) at which the respective mass of particles in saltation travels across a given cross section area (note that the average of $\bar{v}_x$ along the saltation layer equals the average particle streamwise velocity, $\langle v_p \rangle$, defined in Sect. 2.2). During steady-state saltation, $q$, $c$ and $\bar{v}_x$ are invariant along the horizontal directions, but they are expected to vary with the height above the surface. In the following subsections, we discuss the parameterizations used to model these quantities, as well as the experimental, theoretical and numerical results that contributed to their development.

### 2.3.1 The particle concentration

An analytic expression for the vertical profile of particle concentration was derived by Kawamura (1951). In the model, the saltation dynamics is simplified by neglecting the aerodynamic forces on the vertical motion of saltating particles, by assuming that the probability distribution of the vertical component of the particle ejection velocity follows a half-normal distribution and that the saltating particles are uniformly sized. In this way, from a mass balance between the surface and a given height $z$, the particle concentration at each height can be estimated as a function of the probability distribution of the vertical ejection velocity and the vertical mass flux of particles leaving the surface. In Gordon et al. (2009), it is shown that the analytic expression of Kawamura (1951) is well approximated by an exponential decay of the form:

$$c(z) = c_o \exp\left(-\frac{z}{\delta_c}\right) \tag{3}$$

where $c_o$ [kg m$^{-3}$] is the particle concentration at the surface (by extrapolating the exponential profile down to the surface) and $\delta_c$ [m] is the decay height of the particle concentration profile. The decay height, $\delta_c$, defines the height above the surface for which the particle concentration is reduced to $c_o / \exp(1)$. The theoretical expression proposed by Kawamura (1951) tends to infinity at the surface. In contrast, Eq. (3) is expected to underestimate the particle concentration near the surface and to
provide a good approximation only for $z > \langle h_p \rangle$, where $\langle h_p \rangle$ [m] is the average hop height of particles in saltation.

The direct measurement of the particle concentration is challenging because it requires the measurement of the number and size of saltating particles in a well defined volume (Creyssels et al., 2009). Gordon et al. (2009) measured the number density of snow particles in saltation in the first 9 cm above the surface over sea ice in Franklin Bay, Canada. Images of saltating snow were recorded with a camera, which captured the light reflected by snow particles traveling in front of a black background. The
vertical profile of particle concentration was estimated by specifying the width over which the particles were detected (volume depth), by assuming that the particle size distribution is given by a two-parameters Gamma distribution and that the saltating particles have the density of ice. The resultant particle concentration profiles, obtained for friction velocities ranging from 0.25 to 0.5 m s$^{-1}$ and different snow surface conditions, followed mainly an exponential decay as predicted by the theoretical model of Kawamura (1951). However, when the particle concentration was low, the data was more scattered and deviations from the
exponential decay was seen above approximately 2 cm height above the surface.

From fitting the measured vertical profiles with Eq. (3), the respective decay height was estimated. According to Gordon et al. (2009), $\delta_c$ is expected to scale with the average hop height, $\langle h_p \rangle$. In the absence of aerodynamic forces, the hop height of a particle in saltation scales with $v_z^{e2}/g$. Therefore, $\delta_c$ is expected to scale with $\langle v_z^e \rangle^2/g$. Owen (1964) suggested that $\langle v_z^e \rangle$ scales with $u_*$ and, therefore, that $\langle h_p \rangle$ scales with $u_*^2$. However, the measurements of Gordon et al. (2009) revealed a weak correlation
between $\delta_c$ and $u_*^2$. Indeed, for each set of measurements conducted on the same day (similar snow surface conditions), the decay height did not increase monotonically with the friction velocity.

Similar results were found by Creyssels et al. (2009), after performing wind tunnel experiments with uniformly-sized sand particles. As mentioned in Sect. 2.2.2, the experiments were performed with a high-speed camera illuminated by a laser sheet. The particle concentration was computed in the first 5 cm above the surface for friction velocities ranging from 0.24 to 0.67
m s$^{-1}$. In the calculations, the volume depth was assumed equal to the width of the laser. The vertical profiles of particle concentration were found to follow an exponential decay of the form of Eq. (3) and the respective decay height was found to be invariant with respect to the friction velocity. The conclusions of Gordon et al. (2009) and Creyssels et al. (2009) regarding the scaling of $\delta_c$ with $u_*$ are in agreement with the assumption that the ejection velocity is invariant with respect to $u_*$, which is in line with most numerical models and experimental measurements developed over erodible surfaces (see Sect. 2.2.2).
Ho et al. (2011) used a similar technique to measure the particle concentration in the first 6 cm above the surface in a wind tunnel. In their experiments, both a bed composed of loose sand grains and a rigid (non-erodible) surface were used. In agreement with the experiments of Creyssels et al. (2009), the particle concentration over the erodible surface was found to

decrease with increasing height following an exponential decay and the computed decay height was invariant with respect to $u_*$. Differently, when saltation was allowed to develop over the rigid surface, the decay height was found to be higher and to
increase linearly with $u_*$. This result is related to the higher values of the particle slip velocity obtained over the rigid surface in comparison with the erodible one and to the increase of the particle slip velocity with the increase of $u_*$ obtained over the rigid surface, mentioned in Sect. 2.2.2.

Taking into account the challenges and uncertainties of particle concentration measurements, especially in the field, measurements of particle mass flux are more frequent (e.g., Namikas, 2003; Martin and Kok, 2017; Nishimura and Nemoto, 2005).
Therefore, the general approach in drifting snow schemes is to derive the particle concentration as a function of the particle streamwise velocity and mass flux (Amory et al., 2021; van Wessem et al., 2018; Vionnet et al., 2014; Sharma et al., 2023). These two quantities are discussed in the next sections.

### 2.3.2 The particle streamwise velocity

The particle streamwise velocity of saltating sand was measured by Creyssels et al. (2009) and Ho et al. (2011) in the first 5
and 3 cm above the surface, respectively (see details in Sect. 2.2.2). In this region of the saltation layer, the particle streamwise velocity, $\bar{v}_x$, was found to increase linearly with height. Over a bed of loose sand grains, the effect of the friction velocity on the particle slip velocity was almost negligible, as discussed in Sect. 2.2.2. Above the surface, $\bar{v}_x$ was found to increase slightly with the rise of $u_*$. Using also the PTV technique, Aksamit and Pomeroy (2016) measured the velocity of saltating snow in the first 2.5 cm above the surface in a snow-covered area. A linear increase of $\bar{v}_x$ with height was seen in the first 1 cm above the
surface. In addition, $\bar{v}_x$ was found to increase slightly with the friction velocity.

The particle streamwise velocity can also be measured with a Snow Particle Counter (SPC) (Nishimura et al., 2014). The SPC is an optical device that detects the particles that cross a 25 mm long, 2 mm high and 0.5 mm wide laser beam. In general, this device is used to retrieve the number and average diameter of all particles crossing the laser beam during a defined time period. Therefore, it is mainly used to estimate the particle mass flux. The particle streamwise velocity can be computed by
evaluating the time that each particle takes to cross the laser beam. Nishimura et al. (2014) measured the streamwise velocity of saltating snow along the first 10 cm above the surface in a wind tunnel for friction velocities ranging from 0.37 to 0.63 $\mathrm{m\,s^{-1}}$ using this technique. In agreement with the previous studies, the particle streamwise velocity was found to increase as the height above the surface and the friction velocity increase. However, differently from the previous PTV measurements, most of the measurement points were located above 5 cm height. In this way, the increase of $\bar{v}_x$ with increasing height did not
follow a linear trend, but rather a logarithmic one. Similar particle streamwise velocity profiles were found in the wind tunnel measurements performed by Nishimura and Hunt (2000), who recorded saltating snow particles and ice spheres in the first 10 cm above the surface with a video system, and by Rasmussen and Sørensen (2008), who used Laser Doppler Velocimetry (LDV) to measure the streamwise velocity of saltating sand in the first 8 cm above the surface.

Numerical models that represent the surface processes of splash and rebound have also obtained an increase of the particle
streamwise velocity with height (e.g., Kok and Renno, 2009; Durán et al., 2011). As expected, in the first 1 cm above the

surface, the particle velocity is mainly invariant with respect to $u_*$ and, at higher elevations, the particle streamwise velocity increases with the increase of $u_*$.

In the model of Pomeroy and Gray (1990), the particle streamwise velocity is assumed constant in height and characterized by its average value $\langle v_p \rangle$. $\langle v_p \rangle$ is assumed invariant with respect to $u_*$ and proportional to the impact threshold friction velocity,

$u_{*,it}$. This assumption can be found in the drifting snow schemes of RACMO, MAR, Meso-NH and CRYOWRF to compute the particle concentration at the top of the saltation layer as a function of the particle streamwise velocity and mass flux (Lenaerts et al., 2012; Amory et al., 2021; Vionnet et al., 2014; Sharma et al., 2023). The assumption that $\langle v_p \rangle$ scales with $u_{*,it}$ is justified by the observations of Bagnold (1941) regarding the existence of a focus point in the wind velocity profile. According to his observations, during steady-state saltation, the wind velocity inside the saltation layer obtained for different

values of $u_*$ converge to the same value at a given height close to the surface - the focus point. The wind velocity at the focus point scales with the impact threshold friction velocity and is, therefore, solely dependent on the bed characteristics. In this way, the particle streamwise velocity in the vicinity of the focus point is also expected to scale with $u_{*,it}$, which according to Owen (1964) is the value of the surface friction velocity during saltation (see Sect. 2.2.1). The existence of a focus point and the fact that the particle streamwise velocity close to the surface is invariant with respect to $u_*$ are known features of the

saltation system, obtained by different models that represent the surface processes of rebound and splash (e.g., Kok and Renno, 2009; Durán et al., 2011; Melo et al., 2022). However, as previously discussed, experimental measurements performed with both sand and snow show that the particle streamwise velocity above approximately 3 cm height increases with the increase of $u_*$. In this way, the parameterization used for $\bar{v}_x$, in particular, its scaling with $u_{*,it}$ and $u_*$, must be defined according to the height at which the particle streamwise velocity (and, subsequently, the particle concentration) is being computed.

The snow surface characteristics are also expected to influence the particle streamwise velocity, as previously discussed in Sect. 2.2.2. For instance, for the same range of friction velocities, Aksamit and Pomeroy (2016) found a higher particle streamwise velocity in the first 2.5 cm above the surface when saltation developed over a surface of fresh snow than over surfaces of old snow or fine decomposing grains. One can hypothesize that this is related to the smaller density and dendritic shape of the particles in saltation. The particles ejected from a fresh snow surface are not fully rounded and will continuously

fragment and get denser as the particles collide with the bed (Comola et al., 2017). Other snow surface characteristics, like the particle diameter and the existence of cohesive bonds between the snow particles, can also influence the saltation dynamics. When modeling the impact and splash of saltating snow particles, Comola and Lehning (2017) took into account the effect of inter-particle cohesion. In their model, a fraction of the kinetic energy of the impacting particle is used to break the cohesive bonds between each ejected particle and the surrounding ones. Melo et al. (2022) used this splash model to simulate saltating

snow and assessed the effect of the particle diameter and inter-particle cohesion on the particle streamwise velocity. Considering a bed of uniformly-sized particles, the particle streamwise velocity was found to decrease with the increase of the particle diameter. Moreover, by increasing the energy required to break the cohesive bonds, a decrease of the particle concentration and an increase of the particle streamwise velocity was found in the first 5 cm above the surface. In the model, the existence of inter-particle bonds leads to a decrease in the number of particles aloft. This small amount of particles are able to extract more

momentum from the flow and to reach higher velocities. The same trend was found by Comola et al. (2019a) when studying the

effect of inter-particle cohesion on steady-state saltation, using the Discrete Element Method (DEM) and a simple description of the wind velocity field.

### 2.3.3 The particle mass flux

The vertical profile of particle mass flux is expected to follow an exponential decay of the same form of Eq. (3) (e.g., Nalpanis et al., 1993; Namikas, 2003; Martin and Kok, 2017):

$$q(z) = q_o \exp \left( -\frac{z}{\delta_q} \right) \tag{4}$$

where $q_o$ [kg m$^{-2}$ s$^{-1}$] is the surface mass flux (by extrapolating the exponential profile down to the surface) and $\delta_q$ [m] is the decay height of the particle mass flux profile. By definition, the integral of $q$ along the height yields the transport rate, $Q$. Therefore, it follows that $q_o = Q/\delta_q$.

Experiments developed with sand show that the mass flux profile follows an exponential decay at moderate friction velocities, but does not at low values of $u_*$, close to the fluid threshold friction velocity (Nalpanis et al., 1993; Namikas, 2003). In addition, it was shown that the exponential decay tends to underestimate the mass flux in the first 2 cm above the surface (Namikas, 2003; Bauer and Davidson-Arnott, 2014). This might pose a problem when relating the fitted values of $q_o$ and $\delta_q$ with the transport rate, $Q$. Experiments developed with snow, both in wind tunnels (Sugiura et al., 1998; Sato et al., 2001) and in the field (Takeuchi, 1980; Nishimura and Nemoto, 2005), revealed an exponential decay of the mass flux profile in approximately the first 10 cm above the surface. Near the top of the saltation layer, deviations are seen due to the saltation-suspension transition (Gordon et al., 2009). The exponential decay of the particle mass flux with height contrasts with the uniform profile assumed by Pomeroy and Gray (1990), as discussed in Sect. 2.2.4.

Similarly to the decay height presented in Eq. (3), $\delta_q$ is also thought to scale with the average particle hop height in saltation and, therefore, with $\langle v_z^e \rangle^2 / g$ (Nishimura and Hunt, 2000). Based on the assumption made by Owen (1964) regarding the scaling of $\langle v_z^e \rangle$ with $u_*$, Nishimura and Hunt (2000) assumed that $\delta_q$ must scale with $u_*^2/g$. This parameterization for $\delta_q$ is currently used in Meso-NH (Vionnet et al., 2014) and CRYOWRF (Sharma et al., 2023) to compute the particle mass flux in saltation.

The decay height is estimated from fitting Eq. (4) to measured mass flux profiles. In Fig. 2, the values of $\delta_q$ obtained from different experiments developed with sand and snow are presented as a function of the friction velocity, $u_*$ (Sugiura et al., 1998; Namikas, 2003; Nishimura and Nemoto, 2005; Martin and Kok, 2017). The expression proposed by Nishimura and Hunt (2000) is also presented for comparison. The measurements of Sugiura et al. (1998) were performed in a wind tunnel using preserved natural snow, which was disintegrated into individual particles prior to the experiments. The particle mass flux was measured between 1.6 and 6.1 cm above the surface using a SPC. By detecting the number and average diameter of saltating particles, the mass flux is computed by approximating the snow particles to spheres and by assuming a given particle density (generally, 917 kg m$^{-3}$). As seen in Fig. 2, the values of $\delta_q$ obtained for friction velocities ranging from 0.15 to 0.39 m s$^{-1}$ were found to increase with $u_*^2$, as suggested by Nishimura and Hunt (2000).

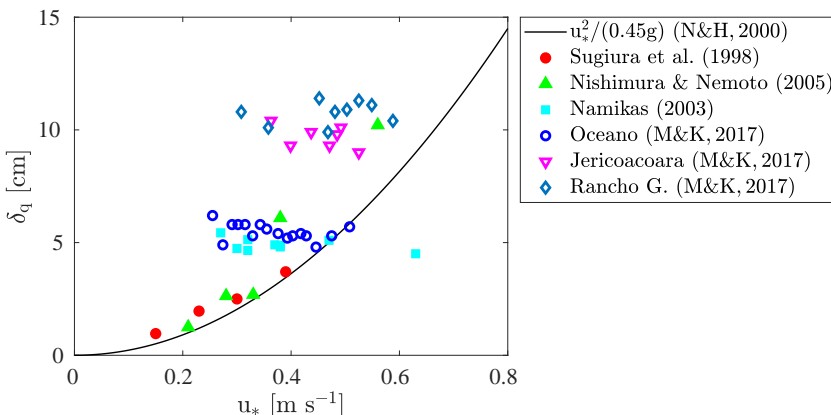

**Figure 2.** Decay height, $\delta_q$, as a function of the friction velocity, $u_*$. Results reported by Sugiura et al. (1998) after wind tunnel experiments conducted with snow and by Namikas (2003) and Martin and Kok (2017) (M&K, 2017) after field experiments conducted with sand. The values of $\delta_q$ referring to the field measurements of Nishimura and Nemoto (2005) conducted with snow were not reported in their work. We have estimated them by fitting the mass flux measurements between 2 and 12 cm height above the surface with Eq. (4). The expression proposed by Nishimura and Hunt (2000) (N&H, 2000) is also drawn for comparison.

The results of Sugiura et al. (1998) are in agreement with the wind tunnel experiments of Sato et al. (2001) performed with two different snow covers. Similarly to the experiments of Sugiura et al. (1998), snow particles were obtained by disintegrating a layer of preserved natural snow. Experiments were performed immediately after spreading the snow particles over the wind
tunnel floor (loose snow cover) and after some time, which allowed sintering to take place (hard snow cover). The vertical profile of particle mass flux followed an exponential decay over both snow surfaces and the decay height was found to increase with the wind velocity. However, significantly different values of $\delta_q$ were obtained with each snow cover: for the same range of wind velocities, $\delta_q$ varied between, approximately, 0.9 and 3 cm over the loose snow cover and between, approximately, 3 and 8 cm, over the hard snow cover. Even though the hard snow cover cannot be considered an erodible surface (particle
ejection from the surface was rare and the mass flux in saltation was highly dependent on the seeding rate of snow particles at the upwind end of the tunnel), the results of Sato et al. (2001) highlight the influence of the snow surface characteristics on the saltation dynamics and, in particular, on the decay height. The decay height obtained by Sato et al. (2001) is not presented in Fig. 2 because the respective value of $u_*$ was not provided by the authors.

The mass flux measurements of Nishimura and Nemoto (2005) were performed in Mizuho, Antarctica, using SPCs. They
reported mass flux profiles obtained for friction velocities ranging from 0.21 to 0.56 $\mathrm{m\,s^{-1}}$ (please note that the mass flux measurements reported by Nishimura and Nemoto (2005) in Fig. 6 must be multiplied by 10 to be in agreement with the units - K. Nishimura, personal communication, November 22, 2023). The respective $\delta_q$ values are not given in the article, but these can be estimated by fitting the mass flux measurements between 2 and 12 cm height to Eq. (4). From Fig. 2, one can conclude that $\delta_q$ increases with $u_*$. In addition, considering the three lowest friction velocities, a fair agreement is obtained between the

computed values of $\delta_q$ and the expression of Nishimura and Hunt (2000). At higher friction velocities, higher values of $\delta_q$ are obtained in comparison with the quadratic expression. However, without a detailed characterization of the snow cover during the different events (which spanned a time period of 1.5 months), it is not possible to evaluate the effect of the friction velocity alone.

    When performing field and wind tunnel experiments with sand, a different trend is obtained. For instance, Namikas (2003)
measured the particle mass flux along the first 35 cm above the surface using wedge-shaped sediment collectors. The measurements were performed for friction velocities ranging from 0.27 to 0.63 $\mathrm{m\,s}^{-1}$. From fitting the measured profiles with Eq. (4), similar values of $\delta_q$ were obtained. Therefore, it was concluded that $\delta_q$ is invariant with respect to $u_*$. The same conclusion was reached by Martin and Kok (2017) after performing field experiments in three different locations: Oceano, Rancho Guadalupe and Jericoacoara. The particle mass flux was measured with Big Spring Number Eight (BSNE) sand traps between
approximately 15 to 60 cm height above the surface. Even though this height range is above the saltation layer in the case of snow saltation, the mass flux of saltating sand still follows an exponential decay in this region (Martin et al., 2018). Based on their results, Martin and Kok (2017) suggest that $\delta_q$ rather scales with the particle diameter. This would justify the agreement between the measurements of Martin and Kok (2017) at Oceano and those from Namikas (2003), which were performed at a similar location. Bigger sand grains were reported at Rancho Guadalupe and Jericoacoara in comparison to Oceano (Mar-
tin and Kok, 2017). Similarly, Nalpanis et al. (1993) could not verify the scaling of $\delta_q$ with $u_*^2$ after performing wind tunnel experiments with sand and numerical simulations.

    By measuring the particle concentration and streamwise velocity, Creyssels et al. (2009) found that the mass flux evolution is mainly driven by the particle concentration profile. Therefore, even though the particle streamwise velocity was found to increase linearly with height in the first 5 cm above the surface, $\delta_c$ and $\delta_q$ were found to be approximately equal. In this way,
the scaling of $\delta_c$ with $u_*$, reported in Sect. 2.3.1, is expected to reflect the scaling of $\delta_q$ with $u_*$. In agreement with the mass flux measurements performed with sand, Creyssels et al. (2009) and Gordon et al. (2009) did not arrive to an increase of $\delta_c$ with the increase of $u_*$. In contrast, when saltation was allowed to develop over a rigid surface, Ho et al. (2011) obtained a linear increase of $\delta_c$ with $u_*$. Despite the findings of Creyssels et al. (2009) regarding the equality between $\delta_c$ and $\delta_q$, the values of $\delta_q$ presented in Fig. 2 and the values of $\delta_c$ presented in the works of Creyssels et al. (2009), Ho et al. (2011) and Gordon
et al. (2009) differ in approximately one order of magnitude. In this way, the comparison between these two quantities must be performed with care.

    The scaling of $\delta_q$ with $u_*$ can also be evaluated from vertical profiles of particle number flux. Considering saltating particles of uniform size, the number flux is proportional to the mass flux and, therefore, expected to follow an exponential decay of the form of Eq. (4). Aksamit and Pomeroy (2016) computed the number flux of snow particles in saltation in the first 1 cm above
the surface from the recordings of high-speed images of saltating snow in a snow-covered area. Even though the observed particle size distribution was not uniform but characterized by a Gamma distribution, the decay height of the number flux profile is not expected to deviate significantly from $\delta_q$. Similarly to the results obtained by Gordon et al. (2009) over sea ice, a clear scaling of the decay height with $u_*$ was not found: it either increased or decreased with the rise of the friction velocity depending on the snow cover characteristics (please note that the values reported by Aksamit and Pomeroy (2016) in Figs. 7d-f

do not correspond to $l_v$ [mm], named decay length, but to $1000/l_v$ - N. O. Aksamit, personal communication, February 11, 2023). In addition, the values obtained are one order of magnitude lower than those reported by Sugiura et al. (1998) for the same range of friction velocities and of the same order of magnitude as those reported in Gordon et al. (2009).

The uncertainty regarding the scaling of $\delta_q$ with $u_*$ can, in part, be linked to the uncertainty regarding the scaling of $\langle v_z^e \rangle$ with $u_*$ when saltation develops over a snow surface (see Sect. 2.2.2). The idea that $\delta_q$ is invariant with respect to $u_*$ is in
agreement with the assumption that the ejection velocity is invariant with respect to $u_*$. The increase of $\delta_q$ with the rise of $u_*$ obtained by Sugiura et al. (1998) and Nishimura and Nemoto (2005) is not in agreement with this assumption and contrasts with the results of Gordon et al. (2009) and Aksamit and Pomeroy (2016) regarding the scaling of the decay height of the particle concentration and number flux profiles with respect to $u_*$. These contradictory results remain to be understood. In addition, more experimental studies are needed to understand and quantify the effect of the snow surface characteristics on $\delta_q$.

## 3   Numerical simulations

In this section, we use a numerical model to investigate the main quantities of interest in snow saltation modeling, previously discussed in Sect. 2. The numerical model comprises a Large Eddy Simulation flow solver, a Lagrangian stochastic model for the particle trajectories and a set of parameterizations that represent the processes of aerodynamic entrainment, rebound and splash (Sharma et al., 2018; Comola et al., 2019b; Sigmund et al., 2022; Melo et al., 2022). A brief description of the model
is given in Sect. 3.1. Detailed numerical models, like the one described, cannot be used in atmospheric models, which require simple and computationally light algorithms. Nevertheless, they provide an excellent test bench for simple parameterizations and their underlying assumptions. The description of the numerical setup is presented in Sect. 3.2 and the main results are presented and discussed in Sect. 3.3.

### 3.1   Model description

The turbulent wind field is described by the continuity and Navier-Stokes equations and assumed incompressible. Following the Large Eddy Simulation (LES) technique, a filtering operator is applied to the equations in order to resolve the turbulent structures larger than the computational grid size. The turbulent structures smaller than the grid size are not resolved, but their effect on the larger structures is parameterized with a sub-grid scale (SGS) model. The numerical implementation of the LES model is based on the work of Albertson and Parlange (1999), which was continuously developed for different boundary
layer studies (e.g., Giometto et al., 2016, 2017; Sharma et al., 2017). In the model, we consider the Lagrangian-averaged scale-dependent SGS model proposed by Bou-Zeid et al. (2005). The computational domain is of cuboid shape, in which periodic boundary conditions are imposed at the vertical walls. Impermeability and zero vertical gradients are assumed at the top boundary. At the bottom boundary, the logarithmic law of the wall is specified as well as the impermeability condition. The flow is driven by a constant streamwise pressure gradient equal to $-\rho_a u_*^2/L_z$, where $L_z$ [m] is the domain height. In the
absence of saltating particles, this imposed pressure gradient guarantees a time-averaged surface shear stress equal to $\rho_a u_*^2$.

The particle trajectories within the turbulent flow are modeled with a Lagrangian stochastic model (LSM) (Comola, 2017). The particle motion is computed with Newton's second law, considering the effect of gravity and aerodynamic drag. The particles are assumed spherical and the drag coefficient is computed with the expression of Schiller and Nauman (1933). Other forces applied to the particles (e.g., aerodynamic lift, electrostatic force, virtual mass) are neglected in the model. The feedback of the particles on the flow momentum is taken into account by the addition of a reaction force in the filtered Navier-Stokes equations. It is equal in magnitude and opposite to the aerodynamic drag applied to the particles.

The surface processes of aerodynamic entrainment, rebound and splash occur at the bottom boundary, which is assumed erodible and composed of particles with a predefined size distribution. The surface processes are described by physical and statistical models, as proposed by Groot Zwaaftink et al. (2014) and Comola and Lehning (2017). For instance, aerodynamic entrainment of surface particles is expected to occur when the surface shear stress is greater than the fluid threshold (Bagnold, 1941):

$$\tau_{ft} = A^2 \left( \rho_p - \rho_a \right) g \bar{d} \tag{5}$$

where $A = 0.1$ is the fluid threshold coefficient, $\rho_p$ [kg m$^{-3}$] is the particle density and $\bar{d}$ [m] is the average particle diameter of the erodible bed. The number of entrained particles is proportional to the difference $\tau_s - \tau_{ft}$ (Anderson and Haff, 1991; Doorschot and Lehning, 2002). Following Clifton and Lehning (2008), it is assumed that the ejection velocity of an aerodynamically entrained particle varies according to a lognormal distribution. The mean and standard deviation of the distribution are assumed to be proportional to $u_{*,s}$. The initial particle concentration at the surface is considered high enough so that there is never a shortage in the supply of erodible particles.

Following a particle impact with the granular surface, the impacting particle can rebound and some particles can be splashed from the surface. The probability of rebound increases as the particle impact velocity increases (Anderson and Haff, 1991) and the rebound velocity is given by $\sqrt{\epsilon_r} |v^i|$, where $\epsilon_r = 0.25$ is the fraction of kinetic energy retained by the rebounding particle (restitution coefficient) and $|v^i|$ [m s$^{-1}$] is the modulus of the impact velocity (Doorschot and Lehning, 2002). The number of splashed particles, $N$, is computed with the model of Comola and Lehning (2017). In their model, the balances of kinetic energy and momentum of the collision are resolved by statistically representing the kinetic energy and momentum of the splashed particles by their mean values. As a result, the number of splashed particles varies with the characteristics of the impacting particle (mass and velocity) and the statistical characteristics of the splashed particles (mean and standard deviation of the particle size and ejection velocity). We assume that the ejection velocity of splashed particles varies according to an exponential distribution. Its mean value is a function of the particle impact velocity and is given by $0.25 |v^i|^{0.3}$ (Sharma et al., 2018). In the model, $N$ is also a function of other parameters that describe the granular bed as, for example, the restitution coefficient, the energy required to break the cohesive bonds between each ejected particle and the surrounding particles, and the fraction of impact kinetic energy and momentum that is lost due to friction. The full description of the LES-LSM model can be found in previous works (e.g., Sharma et al., 2018; Comola et al., 2019b; Melo et al., 2022).

### 3.2 Numerical settings

The computational domain consists of a cube with 6.4 m side length, uniformly discretized in each horizontal direction in 64 cells. The vertical direction is discretized in 128 cells following a hyperbolic stretching, which guarantees a more refined mesh close to the bottom boundary. The simulations are performed for 350 s. The flow is allowed to develop over the first 25 s of the simulations. The start of surface erosion is only allowed after this first time period. The time step of the LES solver is set to $5 \times 10^{-5}$ s.

In order to reproduce the snow surface characteristics reported in Sugiura et al. (1998), the particle size distribution of the granular bed is assumed normal, with a mean diameter of 360 $\mu$m and a standard deviation of 140 $\mu$m. A minimum and maximum particle diameter is imposed, equal to 30 $\mu$m and 2 mm, respectively. In the experiments of Sugiura et al. (1998), the wind tunnel was maintained at -15°C. Therefore, in the simulations, the air density and kinematic viscosity, $\nu$ [m$^2$ s$^{-1}$], are set to the respective values at that temperature ($\rho_a = 1.37$ kg m$^{-3}$, $\nu = 1.2 \times 10^{-5}$ m$^2$ s$^{-1}$). Moreover, snow sublimation is neglected. Taking into account that the snow layer was composed of preserved natural snow that was disintegrated into individual particles prior to the experiments, the effect of inter-particle cohesion is neglected in the simulations. The remaining rebound and splash parameters are set to the values reported in Melo et al. (2022). In addition, the particle density is assumed equal to 917 kg m$^{-3}$, as suggested by Sugiura et al. (1998). The roughness length, $z_o$, is not reported in Sugiura et al. (1998) and a value of $10^{-4}$ m is assumed (Clifton et al., 2006).

A set of seven simulations is performed for which the friction velocity is varied between 0.15 and 0.7 m s$^{-1}$. In order to decrease the computational time, the time step of the LSM solver is set to $10^{-4}$ s (twice the LES time step) for the friction velocities ranging between 0.15 and 0.5 m s$^{-1}$ and to $2 \times 10^{-4}$ s (four times the LES time step) for the higher friction velocities. In addition, the particles in saltation are grouped in parcels, composed of particles with equal diameter and following the same trajectory. Particles from the same parcel were aerodynamically entrained or splashed at the same surface location and time step. In order to guarantee that the simulation results are not affected by this approximation, the number of particles per parcel is set to 5, 50 and 100 for the friction velocities equal to 0.15, 0.23 and 0.3 m s$^{-1}$, respectively, and to 200 for the higher friction velocities.

### 3.3 Results and discussion

The vertical profiles of particle mass flux and velocity are computed by dividing the computational domain in horizontal layers. At each layer, the mass flux is computed by:

$$q(z_k) = \frac{\sum_{n=1}^{N_k} m_n v_{x_n}}{L_x L_y \Delta z_k} \tag{6}$$

where $z_k$ [m] and $\Delta z_k$ [m] are the average height above the surface and the thickness of layer $k$, $N_k$ is the total number of particles in layer $k$, $m_n$ [kg] and $v_{x_n}$ [m s$^{-1}$] are the mass and the instantaneous streamwise velocity of the $n^{th}$ particle in the respective layer, and $L_x$ [m] and $L_y$ [m] are the domain length and width. The particle streamwise velocity at each layer

is given by the average instantaneous particle velocity considering all particles in the layer. The vertical profiles are time-averaged over the last 100 s of the simulations. During this time period, negligible changes in the total mass of particles aloft were obtained and saltation was assumed to be in steady state.

The obtained vertical profiles of particle mass flux are presented in Fig. 3, together with the SPC measurements performed by Sugiura et al. (1998). The results obtained for friction velocities ranging from 0.15 to 0.5 $\mathrm{m\,s^{-1}}$ are presented in a linear plot (Fig. 3a) and those obtained for all simulated friction velocities are presented in a semi-logarithmic plot (Fig. 3b). Curves fitting the mass flux profiles with Eq. (4) are also presented. It can be seen that most of the mass flux profiles follow an exponential decay in approximately the first 8 cm above the surface. The only exception is the mass flux profile obtained with the lowest friction velocity of 0.15 $\mathrm{m\,s^{-1}}$. In this case, the friction velocity outside the saltation layer is slightly lower than the assumed fluid threshold friction velocity (from Eq. (5), $u_{*,ft} = 0.154 \mathrm{\ m\,s^{-1}}$). Therefore, saltation is highly intermittent. Sugiura et al. (1998) have also reported a fluid threshold friction velocity close to 0.15 $\mathrm{m\,s^{-1}}$. During the experiments, saltation was only sustained at this low friction velocity with the addition of seeding particles at the upwind end of the wind tunnel. For the remaining friction velocities, deviations are seen between the modeled mass flux profiles and the exponential decay in approximately the first 1 cm above the surface and above 6-12 cm height, depending on the friction velocity. The deviations from an exponential decay above 6-12 cm height are due to the prevalence of particles in suspension. As discussed in Sect. 2.3.3, these trends were also observed in field measurements of snow and sand saltation (Namikas, 2003; Bauer and Davidson-Arnott, 2014; Nishimura and Nemoto, 2005).

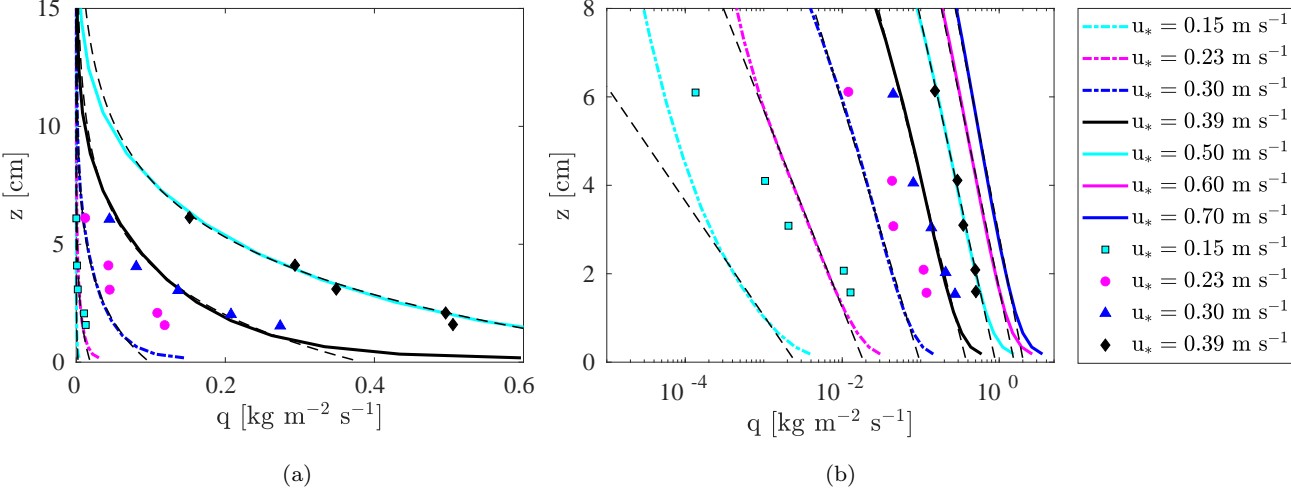

**Figure 3.** Vertical profile of particle mass flux obtained for different friction velocities. The black dashed lines are fits between the computed profiles and Eq. (4). The symbols denote the mass flux measurements performed by Sugiura et al. (1998). a) Linear plot (the mass flux profiles obtained for $u_*$ equal to 0.6 and 0.7 $\mathrm{m\,s^{-1}}$ are not represented in the figure), b) Semi-logarithmic plot.

When comparing the modeled mass flux profiles with the SPC measurements of Sugiura et al. (1998) obtained at the same friction velocities, it is clear that the numerical model is significantly underestimating the mass flux in saltation. This discrepancy could not be solved by decreasing the roughness length from $10^{-4}$ to $10^{-5}$ m, which is more in agreement with previous wind tunnel experiments developed with snow (Nishimura et al., 2014). Negligible improvements were also obtained by decreasing the domain height to half the height of the wind tunnel (please note that the top boundary of the domain is a symmetry plane). Despite the qualitative agreement, a quantitative agreement between simulation results and wind tunnel measurements certainly requires an adjustment in the parameters that control the surface processes of aerodynamic entrainment, rebound and splash. Even though a fitting exercise could be done to achieve a better agreement, we believe that detailed measurements of the near-surface particle-bed interaction and of the snow particle characteristics in saltation are needed for a further understanding of the near-surface processes and the development of improved parameterizations.

From fitting the computed mass flux profiles to Eq. (4), the decay height can be obtained. It is presented in Fig. 4 as a function of the friction velocity. The values obtained by Sugiura et al. (1998) after fitting the measured mass flux profiles to Eq. (4) and the expression proposed by Nishimura and Hunt (2000) are also presented for comparison. For friction velocities ranging from 0.15 to 0.3 m s$^{-1}$, there is a fair agreement between the decay height obtained from measured and modeled mass flux profiles. The decay height increases with the increase of the friction velocity and does not deviate considerably from the evolution proposed by Nishimura and Hunt (2000). However, for friction velocities greater than 0.3 m s$^{-1}$, the values of decay height obtained from the modeled mass flux profiles increase with the increase of the friction velocity at a much lower rate. In particular, the value obtained for $u_* = 0.39$ m s$^{-1}$ is lower than the value obtained by Sugiura et al. (1998) and a significant deviation between the modeled values and the expression proposed by Nishimura and Hunt (2000) is seen.

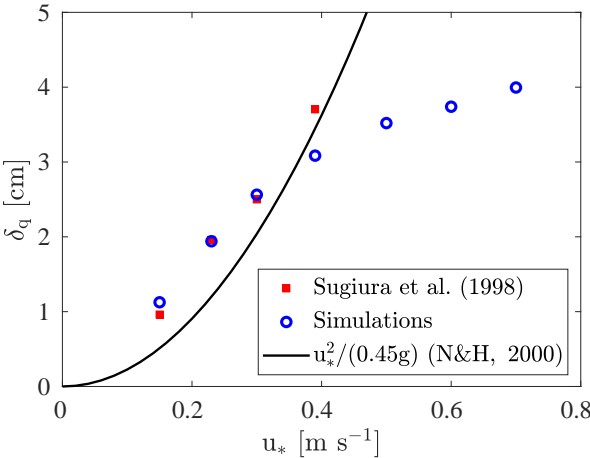

**Figure 4.** Decay height obtained by fitting the mass flux profiles presented in Fig. 3 with Eq. (4). The values reported by Sugiura et al. (1998) and the expression proposed by Nishimura and Hunt (2000) (N&H, 2000) are also presented for comparison.

Overall, the decay height obtained with the simulations increases logarithmically with $u_*$. At low friction velocities, the high rate of increase of $\delta_q$ with $u_*$ suggests that the computed vertical ejection velocity scales with $u_*$, as proposed by Owen (1964) and hypothesized by Nishimura and Hunt (2000) (see discussion in Sect. 2.3.1 and 2.3.3). In Fig. 5, we present the computed vertical profiles of the particle streamwise velocity (Fig. 5a) and of the vertical velocity of upward moving particles (Fig. 5b). In the insets, the respective near-surface values are presented as a function of the friction velocity: $\bar{v}_x^{e,i}$ [m s$^{-1}$] and

$\bar{v}_z^e$ [m s$^{-1}$] can be regarded as the particle slip velocity and the average vertical ejection velocity, respectively. For friction velocities ranging from 0.15 to 0.3 m s$^{-1}$, the particle streamwise velocity increases with height and with the friction velocity. The same trend is found for the vertical velocity of upward moving particles in the first 2 cm above the surface. At these low values of $u_*$, the average vertical ejection velocity and particle slip velocity also increase with the rise of the friction velocity. Therefore, the obtained increase in decay height as the friction velocity rises from 0.15 to 0.3 m s$^{-1}$ is indeed accompanied by

an increase of the vertical ejection velocity.

    Conversely, a different trend is found for $u_*$ ranging between 0.39 and 0.7 m s$^{-1}$. For instance, an increase of the streamwise and vertical velocities with the increase of the friction velocity is only seen above approximately 1.5 cm height above the surface. Below this height, the particle velocity profile is mainly invariant with respect to $u_*$, which is in agreement with several saltation models (e.g., Kok and Renno, 2009; Niiya and Nishimura, 2022) and measurements carried out over sand

(Creyssels et al., 2009; Ho et al., 2011), discussed in Sect. 2.2.2 and 2.3.2. From the insets of Fig. 5, it can be seen that the near-surface velocity actually decreases with the increase of the friction velocity.

    In the model, while the average ejection velocity of splashed particles scales with the particle impact velocity, the average ejection velocity of aerodynamically entrained particles scales with the surface friction velocity, $u_{*,s}$. The average vertical

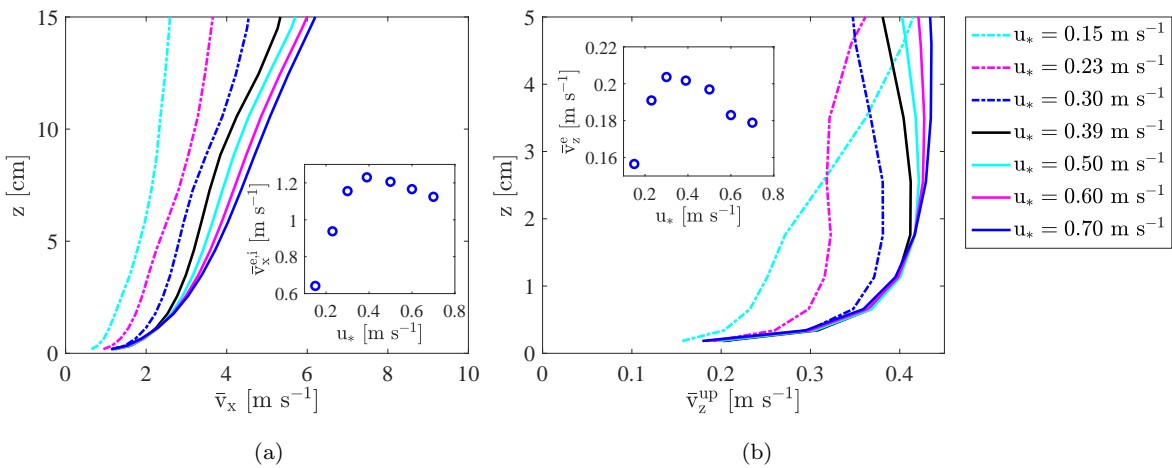

(a)                                 (b)

**Figure 5.** a) Vertical profiles of the particle streamwise velocity. b) Vertical profiles of the vertical velocity of upward moving particles. Results obtained for different friction velocities. The respective near-surface values are presented in the inset as a function of the friction velocity.

mass flux of aerodynamically entrained and splashed particles are presented in Fig. 6a and the average surface friction velocity is presented in Fig. 6b, as a function of the friction velocity. Both quantities are computed by performing a surface average over the erodible bed and a time average over the last 100 s of the simulations. In Fig. 6a, it is seen that the vertical mass flux of aerodynamically entrained particles is greater than the vertical mass flux of particles ejected via splash for $u_*$ equal to 0.15 and 0.23 $\mathrm{m\,s^{-1}}$. For $u_* = 0.3$ $\mathrm{m\,s^{-1}}$, splash entrainment is more significant than aerodynamic entrainment, but they are both of the same order of magnitude. For these three simulations, in which aerodynamic entrainment plays an important role during steady-state saltation, the average surface friction velocity does not deviate significantly from the friction velocity outside the saltation layer (Fig. 6b). Therefore, for $u_*$ ranging from 0.15 to 0.3 $\mathrm{m\,s^{-1}}$, the increase of the vertical ejection velocity with the rise of $u_*$ is a direct consequence of the respective increase of the surface friction velocity.

For friction velocities greater than 0.3 $\mathrm{m\,s^{-1}}$, the vertical mass flux of splashed particles is at least one order of magnitude greater than the vertical mass flux of aerodynamically entrained particles (Fig. 6a). In addition, the surface friction velocity deviates considerably from the friction velocity outside the saltation layer (Fig. 6b). It reaches a value smaller than $u_*$, which decreases slightly with the rise of the friction velocity.

Even though the average ejection velocity of splashed particles scales with the velocity of the impacting particle and not directly with $u_{*,s}$, the computed particle impact velocity is highly correlated with the near-surface wind velocity (and, therefore, with the surface friction velocity) because of the exchange of momentum between the fluid and the particles close to the surface. The vertical profiles of the streamwise wind velocity are presented in Fig. 7. They are computed by averaging the streamwise wind velocity along horizontal planes and over the last 100 s of the simulations. Overall, similar features are seen in comparison

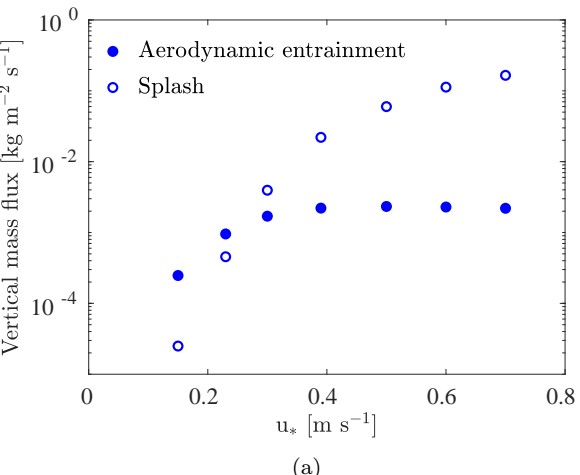
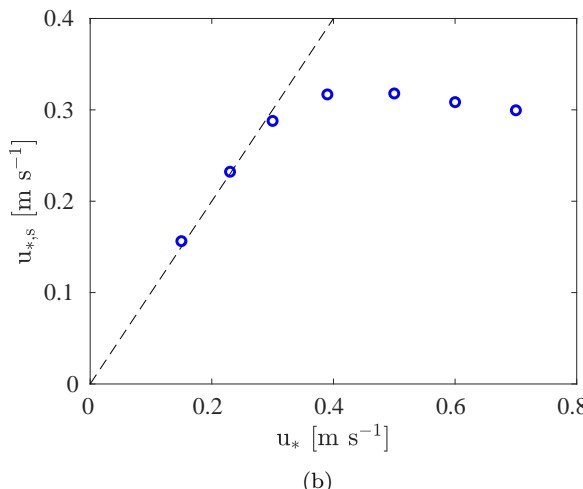

**Figure 6.** a) Vertical mass flux of aerodynamically entrained and splashed particles. b) Surface friction velocity (the dashed line is the 1:1 slope). The results presented are averages over the surface and over the last 100 s of the simulations. They are presented as a function of the friction velocity.

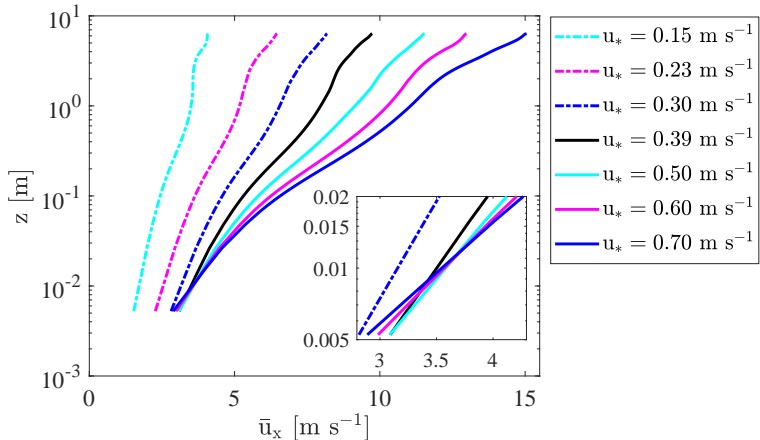

**Figure 7.** Vertical profiles of streamwise wind velocity obtained for different friction velocities during steady-state saltation. The inset is a zoom-in to the near-surface region.

to Fig. 5a: for friction velocities ranging from 0.15 to 0.3 $\mathrm{m\,s^{-1}}$, for which aerodynamic entrainment plays an important role, the streamwise wind velocity increases with height and friction velocity; however, at greater values of $u_*$, for which particle entrainment is dominated by splashed particles, the streamwise wind velocity profiles obtained during steady-state saltation exhibit a focus point at approximately 1 cm above the surface. As discussed in Sect. 2.3.2, the existence of a focus point was observed in wind tunnel experiments of sand saltation (Bagnold, 1941; Li and McKenna Neuman, 2012) and is supported by numerical models that represent the surface processes of rebound and splash (Kok et al., 2012; Durán et al., 2011). Due to the focus point, at high friction velocities, the near-surface wind velocity during steady-state saltation is approximately invariant with respect to $u_*$. Nevertheless, below the focus point, the streamwise wind velocity decreases slightly as $u_*$ increases (see zoom-in to the near-surface region in the inset of Fig. 7). Below the first grid point above the surface ($\sim 5\,\mathrm{mm}$), the wind velocity is assumed to follow a logarithmic profile characterized by a constant roughness length. Therefore, the obtained increase in streamwise wind velocity at the first grid point above the surface as the friction velocity rises from 0.15 to 0.3 $\mathrm{m\,s^{-1}}$ implies an increase in the surface friction velocity with respect to $u_*$ (Fig. 6b). Conversely, the slight decrease in streamwise wind velocity at the first grid point above the surface as $u_*$ rises above 0.39 $\mathrm{m\,s^{-1}}$ justifies the slight decrease in $u_{*,s}$ as the friction velocity increases from 0.39 to 0.7 $\mathrm{m\,s^{-1}}$ observed in Fig. 6b. The lower the wind velocity in the vicinity of the bed, the lower the ability of the flow to accelerate the particles close to the surface. Therefore, the obtained decrease of near-surface particle velocity as the friction velocity rises above 0.39 $\mathrm{m\,s^{-1}}$ (insets in Fig. 5) is due to the respective decrease of streamwise wind velocity below the focus point with respect to $u_*$.

At friction velocities greater than 0.3 $\mathrm{m\,s^{-1}}$, the current numerical model does not confirm neither the scaling of the ejection velocity with the friction velocity outside the saltation layer, nor the scaling of the decay height with the vertical ejection velocity. In fact, even though the vertical ejection velocity decreases with the rise of the friction velocity for $u_* > 0.3\,\mathrm{m\,s^{-1}}$

(inset in Fig. 5b), the decay height increases monotonically with $u_*$ (Fig. 4). Therefore, the model proposed by Nishimura and Hunt (2000) might not be appropriate to parameterize the decay height during splash dominated saltation.

The obtained evolution of $u_{*,s}$ at low friction velocities presented in Fig. 6b contrasts with the results of sand saltation models. For instance, Kok and Renno (2009) neglected the surface process of aerodynamic entrainment and found $u_{*,s}$ to decrease monotonically with the increase of the friction velocity. Sand particles have a higher density than snow particles, which implies a higher fluid threshold. In addition, as discussed in Sect. 2.2.1, the surface shear stress is expected to be lower than the fluid threshold, which restricts the occurrence of aerodynamic entrainment during steady-state saltation, even at low friction velocities.

In contrast, similar features are seen in some snow saltation models in which aerodynamic entrainment is represented. For example, Nemoto and Nishimura (2004) found the surface friction velocity to increase monotonically with the rise of the friction velocity from 0.23 to 0.39 $\mathrm{m\,s}^{-1}$. The same increase in $u_{*,s}$ is seen in Fig. 6b for this range of friction velocities. The values of $u_{*,s}$ found by Nemoto and Nishimura (2004) are greater than the specified fluid threshold friction velocity, which suggests that aerodynamic entrainment also plays a role in their steady-state conditions. The process of aerodynamic entrainment is also taken into account in the numerical model of Doorschot and Lehning (2002). In their model, the parameterizations for aerodynamic entrainment and rebound are similar to those described in Sect. 3.1, but the saltation system is - for simplicity - assumed to be either entirely composed of aerodynamically entrained particles or of particles that continuously rebound from the surface (in their model, no distinction is made between rebound and splash). Similarly to the results presented in Fig. 6, Doorschot and Lehning (2002) found saltation to be dominated by aerodynamic entrainment at low friction velocities and to be dominated by rebounding/splashed particles at high friction velocities. In addition, at low friction velocities, the surface friction velocity increases with the rise of $u_*$ and a negligible deviation is seen between the values of $u_{*,s}$ and $u_*$ due to the low number of particles aloft. At high friction velocities, $u_{*,s}$ decreases slightly with the rise of $u_*$. In their model, the friction velocity at which the transition from aerodynamic entrainment to rebound/splash occurs depends primarily on the assumed restitution coefficient. This coefficient is expected to be highly dependent on the snow surface characteristics, which highlights the importance of the snow cover properties to the correct description of snow saltation and its scaling laws.

In the snow saltation model of Niiya and Nishimura (2022), $u_{*,s}$ is found to decrease slightly with the increase of the friction velocity from 0.24 to 0.3 $\mathrm{m\,s}^{-1}$. This differs from the increase of $u_{*,s}$ found in Fig. 6b for the same range of friction velocities. In their work, the fluid threshold friction velocity is assumed equal to 0.24 $\mathrm{m\,s}^{-1}$ and the surface friction velocity is found to be approximately equal to 0.2 $\mathrm{m\,s}^{-1}$. In comparison with the current simulations, Niiya and Nishimura (2022) assumed a higher value for the fluid threshold friction velocity (0.24 versus 0.154 $\mathrm{m\,s}^{-1}$) and a lower value for the roughness length ($10^{-5}$ versus $10^{-4}$ m). When decreasing $z_o$ from $10^{-4}$ to $10^{-5}$ m in the LES-LSM, we obtain a significant decrease in the surface friction velocity during splash dominated saltation. In this way, a decrease in $z_o$ and an increase in $\tau_{ft}$ reduce the range of friction velocities for which saltation is controlled by aerodynamic entrainment, which might justify the seemingly different results obtained with both models. In addition to the roughness length and the fluid threshold, other parameters that characterize the snow surface and highly influence the surface processes of aerodynamic entrainment, rebound and splash are expected to influence the importance of aerodynamic entrainment during steady-state saltation. The narrower the range of

friction velocities for which aerodynamic entrainment plays an important role, the narrower is the range of friction velocities for which the decay height is well approximated by the expression of Nishimura and Hunt (2000).

The modeled saltation system is further investigated by the analysis of the particle hop height and length. The average hop height and length of saltating particles are computed from the analysis of all particle hops during the last 1 s of simulations. During this time period, the location of all particles aloft was outputted by the model at very high frequency. In this analysis, a particle hop is defined as a fraction of a particle trajectory limited by two consecutive local minimums located close to the surface. The hop height is given by the maximum height attained by the particle along the respective hop. An example of a particle trajectory comprised of several hops is presented in Fig. 8a. The particles seldom reach the surface because the surface processes of rebound and splash are assumed to take place as soon as the particles reach a height lower than $z = 4\bar{d} = 0.14$ cm.

The resultant average hop height and length are presented in Fig. 8b as a function of the friction velocity. The average hop height increases with the rise of the friction velocity until $u_* = 0.5$ m s$^{-1}$. It increases with a higher rate for $u_*$ ranging from 0.15 to 0.3 m s$^{-1}$, which is in agreement with the evolution found for the vertical ejection velocity and the decay height (Figs. 4 and 5b). As $u_*$ increases above 0.3 m s$^{-1}$ and saltation becomes splash dominated, the average hop height increases at a lower rate with the rise of the friction velocity. It reaches a plateau at $u_* = 0.5$ m s$^{-1}$ of approximately 2.8 cm. For $u_* > 0.3$ m s$^{-1}$, even though the vertical ejection velocity decreases with the rise of the friction velocity (Fig. 5b), $\langle h_p \rangle$ is approximately invariant with respect to $u_*$. This apparent discrepancy might be related to the fact that the hop height does not depend solely on the vertical ejection velocity, but also on the vertical component of the aerodynamic force applied to a particle along its trajectory.

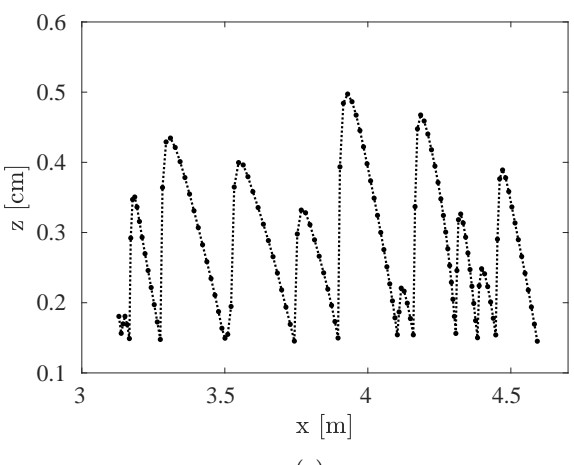

(a)

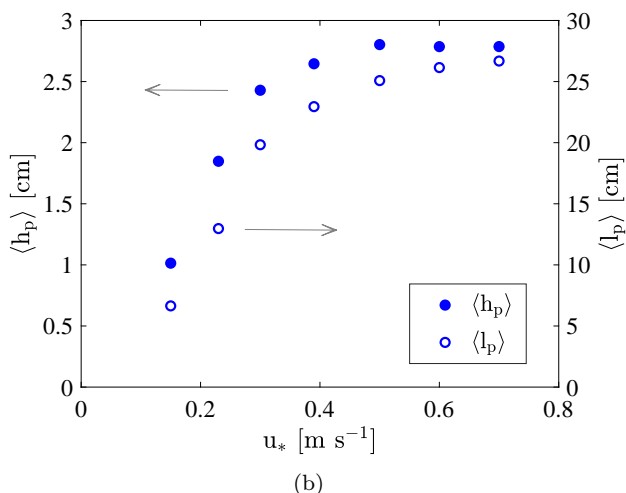

(b)

**Figure 8.** a) Example of a particle trajectory obtained for $u_* = 0.39$ m s$^{-1}$ (2D projection). b) Average hop height and length as a function of the friction velocity (the arrows indicate the respective y-axis of each set of values).

As discussed in Sect. 2.3.1 and 2.3.3, the decay height is expected to scale with the average hop height of particles in saltation (Nishimura and Hunt, 2000; Gordon et al., 2009; Martin and Kok, 2017). However, in the model, a linear relationship between $\langle h_p \rangle$ and $\delta_q$ is not found for $u_* > 0.3 \mathrm{~m~s}^{-1}$. The modeled saltation system is not only comprised of particles undergoing ballistic trajectories, but it also includes particles travelling without regular contact with the surface in the vicinity of the saltation-suspension interface. In this context, it is reasonable to assume that $\delta_q$ does not depend entirely on the hop height of

ballistic hops, but rather on the particle motion of all particles that contribute to the particle mass flux in the saltation layer.

   The average particle hop length, $\langle l_p \rangle$, increases continuously with the rise of the friction velocity. Similarly to the average hop height, it increases at a higher rate when $u_*$ varies between 0.15 and $0.3 \mathrm{~m~s}^{-1}$ than when $u_*$ is greater than $0.3 \mathrm{~m~s}^{-1}$ (Fig. 8b). In this way, when saltation is dominated by splash, the particle hop length can be approximated as invariant with respect to $u_*$, which is in agreement with the saltation model of Ungar and Haff (1987), discussed in Sect. 2.2.3. If we define the impact

threshold friction velocity as the minimum surface friction velocity for which saltation is dominated by splash (in this case, $u_{*,it} \approx 0.31 \mathrm{~m~s}^{-1}$), the studied values of friction velocity are within the low-velocity regime defined by Durán et al. (2011) when modeling sand saltation ($u_*/u_{*,it} < 4$). At these friction velocities, Durán et al. (2011) expect saltation to be dominated by splash and the average hop length to be invariant with respect to $u_*$. Indeed, when saltation is dominated by splash, the current results are in agreement with those from Durán et al. (2011). This more restrictive definition of the impact threshold

friction velocity (recall the definition of $u_{*,it}$ presented in section 2.2.1) is considered in some saltation models (e.g., Comola et al., 2022).

   If $\langle l_p \rangle$ is mainly invariant with respect to $u_*$, the transport rate is expected to scale with $u_*^2$. The transport rate is presented in Fig. 9 as a function of the friction velocity. It is computed by integrating the particle mass flux along the height, from the

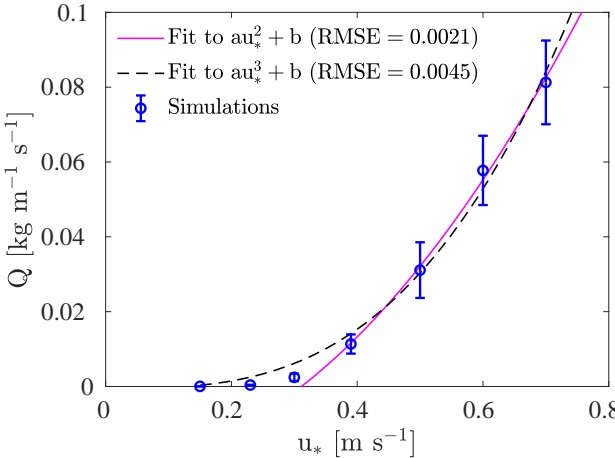

**Figure 9.** Transport rate as a function of the friction velocity. The curves represent fits between the results obtained for $u_* > 0.3 \mathrm{~m~s}^{-1}$ and quadratic and cubic functions. RSME denotes the root mean square error of the fit. The error bars are two times the standard deviation of the results.

surface to 15 cm height. The fits between the computed values obtained for $u_* > 0.3 \,\mathrm{m\,s^{-1}}$ and a quadratic function as well as
a cubic function are also presented. Indeed, the transport rate obtained during splash dominated saltation is in good agreement
with a quadratic curve, but the root mean square error (RMSE) of the cubic function is only slightly higher. When considering
the full range of friction velocities, a better agreement is actually found with the cubic function. These results highlight the
need to acquire mass flux measurements for a wide range of friction velocities and snow surface characteristics to fully evaluate
the scaling of the transport rate with the friction velocity.

The two saltation regimes obtained with the model (dominated by aerodynamic entrainment or splash) are illustrated in Fig.
10. In summary, as the friction velocity increases, the saltation system evolves from an aerodynamic entrainment dominated
system to one dominated by splashed particles. In the first regime, the particle streamwise velocity increases with the rise
of the friction velocity at all heights and the average hop height of saltating particles (represented by the line enclosing the
ballistic trajectories) increases as well. As the friction velocity reaches higher values, the average hop height stops increasing
significantly with the rise of the friction velocity. In fact, once saltation is dominated by splash, the average hop height is
approximately invariant with respect to $u_*$. In addition, the particle streamwise velocity is also mainly invariant with respect to
$u_*$ in the near-surface region where most saltating particles are. Considering the alternative definition for the impact threshold
friction velocity considered above, one can state that these two saltation regimes are typical of saltation systems characterized
by an impact threshold friction velocity, $u_{*,it}$, greater than the fluid threshold friction velocity, $u_{*,ft}$. As previously discussed,
this is not characteristic of sand saltation, but is probably representative of snow saltation for some snow surface characteristics.
In Fig. 10, the line indicating the average hop height of saltating particles separates the regions where particle motion is
controlled by wind-particle interaction only (above) and where it is highly dependent on the particle-bed interaction (below).

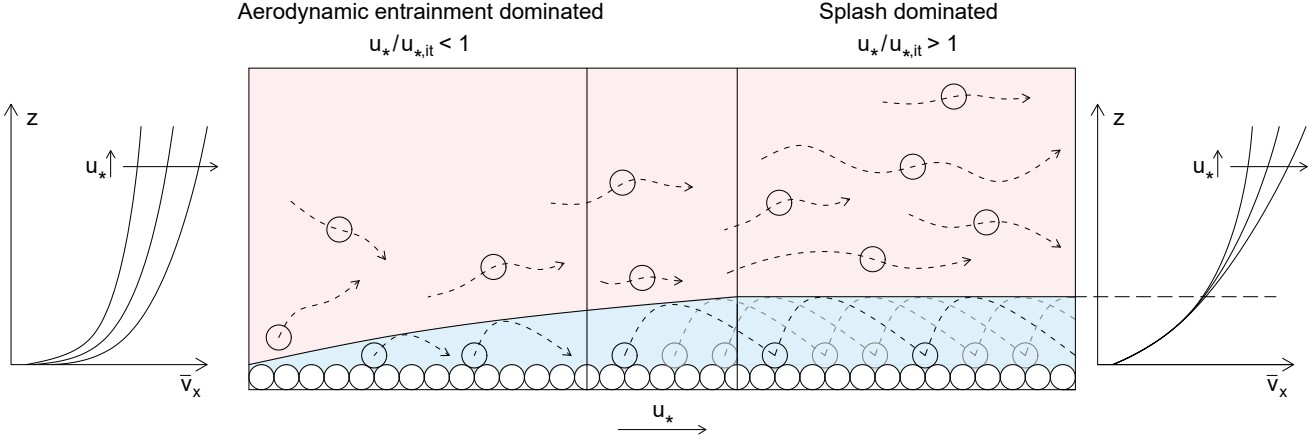

**Figure 10.** Illustration of the saltation system for increasing values of $u_*$: representation of the aerodynamic entrainment dominated and
splash dominated regimes. The line enclosing the ballistic trajectories indicates the average hop height of the saltating particles. The particle
streamwise velocity profiles characteristic of each regime are presented on the left and right-hand sides of the figure for increasing values of
$u_*$. The dashed line on the plot on the right indicates the average hop height during splash dominated saltation.

The conceptual model of snow saltation illustrated in Fig. 10 neglects the effect of large turbulent eddies that exist over snow-covered regions subjected to high friction velocities. Indeed, these large eddies will enhance the entrainment of particles in suspension and, perhaps, limit or suppress the motion of particles in saltation. This limiting case goes beyond the scope of this study and needs further investigation. It cannot be evaluated with the computational domain considered in this analysis nor with wind tunnel experiments. In addition, it represents a challenge for the instruments currently available to assess snow saltation dynamics.

## 4    Summary and outlook

The importance of drifting snow events to the mass and energy balances of snow-covered regions has long been acknowledged by the scientific community. Several atmospheric and snow models are currently enriched with drifting snow schemes that represent the aeolian transport of snow and quantify the induced changes in snow height and the amount of snow sublimation. Nevertheless, the correct prediction of these quantities is still a challenge. This is due to the complexity of the phenomenon, the uncertainties in the parameters that control drifting snow transport and sublimation, but also due to inaccuracies in the drifting snow schemes themselves.

In this work, we focus on an important part of drifting snow schemes that is typically overlooked: the saltation model. By comparing the saltation models used in the drifting snow schemes of RACMO, MAR, Meso-NH and CRYOWRF (Amory et al., 2021; Vionnet et al., 2014; van Wessem et al., 2018; Sharma et al., 2023) with the different theoretical, experimental and numerical studies on sand and snow saltation, we conclude that the parameterizations employed are not fully aligned with the current understanding of snow saltation dynamics. In addition, by performing numerical simulations with a LES-based model, we verify the consistency of some parameterizations and show the relationship between them and the prevalence of aerodynamically entrained or splashed particles.

In the saltation model of RACMO and MAR, the particle mass flux in the saltation layer is assumed uniform in height (Pomeroy and Gray, 1990). This contrasts with the well documented exponential decay found in wind tunnel and field observations of sand and snow saltation, as well as numerical simulations. In addition, the calculation of the transport rate and of the particle concentration at the top of the saltation layer is based on the assumption that the saltation layer height is proportional to $u_*^2$ and that the average particle streamwise velocity in the saltation layer is invariant with respect to $u_*$. Even though both scaling laws can be obtained from theoretical, numerical or experimental studies, depending on the friction velocity and on the snow surface characteristics, these two assumptions are contradictory. While the first is based on the idea that the near-surface particle velocity scales with $u_*$, the latter is based on the notion that the near-surface particle velocity is invariant with respect to $u_*$. As a result, the transport rate is found to scale linearly with $u_*$, which contrasts with most saltation models and measurements available in the literature (e.g., Doorschot and Lehning, 2002; Sørensen, 2004; Melo et al., 2022). Differently, the saltation models used in Meso-NH and CRYOWRF assume that the particle mass flux profile follows an exponential decay. However, when computing the decay height and the particle concentration, which are a function of the height of the saltation

layer and of the average particle streamwise velocity, they consider the same contradictory parameterizations regarding the near-surface particle velocity.

The available experimental and numerical studies show that the near-surface particle velocity is invariant with respect to $u_*$ when saltation develops over a sand covered surface. In this case, the average hop height of saltating particles, the height of the saltation layer and the decay height of the exponential mass flux and concentration profiles are found to be equally invariant with respect to $u_*$. However, when saltation develops over snow-covered surfaces, the scaling of these different quantities with $u_*$ is less clear. While early wind tunnel studies reveal that the decay height of the mass flux profile scales with $u_*^2$ (and, therefore, that the near-surface particle velocity scales with $u_*$) (Sugiura et al., 1998), recent experiments do not reveal a clear scaling of the decay height with the friction velocity and show that the scaling of the near-surface particle velocity with $u_*$ depends on the snow surface characteristics (Aksamit and Pomeroy, 2016; Gordon et al., 2009).

From numerical simulations, we show that the scaling of the near-surface particle velocity, average hop height and decay height are a function of the saltation regime: when saltation is dominated by aerodynamic entrainment, the near-surface particle velocity and the average hop height increase with the friction velocity; when saltation is dominated by splash, these two quantities are mainly invariant with respect to $u_*$. For the full range of friction velocities studied, the decay height is found to increase logarithmically with $u_*$. Therefore, the rate of increase of the decay height with $u_*$ decreases as $u_*$ rises. Thus, when saltation is dominated by splash, the decay height increases at a much lower rate than what is obtained when saltation is dominated by aerodynamic entrainment. When saltation is dominated by splash, the decay height significantly deviates from the evolution proposed by Nishimura and Hunt (2000), which is considered in the drifting snow schemes of Meso-NH and CRYOWRF. The effect of the saltation regime on the saltation dynamics and its scaling laws might partly justify the different results obtained by Sugiura et al. (1998) in a wind tunnel in comparison to the results obtained by Gordon et al. (2009) and Aksamit and Pomeroy (2016) in the field. Indeed, the limited size of a wind tunnel might restrict the development of a fully developed saltation system dominated by splash. This increases the relative importance of aerodynamic entrainment and might justify the observed scaling of the decay height with $u_*^2$ obtained by Sugiura et al. (1998). Other aspects, such as the height at which the particle concentration and mass flux are measured and the snow surface characteristics, are also expected to significantly influence the decay height and its scaling with $u_*$. Further investigation is needed to fully understand the impact of these different aspects on the decay height.

In general, the saltation models used in RACMO, MAR, Meso-NH and CRYOWRF have a poor representation of the effect of the snow surface characteristics on the different quantities of interest. The snow surface characteristics have a direct effect on one single parameter - the fluid threshold - which is used as an approximation for the surface shear stress. Therefore, they influence the transport rate and the average particle streamwise velocity. However, even though both the fluid threshold and the surface shear stress are found to increase as the grain diameter increases, the opposite is found for the particle streamwise velocity from previous works. In addition, the numerical simulations suggest that the surface shear stress might not be well approximated by the fluid threshold and that it increases with the rise of the friction velocity when saltation is dominated by aerodynamic entrainment. Therefore, improvements are needed to correctly take into account the effect of the snow surface characteristics on the snow saltation parameterizations. For instance, the coefficients in the expression of Sørensen (2004) used

in Meso-NH and CRYOWRF to compute the transport rate are assumed constant. However, they are expected to vary with the snow surface characteristics. Extensive measurement campaigns are needed in order to fully understand and quantify the effect of the snow surface on the saltation dynamics. In addition, it is essential that future experimental studies describe the characteristics of both the snow surface, the wind field and the particles in saltation.

In this work, we have mainly analyzed the parameterizations used to compute the transport rate and the vertical profiles of particle mass flux, concentration and streamwise velocity. However, the description of snow saltation in a drifting snow scheme is also dependent on the parameterizations used for the fluid threshold, the particle size distribution and the implementation of the lower boundary condition for the snow suspension scheme. Therefore, a complete assessment of a saltation model must take these parameterizations into account. In addition, the height at which the lower boundary condition is specified must be in agreement with the assumed parameterizations. The actual value used in drifting snow schemes (which is typically given by the height of the saltation layer) is rather irrelevant. However, one must guarantee that all parameterizations that constitute the saltation model are valid at that height. For instance, when saltation is dominated by splash, the presented simulation results show that the average particle streamwise velocity is only invariant with respect to $u_*$ in the first 1-2 cm above the surface. Even though saltating particles can reach higher elevations, the use of the previous assumption supports the application of the lower boundary condition close to the surface. Indeed, all studied drifting snow schemes set the lower boundary condition at a height relatively close to the surface: typically, its value increases with $u_*$, reaching a value of 5-6 cm for $u_* = 0.8 \text{ m s}^{-1}$ (Pomeroy and Gray, 1990; Pomeroy and Male, 1992). However, the particle size distribution at the lower boundary is generally assumed independent of the snow surface characteristics (Vionnet et al., 2014; Sharma et al., 2023), which is not a reasonable approximation close to the surface, but only at higher elevations, close to the saltation-suspension interface (8-10 cm). The lower boundary condition of snow suspension schemes can be specified at this location. However, the parameterization for the particle streamwise velocity would have to be adjusted, in particular its scaling with $u_*$. Even though this would add additional unknown parameters to the model, the use of an advanced parameterization for the particle streamwise velocity would allow the description of saltation systems dominated by aerodynamic entrainment. Nevertheless, when setting the lower boundary at a higher elevation, one must bear in mind that the particle mass flux profile is expected to deviate from an exponential decay at the saltation-suspension interface (e.g., Nishimura and Nemoto, 2005). Taking into account the advantages and disadvantages, this choice is left to the modeler.

The snow saltation parameterizations studied in this work are based on the steady-state saltation assumption. Even though this approximation is widely used, it does not fully represent the phenomenon of snow saltation in natural environments. Non-stationary wind conditions (Aksamit and Pomeroy, 2018) and intermittent saltation (Paterna et al., 2016; Comola et al., 2019c) have been recently studied, but further investigation is needed to incorporate these findings in simple parameterizations.

The correct prediction of snow transport and sublimation at large scales has a variety of challenges. One of them is the representation of snow saltation by means of simple parameterizations. Snow saltation is a complex phenomenon, challenging to measure and model at both small and large scales. The full understanding of the processes involved is a multi-disciplinary endeavour, which requires contributions from both atmospheric and snow scientists. By means of a thorough and careful

analysis of previous developments, as well as new simulations, this work offers a new insight into the saltation system and

presents some pathways to improve the representation of snow saltation in atmospheric models.

*Code and data availability.*  The code version used in the simulations, the results presented and the post-processing scripts used to analyze the simulation outputs and produce the figures are available at the EnviDat repository (https://www.doi.org/10.16904/envidat.479). The code is also available at the institutional GitLab repository (https://gitlabext.wsl.ch/atmospheric-models/les-lsm - see tag 'r2024').

*Author contributions.*  DBM performed the simulations, implemented the post-processing scripts and made the literature review. DBM, AS

and ML contributed to the analysis of the results. AS and DBM contributed to the code development and its maintenance. ML supervised the work and secured its funding. DBM prepared the manuscript with contributions from all co-authors.

*Competing interests.*  The authors declare that they have no conflict of interest.

*Acknowledgements.*  The authors acknowledge the Swiss National Science Foundation (SNSF) for the financial support (project number 179130) and the Swiss National Supercomputing Centre (CSCS) for providing the computational resources (projects s1031 and s1115). Pedro

Cabral is acknowledged for the numerous in-depth discussions about physics and fluid dynamics, which contributed to the presented literature review and subsequent analysis, for preparing Figs. 1 and 10 and for the detailed revision of the text. Varun Sharma is acknowledged for his support in the early stages of this work and for several discussions that have inspired part of the work developed. Raleigh L. Martin, Kouichi Nishimura, Nikolas O. Aksamit, Masaki Nemoto and Konosuke Sugiura are acknowledged for promptly providing extra clarifications about their works. Finally, the authors thank the two reviewers for their constructive comments and suggestions, which have improved the clarity

and completeness of this article.

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
