# Peer review of "Understanding snow saltation parameterizations: lessons from theory, experiments and numerical simulations"

_EGUsphere, 2023_

## Author Comment (AC1)

**Reply to Reviewer #1**

The authors response to the comments of the first reviewer is presented in this document. We would like to thank the reviewer for the constructive comments and suggestions. We believe that the review process will make the manuscript more clear and complete. We apologize for the delay in providing an answer – this was related to the completion of the doctoral studies of the first author. At the end of this document, you can find the Track-changes file where we highlight the modifications made to the manuscript up to now. The line numbering in our answers refers to the Track-changes version of the manuscript. Comments colored in **blue** are related to the Introduction. They require a deeper review of the literature which will be completed by the authors when submitting the revised manuscript.

**General comment:**

*[…] As it stands, the manuscript needs to be improved prior to consideration for publication. The introduction material is currently a bit underdeveloped. The authors are presenting an advancement of a topic that has been studied in a large number of scenarios, which a wide range of applications as is shown in Section 2 when the actual parameterizations are being discussed. As well, the title suggests a kind of review of the state of the field. The authors do not present a sufficiently wide view of the topic in the introduction. I strongly suggest a deeper review of the available studies. Snow transport and snow saltation has been studied and modeled since at least the 1950's, but the case studies selected to summarize the field and the complexities of the topic are a bit sparse and lacking. For example, the authors have a tendency to cite Antarctic studies in the intro, but the benchmark snow flux profile study of Budd et al. (1966) in Antarctica is not referenced. Perhaps the title would be better suited as "Understanding snow saltation in atmospheric modeling parameterizations".*

Thank you for this comment. The main goal of the introduction was to motivate the need to better assess snow saltation parameterizations and, therefore, the analysis performed in this manuscript. However, we agree that the introduction can be expanded to provide to the reader a wider view of the topic. In this regard, the study of Budd et al. (1966) will be analyzed and included in the manuscript, as well as other references addressing the study of snow transport in the Arctic and in mountainous terrain. Nevertheless, we would like to clarify that this work does not intend to be a review of the state of the field as a whole, but to review snow saltation parameterizations and to further investigate them with a numerical model. Regarding the title, we would prefer to keep it as it

is. In our view, it correctly captures the content of the manuscript. Even though we focus on parameterizations used in atmospheric models, we believe that this work is of interested to everyone requiring a simple description of snow saltation.

**Specific comments:**

*L20-22: Sastrugi?*

A mention to sastrugi was added to the text (l.21).

*L28-29: Do you have a citation for this convention?*

We now cite the works of Li and Pomeroy (1997) and Lenaerts et al. (2012), who mention this convention in their articles (l.28-32). In the websites of the World Meteorological Organization (WMO)[1] and of the American Meteorological Society (AMS)[2], the same definitions can be found. The definition given by the WMO was previously cited in the work of Gauer (2001).

[1]WMO  World Meteorological Organization. (2017). Hydrometeors consisting of an ensemble of particles raised by the wind: Drifting and Blowing Snow. Retrieved October 16, 2023, from https://cloudatlas.wmo.int/en/drifting-and-blowing-snow.html

[2]AMS American Meteorological Society. (2012). Glossary of Meteorology: Drifting Snow. Retrieved October 16, 2023, from https://glossary.ametsoc.org/wiki/Drifting_snow

*L28-39: This paragraph started off with descriptions of blowing snow, and then focuses solely on Antarctica. Given the amount of time you spend on non-Antarctic snow studies in the remainder of the manuscript, it's a bit strange to limit to such a specific niche here. Please include a broader review.*

We will include some statements presenting the importance of drifting snow transport and sublimation to the surface mass balance and snowpack stability of non-Antarctic snow covered sites.

*L37-39: I am certain the authors can find an older reference to this complication than this.*

We will replace the citation or add older ones to this statement. Thank you for pointing it out.

*L40-46: Again, this focuses solely on drifting snow in Antarctica. This is a bit strange in a general discussion of drifting snow. Given the number of studies on snow sublimation from field-based, hydrological, and numerical studies, it also seems odd that the authors are using a regional weather model and satellite observations.*

The goal of this paragraph was to showcase the disagreement between large-scale models and measurements. We will analyze the literature so that we can extend the paragraph with additional examples that point in the same or opposite direction. If needed, the statement presented in l.47-49 will be made lighter.

*L53-54: I do not believe it is widely accepted that snow particles that travel in suspension only arise from particles that have undergone fragmentation. There are no whole snow particles in suspension?*

We agree with the reviewer. We were presenting a narrow definition of particles in suspension. The reference to particle fragmentation was therefore deleted from the text (l.56-57).

*L56: This model of snow redistribution far precedes the referenced literature including, but not limited to (Dyunin, 1967; Lee, 1975; Dyunin, 1980; Pomeroy et al., 1993; Gauer, 1998; Bintanja, 2000)*

We agree with the reviewer. In the manuscript, we have cited atmospheric and snow models that use the mentioned approach instead of the works that introduced this methodology. We will modify the citation list.

*L62: It would be helpful to cite the sublimation models that use the Thorpe and Mason approximations.*

The models that use the approximations of Thorpe and Mason (1966) are now cited in the text. In addition, a reference to the model of Lin et al. (1983), used by Amory et al. (2021), was added (l.65-67).

*L66: Again, the references to modeling snow redistribution are a bit silo'd as they tend towards recent research from adjacent groups. These issues have been mentioned or studied for many decades, even in mountainous terrain (e.g. Dyunin et al., 1977; Schmidt, 1982; Gauer, 1998).*

Thank you for this comment. We will modify the citation list so that it better represents the variety of studies on the topic.

*L65-76: There is again a strange tendency towards Antarctic research, which is not where the majority of drifting snow research has actually been conducted. Snow redistribution was coupled in a GCM by Eric Brun back in the 1980's.*

In this paragraph, we intend to showcase the sensitivity of drifting snow models to the parameters chosen in the snow saltation model. We believe that the sensitivity studies performed with MAR and RACMO are good examples. Nevertheless, following the reviewer's comment, we will revise the literature and look for other sensitivity studies.

*L101: Spanwise?*

Thank you for the suggestion. The term "spanwise" can indeed be used as a synonym for "crosswise". However, in our view, the term "crosswise" relates well with the term "cross section area", so we would prefer to keep it.

*L103-105: What do you mean statistically invariant? The temporal-average of windspeed is constant on each wall-parallel plane? And can you define "mainly" aligned?*

Yes, with "statistically invariant along the horizontal directions" we want to convey that the temporal-averages of wind velocity do not vary in x and y, therefore, they are constant on each wall-parallel plane. We have reformulated the sentence to make it more clear (l.120-122). With "mainly aligned with the streamwise direction", we intended to inform the reader that the x-axis is aligned with the time-average horizontal flow direction. However, taking into account that this is a natural consequence of imposing a streamwise pressure gradient, this information was deleted from the text.

*L106-110: Strictly speaking, between the viscous sublayer at the surface and the inertial sublayer is a buffer sublayer where you do not have a fluid velocity profile that provides a constant shear stress. Furthermore, the log-layer or inertial sublayer also only exists for a finite height in the*

*atmosphere. It would benefit the authors and the reader if the authors could address what assumptions we need to make for this sort of shear stress profile they are describing, and what is assumed to happen to the buffer layer in the presence of saltating particles. Typically some sort of roughness height is included in a log-law profile (which comes from the constant shear stress assumption) at which the wind essentially goes to zero above the actual snow surface, though this is widely known to not be physically true, but it helps close the equations. This roughness length is not mentioned anywhere, though often also modified to account for the momentum deficit caused by the saltation layers that the authors are referencing. This dates back to Bagnold.*

The reviewer is making a link between a logarithmic profile and a constant shear stress. However, in rigor, the link that exists is between a logarithmic profile and a constant turbulent shear stress. Over the whole inner layer (viscous, buffer, and logarithmic sublayers), the total shear stress (the sum of viscous and turbulent shear stresses) is expected to be constant (Schlichting and Gersten, 2017: eq. 17.1 in p.520). However, in the viscous sublayer, the viscous shear stress is dominant and the resultant velocity profile is linear, while in the logarithmic sublayer the turbulent shear stress is dominant and the velocity profile is logarithmic. This was made clear in the text (l.124-146). In this new paragraph, we have also stated the region of validity of the logarithmic profile in the atmospheric boundary layer, the effect of roughness on the boundary layer structure, and the use of an equivalent roughness length to describe the velocity profile above the saltation layer. Taking into account that the size of this section increased substantially, we have decided to create a new subsection (2.1 The steady state assumption), which is preceded from an introductory paragraph. In this way, l.163-170 were deleted from the text.

*L171-172: Did Bagnold show that the necessary shear stress was smaller, or that a smaller friction velocity was sufficient, given that \tau=\rho u_*^2 is only an "effective" shear stress at the surface.*

We have clarified the definition of $\tau = \rho.u_*^2$, which is now defined as the "shear stress in the logarithmic sublayer" (l.135-136). Thus, $\tau$ can be interpreted as the value of shear stress measured with the eddy covariance method or from fitting the wind velocity profile to a logarithmic function. Therefore, we would prefer not to name $\tau$ the "effective shear stress at the surface", but to clearly distinguish $\tau$ (the shear stress in the logarithmic sublayer) from $\tau_s$ (the shear stress at the surface). These two quantities assume equal values only in the absence of saltating particles or at the thresholds for saltation onset and cessation. In this context, the difference between "the necessary shear stress was smaller" and "a smaller friction velocity was sufficient" is not clear to us. Bagnold (1941) deduced the friction velocity from wind velocity measurements performed in the logarithmic

sublayer. In this way, Bagnold's conclusions refer to the friction velocity $u_*$ and not $u_{*,s}$ (as defined in the manuscript). However, as mentioned above, at the limiting threshold shear stresses (fluid and impact thresholds) the surface shear stress and the shear stress outside the saltation layer are expected to assume similar figures due to the low number of particles aloft and the respective small momentum exchange between the fluid and the particles. This is now mentioned in the text (l.211-215).

*L173: Requires, not implies?*

This was modified in the text (l.216).

*L192-193: What is the fluid threshold friction velocity?*

In the manuscript, the fluid threshold was defined in the Introduction (l.74-75). However, as this is not ideal, the fluid threshold is now presented at the beginning of section 2.2.1 (l.196-198). The Introduction was modified accordingly (l.74-82).

*L193-194: Please rewrite this sentence. It is equal because it is assumed to be a fraction, and we assume this fraction is 1?*

The assumption that $\tau_s$ is equal to $\tau_{ft}$ (therefore, that $\tau_s$ is a fraction of $\tau_{ft}$ and that this fraction is assumed to be 1, as pointed out by the reviewer) is not in agreement with Owen's claim. Owen (1964) specifically claims that $\tau_s$ must be smaller than $\tau_{ft}$ because the particle bombardment at the surface contributes to dislodging the grains. The text was modified to make this claim more clear (l.208-210, l.236-238).

*L196: Nor is it in agreement with experimental evidence dating back to Bagnold.*

A reference to the wind tunnel experiments of Bagnold (1941) was added to the text (l.240-241).

*L196: Now i'm confused. I thought \tau_s was the approximated surface shear stress, which varies with windspeed. I believe you are actually referencing \tau_it, but using its equality with \tau_s in this one instance. Can you replace \tau_s with \tau_it to make the following inequalities clearer?*

In the text, $\tau_s$ is defined as the actual (not approximate) surface shear stress during saltation (see l.137-138 and l.145-146). In addition, we follow Bagnold (1941) and define $\tau_{it}$ as the lower shear stress outside the saltation layer for which saltating particles maintain a steady state motion (l.207-

208). Therefore, the impact threshold is a well defined value that characterizes the bed and it is invariant with respect to $u_*$. Differently, $\tau_s$ may vary with $u_*$. As discussed in section 2.2.1, Owen (1964) hypothesizes that $\tau_s = \tau_{it}$, atmospheric models frequently assume that $\tau_s = \tau_{ft}$ (e.g., Lenaerts et al., 2012; Amory et al., 2021; Vionnet et al., 2014), but some numerical models revealed that $\tau_s$ is not invariant with respect to $u_*$. We hope that Table 1 (page 9), suggested by the second reviewer, helps clarifying the different shear stresses.

*Does Figure 2 include the decay height of all experiments mentioned in Section 2.2.3?*

No, Figure 2 includes only the experimental measurements of Sugiura et al. (1998), Namikas (2003), Nishimura and Nemoto (2005), and Martin and Kok (2017), as stated in l.515-517. The obtained values of decay height are discussed in the paragraphs that follow. The results of Sato et al. (2001), which are also mentioned, are not presented in Figure 2 because the respective value of $u_*$ was not provided by the authors. This is currently specified in the text (l.534-535). In l.555-556, we also refer to the results of Nalpanis et al. (1993), but we do not present them in the figure. As they consist of only 2 data points, we have decided not to include them in the comparison.

*L552: What is the subgrid-scale relative to the particle size? At what scale can we expect the snow particle to stop responding to fluctuations, and why?*

As mentioned in section 3.2, the domain is a cube of 6.4 m length, uniformly discretized in each horizontal direction in 64 cells (i.e, $\Delta x = \Delta y = 0.1$ m). The mesh in the vertical direction follows a hyperbolic stretching: it varies from $\Delta z = 0.01$ m along the first 15 cm above the surface to $\Delta z = 0.1$ m close to the top boundary. Therefore, $\Delta = (\Delta x . \Delta y . \Delta z)^{1/3}$ varies between 0.05 and 0.1 m. Taking into account that the mean diameter is set to $360.10^{-6}$ m, the ratio between $\Delta$ and the mean diameter varies between approximately 130 and 280.

The ability of the snow particles to follow the velocity fluctuations can be evaluated by computing the Stokes number, St, which is a function of the particle relaxation time and the velocity and length scales of interest. If St<<1, the particles will mainly follow the flow. The opposite is true if St>>1. Therefore, the maximum frequency of the wind velocity signal that is able to influence a particle trajectory depends on the particle diameter and Reynolds number. This analysis was not performed in the context of this work.

*L559-561: This is not a comprehensive list of all forces acting on a particle in a turbulent flow. Beyond gravity (buoyancy) and drag, there are also force terms from the wind, the force from the fluid moving with the particle, and the Basset-Boussinesq memory (Maxey and Riley, 1983; Talaei and Garrett, 2020).*

We agree with the reviewer. Indeed, we are not presenting a complete list of forces applied on the particles. Our intention was to provide some examples. We have rephrased the text to make this more clear (l.608-609).

*L566: "entrainment"*

Corrected.

*L570: Is it assumed there is an infinite number of particles that are available to be entrained?*

A sufficiently large initial deposition is specified in the simulations in order to guarantee that snow transport is never limited by a shortage in the supply of erodible particles. This is now mentioned in the text (l.624-625).

*L576: Modulus*

Corrected.

*Figure 5: Can you discuss these decreases in the inset plots more? This seems like an interesting point that is glossed over rather quickly. As well, it would be helpful to include two more plots that compare the average particle velocities with the average wind speeds in the two directions at each height, for the various scenarios. The influence of vertical winds on ascending saltating particle is particularly interesting to think about as this is a primary factor driving the transition to suspension and away from the paradigms described here. Perhaps a vertical profile of vertical drag (normalized by gravity forces) could reveal these relationships?*

Thank you for this comment. We have added the average vertical profiles of streamwise wind velocity to the manuscript (Figure 7). This plot shows that the model correctly predicts the existence of a focus point when saltation is dominated by splash (please note that the focus point was previously introduced in section 2.3.2, l.464-467). The analysis of the wind velocity profiles allowed us to better understand the observed decrease in near-surface particle velocity (l.745-763). In this context, the comment regarding the particles deceleration as they approach the surface was

deleted (l.763-766). In fact, above 5 mm height above the surface (first grid point), the average wind velocity is always greater than the average particle streamwise velocity. This can be seen in more detail in the figures below, where we present separately the average streamwise velocity of the upward and downward moving particles (not shown in the manuscript). If we assume a logarithmic profile below the first grid point characterized by $u_{*,s}$ and $z_o=10^{-4}$ m, the average streamwise wind velocity will be smaller than the average streamwise velocity of the downward moving particles below approximately 2 times the average grain diameter. However, in the model, the surface processes of rebound and splash occur as soon as the particles reach a height lower than 4 times the average particle diameter (1.4 mm in the current setup). Therefore, in the simulations, the drag force in the streamwise direction is on average always positive and the decrease of the near-surface particle velocity as $u_*$ rises above 0.39 m s$^{-1}$ is mainly related to the respective decrease in streamwise wind velocity below the focus point (~ 1 cm height).

[Figure]

Figure I: Vertical profiles of average wind and particle streamwise velocity. The thick lines correspond to the wind velocity, the thin lines with upward pointing triangles correspond to the upward moving particles, and the dashed lines with downward pointing triangles correspond to the downward moving particles.

The average wind velocity in the vertical direction is presented on the figure below (not shown in the manuscript). As expected from a half-channel flow, it is mainly zero. It reaches higher values close to the surface, but it is nevertheless several orders of magnitude smaller than the average particle vertical velocity (Figure 5b). Therefore, over a flat surface subjected to a streamwise pressure gradient, the average vertical wind velocity has a negligible effect on driving the transition from saltation to suspension. Small particles reach higher elevations mainly due to the effect of velocity fluctuations. High instantaneous velocities may induce an instantaneous vertical drag force

greater than the particle weight, which transports the particles upward. Overall, while the gravitational force pushes the particles downward, turbulence promotes particle dispersion and a homogeneous concentration of particles aloft.

[Figure]

Figure II: Vertical profile of average wind velocity in the vertical direction obtained for different friction velocities. In the inset, a zoom-in to the near-surface region is presented.

The vertical profiles of aerodynamic drag in the streamwise and vertical directions are presented below for completeness (not shown in the manuscript). As expected, the vertical component of aerodynamic drag is much smaller than its streamwise component.

[Figure]

Figure III: Vertical profiles of aerodynamic drag in the streamwise and vertical directions normalized by the gravitational force. The different colors correspond to the different friction velocities. The upward pointing triangles correspond to the upward moving particles and the downward pointing triangles correspond to the downward moving ones.

*L606-607: How many actual lagrangian particles were in transport at a given time? How does this assumption of clustering work? When a splash event occurs at a friction velocity of 0.3, you automatically assume 50 snow particles are ejected at that single location? And when you have an impact of a Lagrangian parcel, are you calculating the force of 50 times the mass equally distributed among the splashed particles? Admittedly this would create more pseudo-particles in transport while not requiring any additional Lagrangian computations, but the fact that each parcel at a given friction velocity has the same number of particles is worrisome. I see it would be difficult to truly track all the particles in a 6 m cube, but you are also manipulating the mass flux by enforcing these rules. I think there may be negligible effect on the relative concentration profiles, but some further explanation is needed here. If you instead chose to have 1000 particles in a lagrangian parcel, would this have a noticeable impact on your snow-wind feedback?*

The total number of parcels aloft (lagrangian particles) averaged along the last 100 s of each simulation are presented in Figure IVa (not shown in the manuscript). Depending on the friction velocity, we simulate on average between $10^4$ and $3.5 \times 10^4$ parcels. The total number of particles aloft (number of parcels x number of particles per parcel (PPP)) is presented in Figure IVb (not shown in the manuscript). Particles of the same parcel have the same diameter and follow the same trajectory. Therefore, they were aerodynamically entrained or splashed at the same surface location and time step (this is stated in the text in l.660-662).

[Figure]

Figure IV: Average number of parcels aloft obtained for different friction velocities (a) and the respective number of particles aloft (b). The error bars correspond to two times the standard deviation.

For $u_*=0.3$ m s$^{-1}$, for which PPP=100 (see corrected PPP values in l.663), the impact of one parcel corresponds to the impact of 100 particles. Taking into account the mass and impact kinetic energy

of 1 particle, we compute how many particles are splashed. Let's imagine that the model yields the value of 3. In this case, each particle ejects 3 particles, which means that the 100 particles will eject a total of 300 particles. Therefore, 3 new parcels (with 100 particles each) are ejected into the flow. For each ejected parcel, the diameter and initial ejection velocity of all particles in that parcel are a random realization of a normal and exponential distributions, respectively.

Indeed, the clustering of particles in groups reduces the variability of trajectories and grain size aloft represented by the model. Nevertheless, this simplification is expected to yield acceptable figures if the total number of parcels aloft is large enough so that the variability of trajectories and grain size is sufficiently represented. In the figure below, we present the effect of the number of PPP on the surface friction velocity averaged over the last 100 s of the simulations. It can be seen that the number of PPP has a more significant impact on the results for $u_* > 0.3$ m s$^{-1}$. For $u_*$=0.39 m s$^{-1}$, similar results were obtained when considering PPP = 200 in comparison with the simulations for which the number of PPP was allowed to vary between 500 and 1000. Therefore, PPP = 200 was considered acceptable for the analysis of average quantities undertaken. For the higher friction velocities ($u_* > 0.4$ m s$^{-1}$), additional simulations with a PPP smaller than 200 are still needed to confirm the convergence of the simulation results. However, this verification was not performed due to the computational cost of the simulations.

In our view, a sensitivity analysis to the different model parameters is outside the scope of this article. Therefore, this discussion was not added to the manuscript.

[Figure]

Figure V: Average surface friction velocity obtained for different friction velocities and different number of PPP. In the legend, the number of PPP is presented in increasing order of $u_*$. The last value of the array is representative of the remaining friction velocities.

*L619: How is steady state quantified? Any subset of the 100 seconds would provide the same mean profiles? Can you please provide some verification of this?*

The time series of the mass aloft per unit surface area is presented on the figure below for the various friction velocities studied (not shown in the manuscript). In the simulations, saltation is allowed to take place after t=25 s. Following an initial rise in mass aloft, the mass of particles decreases with time until it reaches a stable value. From observing the time series, we have assumed that this equilibrium was reached after 250 s for all simulations. We have adapted l.673-674 to better illustrate our approach.

[Figure]

Figure VI: Time series of the mass of particles aloft per unit surface area obtained with different friction velocities. The inset is a zoom-in to the three lowest values of $u_*$.

In Figure VII (not shown in the manuscript), we present the average mass aloft computed with an increasing number of points (the value at t=250 s is the instantaneous mass aloft at t=250 s and the value at t=350 s is the average considering the previous 100 s of the simulations). It can be seen that 100 s is sufficient to arrive to a relatively stable average. For the higher friction velocities ($u_* > 0.5$ m s$^{-1}$), improved statistics would probably be obtained by extending the computational time (for instance, with additional 100 s). This was not performed due to the computational cost of the simulations.

[Figure]

Figure VII: Average mass of particles aloft per unit surface area for an increasing number of data points. Results obtained for all friction velocities studied.

*L623-624: Most of the numerical mass flux profiles.*

Even though the fit is not shown, this is also true for the presented experimental profiles. Therefore, no modification was made in the text.

*L657: Does the average vertical ejection velocity combine both the splashed particles and the rebounded particles? Please clarify here.*

The average vertical ejection velocity is computed from all particles aloft (aerodynamically entrained, splashed or rebounded particles). Throughout the text, we use the term "ejection velocity" as a synonym for the initial particle velocity without making any distinction regarding its origin (see sections 2.2, 2.2.2, and 3.1). Therefore, we believe that it is not necessary to clarify this in the text.

*Figure 5: I think v_z^{up} would be a better notation as you are elsewhere using superscripts to denote subsets of velocities in the direction specified in the subscript*

Thank you for the suggestion. This was modified in the figure.

*L664: Particle velocity profile is mainly invariant*

This was modified in the text.

*L678-679: Please clarify if this is something you are enforcing in the model, or a result of your model.*

In the model, the ejection velocity of aerodynamically entrained particles is a random realization from a lognormal distribution. However, the mean value of the distribution is proportional to $u_{*,s}$. This was made clear in the text when describing the numerical model (l.621-623). Therefore, when aerodynamic entrainment is the main entrainment mechanism, which is the case for $u_*$ varying between 0.15 and 0.3 m s$^{-1}$, the average ejection velocity will be necessarily proportional to $u_{*,s}$. This was made clearer in the text by replacing "is justified by" with "is a direct consequence of" (l.737). Nonetheless, the fact that aerodynamic entrainment is the main entrainment mechanism for $u_*$ varying between 0.15 and 0.3 m s$^{-1}$ is a result of the model.

*Figure 6: As stated above I think it's weird to be using mass flux instead of a relative number flux given that the mass flux seems to be directly modified by this lagrangian parcel idea.*

When computing the mass flux profile, the mass of each parcel aloft is computed by multiplying the mass of one individual particle times the number of PPP. Similarly, to compute the number flux profile, the number of particles in each parcel aloft would be equal to the number of PPP. In both calculations, each of these particles is assumed to travel at the same velocity: the velocity of the parcel. Therefore, we are expecting the mass flux and the number flux to be equally sensitive to the number of PPP. Therefore, we disagree with the reviewer's comment and no modifications were made in the text.

*L682-683: I thought the relationship between impacting particle and splash velocity was stochastic?*

In the model, the ejection velocity of splashed particles is a random realization from an exponential distribution. The mean value of the distribution is a function of the particle impact velocity. This was made clear in the text when describing the numerical model (l.634-635). In addition, the word "average" was added to the text (l.727).

*L685-686: It would be good to show wind profiles for your full range of friction velocities from the surface to above the saltation layer and that show and quantify this invariance as that is actually a non-trivial conclusion. What do relative velocity changes look like at each height?*

As suggested, the streamwise wind velocity profiles were added to the manuscript (see reply to comment on Figure 5 in page 8). The streamwise wind velocity is approximately invariant with respect to $u_*$ in the near-surface region when saltation is dominated by splash. This is due to the existence of a focus point which was previously observed by Bagnold (1941) in wind tunnel experiments. From Figure I in this document, one can assess the relative velocity in the first 5 cm above the surface. As expected, the upgoing particles experience a greater relative velocity than the downgoing ones.

*L686-687: This is an interesting point as well. When do particles in saltation have a change in sign for the drag? Assuming a non-slip velocity at the immediate snow surface, it should always occur somewhere. Is this accounted for in your snow-wind feedback? As we approach steady-state, and snow particles are already quickly moving, at what fraction of maximum trajectory height does that of transition drag sign-change occur? How does this momentum source for the wind balance with the momentum sink caused by the presence of particles? In turn, what role does this play in modifying $u^*$ or $u^*,s$?*

As mentioned in the reply to comment on Figure 5 in page 8, for the simulated conditions, the change in sign for the drag may only occur below a height of approximately 2 times the average particle diameter. However, on average, this does not occur in the simulations because the surface processes of rebound and splash are computed as soon as the particles are at a height lower than 4 times the average particle diameter. In this way, the statement to whom this comment is concerned was deleted from the text. As mentioned in l.610-612, the feedback of the particles on the flow momentum is taken into account by the addition of a reaction force in the filtered Navier-Stokes equations, which is equal in magnitude and opposite to the aerodynamic drag applied to the particles. Therefore, if a particle is eventually faster than the surrounding wind velocity, it will experience a negative drag force, which is taken into account in the momentum equations as a positive source term. Nevertheless, these body forces on the momentum equations are generally negative (sink terms) and lead to an overall reduction of the wind velocity. As discussed in the reply to comment on Figure 5 (page 8), when considering a constant roughness length, a smaller near surface wind velocity implies a lower surface friction velocity $u_{*,s}$. This was made clear in the manuscript when discussing Figure 7 (l.755-758).

Regarding $u_*$, it is defined in the text as the friction velocity outside the saltation layer, $u_* = \sqrt{(\tau/\rho_a)}$, where $\tau$ is the shear stress outside the saltation layer (equal to the surface shear stress, $\tau_s$, only in the absence of saltating particles) and $\rho_a$ is the air density. As previously mentioned, this was made more clear in the text (l.135-136). In a half-channel flow without saltating particles, the fluid shear stress depends only on the imposed pressure gradient and on the height of the channel (l.604-605). Therefore, $u_*$ does not vary with the surface roughness. As described in Bagnold (1941, page 53-55), changes in the roughness length modify the magnitude of the wind velocity, but not the shear stress profile, and therefore, the value of $u_*$. During saltation, Owen (1964) hypothesized that "the saltation layer behaves, so far as the flow outside it is concerned, as an aerodynamic roughness whose height is proportional to the thickness of the layer". Therefore, outside the saltation layer, similar values of $u_*$ will be obtained with or without particles in saltation. The feedback of the particles on the flow decreases the streamwise wind velocity, but does not modify the shear stress outside the saltation layer.

*L732-733: What do you mean perturbed? How was this set of trajectories filtered? It sounds like you may be intentionally neglecting the impact of wall-normal winds and trying to only focus on ballistic-like?*

In the analysis presented in the manuscript, we have neglected saltating particles whose trajectory exhibited at least one local minimum far away from the surface (above 1 cm height). The goal was indeed to only consider trajectories that did not deviate considerably from classic ballistic ones because the hop height is expected to be a stronger function of the vertical ejection velocity in these cases. However, taking into account that we compare the average hop height with the average vertical ejection velocity considering all particles in saltation (and not only those that follow ballistic trajectories), we now recognize that this is not ideal. Therefore, we have modified the post-processing of the results and updated Figure 8b. In the current version, we have defined a particle hop as a fraction of a particle trajectory limited by two consecutive local minimums located close to the surface. The hop height is given by the maximum height attained by the particle between these two local minimums. This is currently explained in the text (l.810-811). Higher values of average hop height are obtained with the new post-processing, but similar conclusions are obtained. Slight changes were made in the text regarding the interpretation of the results (l.817-838).

*L735-736: The particles never reach the surface? Does the wind field extend all the way to the surface?*

As mentioned in the manuscript (l.814-816), in the model, the surface processes of rebound and splash are computed as soon as the particles reach a height lower than 4 times the average grain diameter (0.14 cm in this case). Therefore, the particles are assumed to rebound or deposit before reaching $z = 0$ m.

Regarding the wind field, the first grid point of the Eulerian mesh is placed at 5 mm above the surface. Below this height, the wind velocity in the horizontal direction is assumed to follow a logarithmic profile. This is now specified in the text when discussing the wind velocity profile (l.755-756).

*Figure 9: In the splash-dominated saltation regime, it appears the saltation height is constant with variations in u\*, and with constant mean particle velocity, correct? Then, researchers that are studying blowing snow in highly-unsteady wind should always measure the same snow particle velocities near the surface, as long as the windspeeds stay above the threshold and are averaged for 100 seconds? This seems like an unphysical conclusion to come to as there is no upper limit on wind speeds at the snow surface above the splash-dominated threshold, but you are imposing a clear upper limit on particle speeds. As well, would the height of the saltation layer not actually decrease with increasing windspeed as near-surface wind fluctuations (and fast particles) lead to quick suspension? This seems like a highly relevant question, especially for polar (and even high mountain regions) where 100km windspeeds are easily attained and shape the landscape.*

Yes, in the splash-dominated saltation regime, both the near-surface particle velocity and the saltation layer height are expected to be invariant with respect to $u_*$. This result is not only based on the current numerical simulations, but also on the wind tunnel measurements of particle velocity of Creyssels et al. (2009) and Ho et al. (2011) developed with sand for $u_*$ varying from 0.24 to 0.67 m s$^{-1}$ and from 0.42 to 1 m s$^{-1}$, respectively, the field measurements of Aksamit and Pomeroy (2016) of particle velocity over fresh snow ($u_*$=0.21-0.35 m s$^{-1}$) and old snow ($u_*$=0.24-0.54 m s$^{-1}$), and the numerical results of Ungar and Haff (1987), Kok and Renno (2009), and Durán et al. (2011). As long as the wind profiles form a focus point, there will be an upper limit for the wind speed close to the surface as $u_*$ increases. However, we agree that the presented dynamics might not represent high wind velocity conditions in the field, where the size of the turbulent eddies is larger than the size of those obtained in wind tunnels with a height of O(1) m and simulated in a half-channel with a height of O(1-10) m. These large eddies will enhance the entrainment of particles in suspension and,

perhaps, limit or suppress the motion of particles in saltation. The situation described goes beyond what the current model is able to simulate and what the existing wind tunnel and field measurements were able to assess. This is now acknowledged in the text (l.866-871).

**References not mentioned in the Track-changes:**

Dyunin, A. K. (1967), Fundamentals of the mechanics of snow storms, in Physics of Snow and Ice: proceedings, pp. 1065–1073.

Lee, L. W., and L. Wah (1975), Sublimation of Snow In Turbulent Atmosphere, PhD Dissertation, Department of Mechanical Engineering, University of Wyoming.

Dyunin, A. K., Anfilofiyev, B. A., Istrapilovich, M. G., Mamayeva, N. T. and Kvon, Y. D.: Strong Snow-Storms, their Effect on Snow Cover and Snow Accumulation, J. Glaciol., 19(81), 441–449, 1977.

Dyunin, A. K., and V. Kotlyakov (1980), Redistribution of snow in the mountains under the effect of heavy snow-storms, Cold Reg. Sci. Technol., 3, 287–294.

Schmidt, R. A.: Properties of blowing snow (1982), Rev. Geophys., 20(1), 39–44.

Pomeroy, J. W., D. M. Gray, and P. G. Landine (1993), The Prairie Blowing Snow Model :character istics, validation, operation, J. Hydrol., 144, 165–192.

Gauer, P. (1998), Blowing and drifting snow in Alpine terrain: numerical simulation and related field measurements, Ann. Glaciol., 26, 174–178.

Gauer, P. (2001), Numerical modeling of blowing and drifting snow in Alpine terrain, Journal of Glaciology, 47 (156), pp.97-110.

Maxey, M., and J. Riley, 1983: Equations of motion for a small rigid sphere in a nonuniform flow. Phys. Fluids, 26, 883–889, https:// doi.org/10.1063/1.864230.

Talaei, A. and Garrett, T. J.: On the Maxey-Riley equation of motion and its extension to high Reynolds numbers, , 1–19 [online] Available from: http://arxiv.org/abs/2006.16577, 2020.

Bintanja, R. (2000), Snowdrift suspension and atmospheric turbulence. Part I: Theoretical background and model description, Boundary-layer Meteorol., 95, 343–368.

[revised manuscript text omitted]

---

## Author Comment (AC2)

**Reply to Reviewer #2**

The authors response to the comments of the second reviewer is presented in this document. We would like to thank the reviewer for the constructive comments and suggestions. We believe that the review process will make the manuscript more clear and complete. We apologize for the delay in providing an answer – this was related to the completion of the doctoral studies of the first author. At the end of this document, you can find the Track-changes file where we highlight the modifications made to the manuscript up to now. The line numbering in our answers refers to the Track-changes version of the manuscript. Comments colored in **blue** require the study of some aspects of drifting snow dynamics that were not mentioned in the manuscript. This will be undertaken by the authors when submitting the revised manuscript.

**Specific comments:**

*1. The authors go through great lengths to distinguish the different types of shear stresses within the surface layer (e.g., fluid, surface, impact, etc). Given that this paper has a strong focus on the atmospheric aspects, I think it's worth discussing how u_star is determined through atmospheric models and how that value can relate to the different u_star values that comprise the different surface stresses. Further, I think it is also worth a brief discussion of drag partitioning (i.e., the loss of surface momentum to non-erodible elements) (e,g., Marsh et al., 2020, as an example of a parameterization with drag-partitioning). While maybe not critically important for much of the work performed in this paper, since much of this paper reviews the details of snow saltation at very small scales, it may be worth some brief discussion to tie in with other parameterzations used at regional atmospheric modeling scales.*

In general, the friction velocity in atmospheric models is computed by assuming a logarithmic profile (modified according to stability corrections). Therefore, for neutral stability conditions, $u_*$ is a function of the horizontal wind velocity at the first grid point above the surface and of the assumed roughness length, which varies according to the land-use. The available literature on this topic will be revised and an explanation with be added to the text, probably on section 2.1 (please note that the sections numbering changed – see Track-changes). In addition, the available parameterizations for shear stress partitioning will be analyzed and a comment will be added, probably on section 2.2 when explaining equation 1.

*2. As a sub-point, I think it may improve readability to have a table somewhere in section 2.1.1 listing the different surface stresses, though I will leave that choice to the authors. It can be hard to keep all of these straight, especially when many models use only one friction speed value.*

Thank you for this suggestion. We have added a table in section 2.2.1 (Table 1). As suggested we present the different shear stresses defined in the manuscript.

*3. The authors several times make reference to the fact that "snow surface" characteristics are critically important for accurately simulating the different variables that contribute the particle mass flux at the top of the saltation layer, however there is very little comment on the research that has been performed characterizing the threshold friction speeds for blowing snow algorithms, or any of the existing parameterizations in the literature. A brief discussion here may be warranted, especially an explanation on how values for u_star_threshold tie into the fluid stress and impact stress thresholds discussed here. A good "first principles" reference is in Schmidt, 1980.*

We agree with the reviewer. A summary of the main developments on parameterizations for the fluid threshold can indeed be included on section 2.2.1 when discussing the effect of the surface characteristics on the surface shear stress. As a follow-up, we will modify l.925-937 in the conclusions to acknowledge these important developments. However, we do not intend to make a full review of these parameterizations and how they are computed. Some of them are model-specific and cannot be easily used in other snow models that consider different parameters to describe the snow morphology. In l.938-942, we state that some aspects of the drifting snow models were not covered in this work (e.g., the fluid threshold) and that they need to be assessed for a complete evaluation of the model.

*4. The authors claim on lines 317 – 322 that the linear relationship between u\* and Q suggested by Pomeroy and Gray is not in agreement with most saltation models. I think this may be, at least partially, a consequence of that fact that many of the snow a majority saltation parameterizations referenced throughout (e.g., Vionnet et al., 2014) are based on parameterizations originally configured for soils. In consideration of the fact that Pomeroy and Gray, specifically claim that the linear relationship is a consequence of snow vs. soil, and that soil parameterizations are not "directly applicable to snow," I think this is worth elaborating on here. While understanding this is an older paper, and that more recent work (e.g., Melo et al. 2022) provides some evidence that PG is incomplete even when taking into reasonable account interparticle forces, it's worth making the*

*distinction between PG and other parameterizations here, since PG was based on measurements of natural snow in the field instead of wind-tunnels. I think this is important, especially in light of the fact that the LES simulations in this paper specifically ignore the inter-particle forces of the snow surface in an effort to replicate wind-tunnel measurements.*

Indeed, the model of Pomeroy and Gray (1990) is one of the few that take into account field measurements of snow saltation. This is now acknowledged in the text (l.368-370). However, we would like to highlight that these measurements are limited: they correspond to a series of mass flux measurements at a single height above the surface (~2 cm). This experimental set-up has two issues: 1) it is extremely challenging to keep a sensor at exactly 2 cm above the snow surface during a drifting snow event; 2) the mass flux is expected to decrease exponentially with height in the saltation layer. The exponential nature of the mass flux profile of snow particles in saltation was not only observed in wind tunnels, but also in the field (e.g., Nishimura and Nemoto, 2005). This means that small changes in height can lead to significant changes in the measured mass flux, which are not related to modifications in $u_*$ or snow surface characteristics. Therefore, we cannot neglect the errors introduced by considering a single point measurement and the assumption of a constant mass flux in the saltation layer. In addition, Pomeroy and Gray (1990) assumed that the particle velocity is constant along the saltation layer. From field measurements of snow saltation (Aksamit and Pomeroy, 2016), this approximation does not seem to be accurate. The particle velocity is expected to increase with height and $u_*$. This result is confirmed by wind tunnel measurements performed with snow, as well as numerical simulations (see section 2.3.2). The last assumption made by Pomeroy and Gray (1990) was that the height of the saltation layer scales with $u_*^2$. This scaling is discussed in this work. According to the findings of section 3.3, this result is expected if saltation is dominated by aerodynamic entrainment. Differently from sand or soils, this can indeed be the case for some snow surfaces. With all of this in mind, we find it difficult to claim that the results of Pomeroy and Gray (1990) are related to the particular nature of snow. By using the same dataset, but considering different assumptions for the particle mass flux and velocity profiles, a different scaling would be obtained. In our view, the fact that the model of Pomeory and Gray (1990) is one of the few that uses field measurements of snow saltation is the main reason for its popularity in the snow community. However, we would argue that that alone is not enough. More experimental measurements, particularly in the field, are needed to fully assess the validity of the model of Pomeory and Gray (1990). However, at the moment, the existing field, wind tunnel, and numerical studies performed with snow do not support its conclusions. We would like to keep this clear in the text.

*5. Equation 5 and Lines 595 – 602: The value of 0.1 for A in equation 5 for the aerodynamic fluid threshold, was that tuned to match the "disintegrated individual particles?" How were the equations for splash and rebound tuned to match this assumption? Were any other roughness lengths tried, I've often seen snow reported as 2x10-4 meters (double that of the value used here), and even higher for some land surface models. Presumably, the roughness length of the surface is used to generate the log-wind profile in the LES?*

The value of A=0.1 was proposed by Bagnold (1941) for saltating sand. Taking into account that we are neglecting interparticle cohesion, we have decided to use the value proposed by Bagnold (1941). This yielded a fluid threshold friction velocity of 0.154 m s$^{-1}$. As mentioned in l.680-684, Sugiura et al. (1998) have reported that at $u_* = 0.15$ m s$^{-1}$ saltation was only sustained with the addition of seeding particles. Therefore, this proved to be a good assumption that correctly reproduces the experiments.

Regarding the remaining rebound and splash parameters, they were not adjusted to better reproduce the experiments. The same values considered in Melo et al. (2022) were considered. This is now mentioned in the text (l.653-654). In l.693-697, we acknowledge that this fact is one of the potential reasons for the quantitative mismatch between simulations and measurements.

Regarding the roughness length, we have tried a lower value ($z_o=10^{-5}$ m), which is more in agreement with previous wind tunnel experiments developed with snow (Nishimura et al., 2014). However, negligible improvements were obtained. This is mentioned in the text (l.689-691). The value of $z_o=2x10^{-4}$ m is more in agreement with field measurements, but it might not be appropriate to reproduce wind tunnel experiments. The assumed roughness length is indeed an important parameter of the lower boundary condition of the LES solver (logarithmic law). As currently stated in l.755-758, the surface friction velocity estimated by the model is a direct function of the horizontal wind velocity resolved by the model at the first grid point (~5 mm) and the roughness length.

*6. Figure 4: "The decay height …. Follows approximately the trend proposed by NH (2002)". I'd disagree, to me it looks like there is simulations follow a relationship that looks more continuously proportional to the square-root of u_star, even below 0.3 m/s. It never appears to me that it is proportional to u^2. Though it would be helpful to see a value lower that 0.2 (if there is one). Given that this "agreement" is used to support a statement on Lines 653 – 654, it's worth a second look.*

Thank you for this comment. We agree with the reviewer that the decay height seems to follow a continuous evolution with respect to $u_*$. We have fitted the decay height to a square root function and to a logarithmic function (see figure below, not shown in the manuscript). A better fit was obtained with the latter. As expected from a logarithmic function, the rate of increase of the decay height with $u_*$ is higher at low friction velocities. Therefore, we believe we can still relate this result to the increase of the vertical ejection velocity as $u_*$ rises from 0.15 to 0.3 m s$^{-1}$. This is now acknowledged in the text (l.708-710). In our view, it is fair to state that the simulation results do not deviate considerably from the parameterization of Nishimura and Hunt (2000) at this low friction velocities. This statement was added to the text as a replacement to the original one (l.702-703). The conclusions were also slightly modified to better reflect the updated analysis of the results (l.911-913). The lowest value of $u_*$ considered in the simulations is equal to 0.15 m s$^{-1}$, which is slightly lower than the assumed fluid threshold friction velocity ($u_{*,ft}$ = 0.154 m s$^{-1}$). For $u_*$<0.15 m s$^{-1}$, the mass flux would be highly intermittent.

[Figure]

Figure I: Decay height as a function of the friction velocity. Fit to a square root function as well as a logarithmic function.

*7. Figure 9: Great figure!*

Thank you.

*8. Line 847: "particle mas flux" should be "particle mass flux"*

This was corrected in the text.

[revised manuscript text omitted]

---

## Author Response (AR1)

**Point-by-point reply to the comments**

The authors response to the comments of the first and second reviewers is presented in this document. Most replies are equal to those submitted on November 1ˢᵗ, 2023. Only the replies to the comments colored in **blue** were significantly edited. The line numbering in this document refers to the new Track-changes file. Please disregard the list of references and the location of figures and tables in the Track-changes file. Additional changes made in the manuscript are highlighted at the end of this document. Once again, we would like to thank the reviewers for their constructive comments and suggestions. In our view, they have truly improved the manuscript.

**Reply to Reviewer #1**

**General comment:**

*[…] As it stands, the manuscript needs to be improved prior to consideration for publication. The introduction material is currently a bit underdeveloped. The authors are presenting an advancement of a topic that has been studied in a large number of scenarios, which a wide range of applications as is shown in Section 2 when the actual parameterizations are being discussed. As well, the title suggests a kind of review of the state of the field. The authors do not present a sufficiently wide view of the topic in the introduction. I strongly suggest a deeper review of the available studies. Snow transport and snow saltation has been studied and modeled since at least the 1950's, but the case studies selected to summarize the field and the complexities of the topic are a bit sparse and lacking. For example, the authors have a tendency to cite Antarctic studies in the intro, but the benchmark snow flux profile study of Budd et al. (1966) in Antarctica is not referenced. Perhaps the title would be better suited as "Understanding snow saltation in atmospheric modeling parameterizations".*

Thank you for this comment. The introduction seeks to motivate the need to better assess snow saltation parameterizations and, therefore, the analysis performed in this manuscript. However, we agree that it can be expanded to provide to the reader a wider view of the topic. On this regard, the study of Budd et al. (1966) and the remaining works suggested by the reviewer were analyzed. Most of them are now cited in the text (Budd et al., 1966; Dyunin, 1967; Pomeroy et al., 1993; Schmidt, 1982; Dyunin and Kotlyakov, 1980; Gauer, 1998). In addition, we have addressed most of the specific comments targeting the introduction. Please check the replies to those comments (colored in blue). Nevertheless, we would like to clarify that this work does not intend to be a review of the state of the field as a whole, but to review snow saltation parameterizations and to further investigate them with a numerical model. Regarding the title, we would prefer to keep it as it is. In our view, it correctly captures the content of the manuscript. Even though we focus on parameterizations used in atmospheric models, we believe that this work is of interested to everyone requiring a simple description of snow saltation.

**Specific comments:**

*L20-22: Sastrugi?*

A mention to sastrugi was added to the text (l.21).

We now cite the works of Li and Pomeroy (1997) and Lenaerts et al. (2012), who mention this convention in their articles (l.29-33). In the websites of the World Meteorological Organization (WMO)[1] and of the American Meteorological Society (AMS)[2], the same definitions can be found. The definition given by the WMO was previously cited in the work of Gauer (2001).

[1]WMO  World Meteorological Organization. (2017). Hydrometeors consisting of an ensemble of particles raised by the wind: Drifting and Blowing Snow. Retrieved October 16, 2023, from https://cloudatlas.wmo.int/en/drifting-and-blowing-snow.html

[2]AMS American Meteorological Society. (2012). Glossary of Meteorology: Drifting Snow. Retrieved October 16, 2023, from https://glossary.ametsoc.org/wiki/Drifting_snow

*L28-39: This paragraph started off with descriptions of blowing snow, and then focuses solely on Antarctica. Given the amount of time you spend on non-Antarctic snow studies in the remainder of the manuscript, it's a bit strange to limit to such a specific niche here. Please include a broader review.*

We have included some estimates of drifting snow sublimation in the low Arctic (Pomeroy et al., 1997) and in Saskatchewan (Pomeroy et al., 1993). In addition, we now refer to some studies developed in mountain areas (Groot Zwaaftink et al., 2011, Lehning et al., 2000, Das et al., 2012) (l.42-50).

*L37-39: I am certain the authors can find an older reference to this complication than this.*

We now cite the work of Liston and Sturm (2002), who directly state and tackle this problematic (l.49-50). We have kept the citation to the work of Mott and Lehning (2010) because it distinguishes the effect of preferential deposition of snow during snowfall events with wind, from the effect of snow transport and redistribution.

*L40-46: Again, this focuses solely on drifting snow in Antarctica. This is a bit strange in a general discussion of drifting snow. Given the number of studies on snow sublimation from field-based, hydrological, and numerical studies, it also seems odd that the authors are using a regional weather model and satellite observations.*

With this paragraph, we intend to showcase the uncertainty of large-scale predictions of snow transport and sublimation. In the literature, we can find local measurements of snow transport and

sublimation. However, regional assessments based on extended measurements are challenging to obtain. The estimates based on satellite measurements rely on several assumptions and are characterized by a large standard deviation. Nevertheless, we believe that the disagreement between those estimates and the model results obtained with RACMO highlight the need for further investigation on the modeling side. In order to reduce the focus on Antarctica, we start the paragraph with a more general statement which highlights some experimental studies of snow transport performed in various snow-covered regions (l.52-54). The last sentence of the paragraph was also modified to be more in agreement with the text (l.59-61).

*L53-54: I do not believe it is widely accepted that snow particles that travel in suspension only arise from particles that have undergone fragmentation. There are no whole snow particles in suspension?*

We agree with the reviewer. We were presenting a narrow definition of particles in suspension. The reference to particle fragmentation was therefore deleted from the text (l.70-71).

*L56: This model of snow redistribution far precedes the referenced literature including, but not limited to (Dyunin, 1967; Lee, 1975; Dyunin, 1980; Pomeroy et al., 1993; Gauer, 1998; Bintanja, 2000)*

We agree with the reviewer. We have cited atmospheric and snow models that use the mentioned approach instead of the works that introduced this methodology. We have moved those references to the beginning of the paragraph and have added citations to the works of Dyunin and Kotlyakov (1980), Pomeroy and Male (1992), and Déry et al. (1998) (l.73-77). Earlier snow suspension models, as the one of Shiotani and Arai (1967), are not mentioned because they neglect spatial differences of particle concentration.

*L62: It would be helpful to cite the sublimation models that use the Thorpe and Mason approximations.*

The models that use the approximations of Thorpe and Mason (1966) are now cited in the text (l.81-82). The model MAR is an exception. Snow sublimation is computed with the model of Lin et al. (1983), which considers different assumptions in comparison with the model of Thorpe and Mason (1966).

*L66: Again, the references to modeling snow redistribution are a bit silo'd as they tend towards recent research from adjacent groups. These issues have been mentioned or studied for many decades, even in mountainous terrain (e.g. Dyunin et al., 1977; Schmidt, 1982; Gauer, 1998).*

Thank you for this comment. We have suppressed the reference to the work of Mott and Lehning (2010) and added the citation to the works of Gauer (1998) and Bernhardt et al. (2009) (l.85-86). We believe the reference list is now more diverse. Our goal was not to cite models of snow redistribution, but studies that directly address or question the impact of the topography on the wind field and, consequently, on snow transport and sublimation. This aspect is not present in the works of Dyunin et al. (1977) and Schmidt (1982) with sufficient extent. Therefore, we have decided not to cite them in this part of the text.

*L65-76: There is again a strange tendency towards Antarctic research, which is not where the majority of drifting snow research has actually been conducted. Snow redistribution was coupled in a GCM by Eric Brun back in the 1980's.*

In this paragraph, we intend to showcase the sensitivity of drifting snow models to the parameters chosen in the snow saltation model. In this context, we believe that the sensitivity studies performed with MAR and RACMO are good examples. In Noël et al. (2018), the atmospheric model RACMO is also used to compute the surface mass balance of Greenland. The same adjustment in the snow saltation parameterization performed by van Wessem et al. (2018) for Antarctica is considered in their study. However, no values of drifting snow sublimation are provided in the article. In Déry et al. (1998), the sensitivity of the model to some snow suspension parameters is presented, but no assessment is performed regarding the snow saltation parameterization. Even though their analysis is of interest to the overall assessment of drifting snow schemes, the assessment of snow suspension is outside the scope of this article. In Lehning et al. (2008), the effect of the threshold for saltation onset on the snow height along a ridge is investigated using Alpine3D. Negligible differences were obtained in the snow height patterns after reducing the threshold to half. However, during the studied event, snow saltation plays overall a minor role. Snow transport develops with concurrent snowfall and it is shown that the snow height variability is mainly due to the preferential deposition of snowfall. Additional post-processing of their results would be required to conclude about the effect of the threshold for saltation onset on the total mass aloft due to wind erosion and its sublimation. Taking into account the lack of additional sensitivity analysis on the literature, no changes were made in the text regarding this comment.

Thank you for the suggestion. We agree that the term "spanwise" can be used as a synonym for "crosswise". However, in our view, the term "crosswise" relates well with the term "cross section area", so we would prefer to keep it.

*L103-105: What do you mean statistically invariant? The temporal-average of windspeed is constant on each wall-parallel plane? And can you define "mainly" aligned?*

Yes, with "statistically invariant along the horizontal directions" we want to convey that the temporal-averages of wind velocity do not vary in x and y, therefore, they are constant on each wall-parallel plane. We have reformulated the sentence to make it more clear (l.135-138). With "mainly aligned with the streamwise direction", we intended to inform the reader that the x-axis is aligned with the time-average horizontal flow direction. However, taking into account that this is a natural consequence of imposing a streamwise pressure gradient, this information was deleted from the text.

*L106-110: Strictly speaking, between the viscous sublayer at the surface and the inertial sublayer is a buffer sublayer where you do not have a fluid velocity profile that provides a constant shear stress. Furthermore, the log-layer or inertial sublayer also only exists for a finite height in the atmosphere. It would benefit the authors and the reader if the authors could address what assumptions we need to make for this sort of shear stress profile they are describing, and what is assumed to happen to the buffer layer in the presence of saltating particles. Typically some sort of roughness height is included in a log-law profile (which comes from the constant shear stress assumption) at which the wind essentially goes to zero above the actual snow surface, though this is widely known to not be physically true, but it helps close the equations. This roughness length is not mentioned anywhere, though often also modified to account for the momentum deficit caused by the saltation layers that the authors are referencing. This dates back to Bagnold.*

The reviewer is making a link between a logarithmic profile and a constant shear stress. However, in rigor, the link that exists is between a logarithmic profile and a constant turbulent shear stress. Over a smooth surface and in the absence of saltating particles, the total shear stress (the sum of viscous and turbulent shear stresses) is expected to be constant over the whole inner layer (viscous, buffer, and logarithmic sublayers) (Schlichting and Gersten, 2017: eq. 17.1 in p.520). However, in the viscous sublayer, the viscous shear stress is dominant and the resultant velocity profile is linear, while in the logarithmic sublayer the turbulent shear stress is dominant and the velocity profile is

logarithmic. Over a rough surface, the viscous sublayer is disrupted by the roughness elements and the inner layer is better characterized by a roughness sublayer and a logarithmic sublayer. This was made clear in the text (l.140-167). In this new paragraph, we have also stated the region of validity of the logarithmic profile in the atmospheric boundary layer and the use of an equivalent roughness length to describe the velocity profile above the saltation layer. In addition, we have stated the assumption used in atmospheric models as suggested by the second reviewer. Taking into account that the size of this section increased substantially, we have decided to create a new subsection (2.1 The steady-state assumption), which is preceded from an introductory paragraph (l.122-131). In this way, l.184-191 were deleted from the text.

*L171-172: Did Bagnold show that the necessary shear stress was smaller, or that a smaller friction velocity was sufficient, given that \tau=\rho u_*^2 is only an "effective" shear stress at the surface.*

We have clarified the definition of $\tau=\rho.u_*^2$, which is now defined as the "time-averaged fluid shear stress in the logarithmic sublayer" (l.151). Thus, $\tau$ can be interpreted as the value of shear stress measured with the eddy covariance method or from fitting the wind velocity profile to a logarithmic function. Therefore, we would prefer not to name $\tau$ the "effective shear stress at the surface", but to clearly distinguish $\tau$ (the shear stress in the logarithmic sublayer) from $\tau_s$ (the shear stress at the surface). These two quantities assume equal values only in the absence of saltating particles or at the thresholds for saltation onset and cessation. In this context, the difference between "the necessary shear stress was smaller" and "a smaller friction velocity was sufficient" is not clear to us. Bagnold (1941) deduced the friction velocity from wind velocity measurements performed in the logarithmic sublayer. In this way, Bagnold's conclusions refer to the friction velocity $u_*$ and not $u_{*,s}$ (as defined in the manuscript). However, as mentioned above, at the limiting threshold shear stresses (fluid and impact thresholds) the surface shear stress and the shear stress outside the saltation layer are expected to assume similar figures due to the low number of particles aloft and the respective small momentum exchange between the fluid and the particles. This is now mentioned in the text (l.239-243).

*L173: Requires, not implies?*

This was modified in the text (l.244).

*L192-193: What is the fluid threshold friction velocity?*

In the manuscript, the fluid threshold was defined in the introduction (l.90-91). However, as this is

not ideal, the fluid threshold is now presented at the beginning of section 2.2.1 (l.224-226). The introduction was modified accordingly (l.90-91, l.94-95).

*L193-194: Please rewrite this sentence. It is equal because it is assumed to be a fraction, and we assume this fraction is 1?*

The assumption that $\tau_s$ is equal to $\tau_{ft}$ (therefore, that $\tau_s$ is a fraction of $\tau_{ft}$ and that this fraction is assumed to be 1, as pointed out by the reviewer) is not in agreement with Owen's claim. Owen (1964) specifically claims that $\tau_s$ must be smaller than $\tau_{ft}$ because the particle bombardment at the surface contributes to dislodging the grains. The text was modified to make this claim more clear (l.236-238, l.262-269). Following a comment from the second reviewer, the parameterizations used in atmospheric/snow models to compute the fluid threshold are now mentioned in the text.

*L196: Nor is it in agreement with experimental evidence dating back to Bagnold.*

A reference to the wind tunnel experiments of Bagnold (1941) was added to the text (l.272).

*L196: Now i'm confused. I thought \tau_s was the approximated surface shear stress, which varies with windspeed. I believe you are actually referencing \tau_it, but using its equality with \tau_s in this one instance. Can you replace \tau_s with \tau_it to make the following inequalities clearer?*

In the text, $\tau_s$ is defined as the actual (not approximate) surface shear stress during saltation (see l.157-159 and l.166-167). In addition, we follow Bagnold (1941) and define $\tau_{it}$ as the lower shear stress outside the saltation layer for which saltating particles maintain a steady state motion (l.235-236). Therefore, the impact threshold is a well defined value that characterizes the bed and it is invariant with respect to $u_*$. Differently, $\tau_s$ may vary with $u_*$. As discussed in section 2.2.1, Owen (1964) hypothesizes that $\tau_s = \tau_{it}$, atmospheric models frequently assume that $\tau_s = \tau_{ft}$ (e.g., Lenaerts et al., 2012; Amory et al., 2021; Vionnet et al., 2014), but some numerical models revealed that $\tau_s$ is not invariant with respect to $u_*$. We hope that Table 1 (page 9 in Track-changes file), suggested by the second reviewer, helps clarifying the different shear stresses.

*Does Figure 2 include the decay height of all experiments mentioned in Section 2.2.3?*

No, Figure 2 includes only the experimental measurements of Sugiura et al. (1998), Namikas (2003), Nishimura and Nemoto (2005), and Martin and Kok (2017), as stated in l.547-549. The obtained values of decay height are discussed in the paragraphs that follow. The results of Sato et al.

(2001), which are also mentioned, are not presented in Figure 2 because the respective value of $u_*$ was not provided by the authors. This is currently specified in the text (l.566-567). In l.589-590, we also refer to the results of Nalpanis et al. (1993), but we do not present them in the figure. As they consist of only 2 data points, we have decided not to include them in the comparison.

*L552: What is the subgrid-scale relative to the particle size? At what scale can we expect the snow particle to stop responding to fluctuations, and why?*

As mentioned in section 3.2, the domain is a cube of 6.4 m length, uniformly discretized in each horizontal direction in 64 cells (i.e, $\Delta x=\Delta y=0.1$ m). The mesh in the vertical direction follows a hyperbolic stretching: it varies from $\Delta z=0.01$ m along the first 15 cm above the surface to $\Delta z=0.1$ m close to the top boundary. Therefore, $\Delta=(\Delta x.\Delta y.\Delta z)^{1/3}$ varies between 0.05 and 0.1 m. Taking into account that the mean diameter is set to $360.10^{-6}$ m, the ratio between $\Delta$ and the mean diameter varies between approximately 130 and 280.

The ability of the snow particles to follow the velocity fluctuations can be evaluated by computing the Stokes number, St, which is a function of the particle relaxation time and the velocity and length scales of interest. If St<<1, the particles will mainly follow the flow. The opposite is true if St>>1. Therefore, the maximum frequency of the wind velocity signal that is able to influence a particle trajectory depends on the particle diameter and Reynolds number. This analysis was not performed in the context of this work.

*L559-561: This is not a comprehensive list of all forces acting on a particle in a turbulent flow. Beyond gravity (buoyancy) and drag, there are also force terms from the wind, the force from the fluid moving with the particle, and the Basset-Boussinesq memory (Maxey and Riley, 1983; Talaei and Garrett, 2020).*

We agree with the reviewer. Indeed, we are not presenting a complete list of forces applied on the particles. Our intention was to provide some examples. We have rephrased the text to make this more clear (l.642-643).

*L566: "entrainment"*

Corrected.

*L570: Is it assumed there is an infinite number of particles that are available to be entrained?*

A sufficiently large initial deposition is specified in the simulations in order to guarantee that snow transport is never limited by a shortage in the supply of erodible particles. This is now mentioned in the text (l.658-659).

Corrected.

*Figure 5: Can you discuss these decreases in the inset plots more? This seems like an interesting point that is glossed over rather quickly. As well, it would be helpful to include two more plots that compare the average particle velocities with the average wind speeds in the two directions at each height, for the various scenarios. The influence of vertical winds on ascending saltating particle is particularly interesting to think about as this is a primary factor driving the transition to suspension and away from the paradigms described here. Perhaps a vertical profile of vertical drag (normalized by gravity forces) could reveal these relationships?*

Thank you for this comment. We have added the average vertical profiles of streamwise wind velocity to the manuscript (Figure 7). This plot shows that the model correctly predicts the existence of a focus point when saltation is dominated by splash (please note that the focus point was previously introduced in section 2.3.2, l.496-497). The analysis of the wind velocity profiles allowed us to better understand the observed decrease in near-surface particle velocity (l.780-798). In this context, the comment regarding the particles deceleration as they approach the surface was deleted (l.798-801). In fact, above 5 mm height above the surface (first grid point), the average wind velocity is always greater than the average particle streamwise velocity. This can be seen in more detail in the figures below, where we present separately the average streamwise velocity of the upward and downward moving particles (not shown in the manuscript). If we extend the wind velocity profile below the first grid point by assuming a logarithmic profile characterized by $u_{*,s}$ and $z_o=10^{-4}$ m, the average streamwise wind velocity will be smaller than the average streamwise velocity of the downward moving particles below approximately 2 times the average grain diameter. However, in the model, the surface processes of rebound and splash occur as soon as the particles reach a height lower than 4 times the average particle diameter (1.4 mm in the current setup). Therefore, in the simulations, the drag force in the streamwise direction is on average always positive and the decrease of the near-surface particle velocity as $u_*$ rises above 0.39 m s$^{-1}$ is mainly related to the respective decrease in streamwise wind velocity below the focus point (~ 1 cm

height). The lower the wind velocity in the vicinity of the bed, the lower the ability of the flow to accelerate the particles close to the surface.

[Figure]

Figure I: Vertical profiles of average wind and particle streamwise velocity. The thick lines correspond to the wind velocity, the thin lines with upward pointing triangles correspond to the upward moving particles, and the dashed lines with downward pointing triangles correspond to the downward moving particles.

Following the reviewer's comment, the average wind velocity in the vertical direction is presented in the figure below (not shown in the manuscript). As expected from a half-channel flow, it is mainly zero (a zero vertical velocity is imposed at the bottom and top boundaries). It reaches higher values close to the surface, but it is nevertheless several orders of magnitude smaller than the average particle vertical velocity (Figure 5b). Therefore, over a flat surface subjected to a streamwise pressure gradient, the average vertical wind velocity has a negligible effect on driving the transition from saltation to suspension. Of course, this might not be the case over complex terrain where vertical updrafts and downdrafts are common. In the absence of an average vertical wind velocity different from zero, small particles reach high elevations due to the effect of velocity fluctuations. High instantaneous velocities may induce an instantaneous vertical drag force greater than the particle weight, which transports the particles upwards. Overall, while the gravitational force pushes the particles downwards, turbulence promotes particle dispersion and a homogeneous concentration of particles aloft.

[Figure]

Figure II: Vertical profile of average wind velocity in the vertical direction obtained for different friction velocities. In the inset, a zoom-in to the near-surface region is presented.

The vertical profiles of aerodynamic drag in the streamwise and vertical directions are presented below for completeness (not shown in the manuscript). As expected, the vertical component of aerodynamic drag, $F_z$, is much smaller than its streamwise component, $F_x$.

[Figure]

Figure III: Vertical profiles of aerodynamic drag in the streamwise and vertical directions normalized by the gravitational force. The different colors correspond to the different friction velocities. The upward pointing triangles correspond to the upward moving particles and the downward pointing triangles correspond to the downward moving ones.

*L606-607: How many actual lagrangian particles were in transport at a given time? How does this assumption of clustering work? When a splash event occurs at a friction velocity of 0.3, you automatically assume 50 snow particles are ejected at that single location? And when you have an impact of a Lagrangian parcel, are you calculating the force of 50 times the mass equally distributed among the splashed particles? Admittedly this would create more pseudo-particles in transport while not requiring any additional Lagrangian computations, but the fact that each parcel at a given friction velocity has the same number of particles is worrisome. I see it would be difficult to truly track all the particles in a 6 m cube, but you are also manipulating the mass flux by enforcing these rules. I think there may be negligible effect on the relative concentration profiles, but some further explanation is needed here. If you instead chose to have 1000 particles in a lagrangian parcel, would this have a noticeable impact on your snow-wind feedback?*

The total number of parcels aloft (lagrangian particles) averaged along the last 100 s of each simulation are presented in Figure IVa (not shown in the manuscript). Depending on the friction velocity, we simulate on average between $10^4$ and $3.5 \times 10^4$ parcels. The total number of particles aloft (number of parcels x number of particles per parcel (PPP)) is presented in Figure IVb (not shown in the manuscript). Particles of the same parcel have the same diameter and follow the same trajectory. Therefore, they were aerodynamically entrained or splashed at the same surface location and time step (this is stated in the text in l.694-696).

[Figure]

Figure IV: Average number of parcels aloft obtained for different friction velocities (a) and the respective number of particles aloft (b). The error bars correspond to two times the standard deviation.

For $u_*=0.3$ m s$^{-1}$, for which PPP=100 (see corrected PPP values in l.697), the impact of one parcel corresponds to the impact of 100 particles. Taking into account the mass and impact kinetic energy

of 1 particle, we compute how many particles are splashed. Let's imagine that the model yields the value of 3. In this case, each particle ejects 3 particles, which means that the 100 particles will eject a total of 300 particles. Therefore, 3 new parcels (with 100 particles each) are ejected into the flow. For each ejected parcel, the diameter and initial ejection velocity of all particles in that parcel are a random realization of a normal and exponential distributions, respectively.

Indeed, the clustering of particles in groups reduces the variability of trajectories and grain size aloft represented by the model. Nevertheless, this simplification is expected to yield acceptable figures if the total number of parcels aloft is large enough so that the variability of trajectories and grain size is sufficiently represented. In the figure below, we present the effect of the number of PPP on the surface friction velocity averaged over the last 100 s of the simulations. It can be seen that the number of PPP has a more significant impact on the results for $u_* > 0.3$ m s$^{-1}$. For $u_*=0.39$ m s$^{-1}$, similar results were obtained when considering PPP = 200 in comparison with the simulations for which the number of PPP was allowed to vary between 500 and 1000. Therefore, PPP = 200 was considered acceptable for the analysis of average quantities undertaken. For the higher friction velocities ($u_* > 0.4$ m s$^{-1}$), additional simulations with a PPP smaller than 200 are still needed to confirm the convergence of the simulation results. However, this verification was not performed due to the computational cost of the simulations. In our view, a sensitivity analysis to the different model parameters is outside the scope of this article. Therefore, this discussion was not added to the manuscript.

[Figure]

Figure V: Average surface friction velocity obtained for different friction velocities and different number of PPP. In the legend, the number of PPP is presented in increasing order of $u_*$. The last value of the array is representative of the remaining (larger) friction velocities.

*L619: How is steady state quantified? Any subset of the 100 seconds would provide the same mean profiles? Can you please provide some verification of this?*

The time series of the mass aloft per unit surface area is presented on the figure below for the various friction velocities studied (not shown in the manuscript). In the simulations, saltation is allowed to take place after t=25 s. Following an initial rise in mass aloft, the mass of particles decreases with time until it reaches a stable value. From observing the time series, we have assumed that this equilibrium was reached after 250 s for all simulations. We have adapted l.707-708 to better illustrate our approach.

[Figure]

Figure VI: Time series of the mass of particles aloft per unit surface area obtained with different friction velocities. The inset is a zoom-in to the three lowest values of u∗.

In Figure VII (not shown in the manuscript), we present the average mass aloft computed with an increasing number of points (the value at t=250 s is the instantaneous mass aloft at t=250 s and the value at t=350 s is the average considering the previous 100 s of the simulations). It can be seen that 100 s is sufficient to arrive to a relatively stable average. For the higher friction velocities ($u_* > 0.5$ m s$^{-1}$), improved statistics would probably be obtained by extending the computational time (for instance, with additional 100 s). This was not performed due to the computational cost of the simulations.

[Figure]

Figure VII: Average mass of particles aloft per unit surface area for an increasing number of data points. Results obtained for all friction velocities studied.

*L623-624: Most of the numerical mass flux profiles.*

Even though the fit is not shown, this is also true for the presented experimental profiles. Therefore, no modification was made in the text.

*L657: Does the average vertical ejection velocity combine both the splashed particles and the rebounded particles? Please clarify here.*

The average vertical ejection velocity is computed from all particles aloft (aerodynamically entrained, splashed or rebounded particles). Throughout the text, we use the term "ejection velocity" as a synonym for the initial particle velocity without making any distinction regarding its origin (see sections 2.2, 2.2.2, and 3.1). Therefore, we believe that it is not necessary to clarify this in the text.

*Figure 5: I think $v_z^{up}$ would be a better notation as you are elsewhere using superscripts to denote subsets of velocities in the direction specified in the subscript*

Thank you for the suggestion. This was modified in the figure.

*L664: Particle velocity profile is mainly invariant*

This was modified in the text.

*L678-679: Please clarify if this is something you are enforcing in the model, or a result of your model.*

In the model, the ejection velocity of aerodynamically entrained particles is a random realization from a lognormal distribution. However, the mean value of the distribution is proportional to $u_{*,s}$. This was made clear in the text when describing the numerical model (l.655-657). Therefore, when aerodynamic entrainment is the main entrainment mechanism, which is the case for $u_*$ varying between 0.15 and 0.3 m s$^{-1}$, the average ejection velocity will be necessarily proportional to $u_{*,s}$. This was made clearer in the text by replacing "is justified by" with "is a direct consequence of" (l.772). Nonetheless, the fact that aerodynamic entrainment is the main entrainment mechanism for $u_*$ varying between 0.15 and 0.3 m s$^{-1}$ is a result of the model.

*Figure 6: As stated above I think it's weird to be using mass flux instead of a relative number flux given that the mass flux seems to be directly modified by this lagrangian parcel idea.*

When computing the mass flux profile, the mass of each parcel aloft is computed by multiplying the mass of one individual particle times the number of PPP. Similarly, to compute the number flux profile, the number of particles in each parcel aloft would be equal to the number of PPP. In both calculations, each of these particles is assumed to travel at the same velocity: the velocity of the parcel. Therefore, we are expecting the mass flux and the number flux to be equally sensitive to the number of PPP. In this way, we disagree with the reviewer's comment and no modifications were made in the text.

*L682-683: I thought the relationship between impacting particle and splash velocity was stochastic?*

In the model, the ejection velocity of splashed particles is a random realization from an exponential distribution. The mean value of the distribution is a function of the particle impact velocity. This was made clear in the text when describing the numerical model (l.668-669). In addition, the word "average" was added to the text (l.762).

*L685-686: It would be good to show wind profiles for your full range of friction velocities from the surface to above the saltation layer and that show and quantify this invariance as that is actually a non-trivial conclusion. What do relative velocity changes look like at each height?*

As suggested, the streamwise wind velocity profiles were added to the manuscript (see reply to comment on Figure 5 in page 10). The streamwise wind velocity is approximately invariant with respect to $u_*$ in the near-surface region when saltation is dominated by splash. This is due to the existence of a focus point which was previously observed by Bagnold (1941) in wind tunnel experiments. From Figure I in this document, one can assess the relative velocity in the first 5 cm above the surface. As expected, the upgoing particles experience a greater relative velocity than the downgoing ones.

*L686-687: This is an interesting point as well. When do particles in saltation have a change in sign for the drag? Assuming a non-slip velocity at the immediate snow surface, it should always occur somewhere. Is this accounted for in your snow-wind feedback? As we approach steady-state, and snow particles are already quickly moving, at what fraction of maximum trajectory height does that of transition drag sign-change occur? How does this momentum source for the wind balance with the momentum sink caused by the presence of particles? In turn, what role does this play in modifying u\* or u\*,s?*

As mentioned in the reply to comment on Figure 5 in page 10, for the simulated conditions, the change in sign for the drag may only occur below a height of approximately 2 times the average particle diameter. However, on average, this does not occur in the simulations because the surface processes of rebound and splash are computed as soon as the particles are at a height lower than 4 times the average particle diameter. In this way, the statement to whom this comment is concerned was deleted from the text. As mentioned in l.644-646, the feedback of the particles on the flow momentum is taken into account by the addition of a reaction force in the filtered Navier-Stokes equations, which is equal in magnitude and opposite to the aerodynamic drag applied to the particles. Therefore, if a particle is eventually faster than the surrounding wind velocity, it will experience a negative drag force, which is taken into account in the momentum equations as a positive source term. Nevertheless, these body forces on the momentum equations are generally negative (sink terms) and lead to an overall reduction of the wind velocity. As discussed in the reply to comment on Figure 5 (page 10), when considering a constant roughness length, a smaller near surface wind velocity implies a lower surface friction velocity $u_{*,s}$. This was made clear in the manuscript when discussing Figure 7 (l.790-795).

Regarding $u_*$, it is defined in the text as the friction velocity in the logarithmic region, $u_* = \sqrt{(\tau/\rho_a)}$, where $\tau$ is the shear stress in the logarithmic region (equal to the surface shear stress, $\tau_s$, only in the absence of saltating particles) and $\rho_a$ is the air density. As previously mentioned, this was made more clear in the text (l.148-153). In a half-channel flow without saltating particles, the fluid shear stress depends only on the imposed pressure gradient and on the height of the channel (l.638-639). Therefore, $u_*$ does not vary with the surface roughness. As described in Bagnold (1941, page 53-55), changes in the roughness length modify the magnitude of the wind velocity, but not the shear stress profile, and therefore, the value of $u_*$. During saltation, Owen (1964) hypothesized that "the saltation layer behaves, so far as the flow outside it is concerned, as an aerodynamic roughness whose height is proportional to the thickness of the layer". Therefore, outside the saltation layer, similar values of $u_*$ will be obtained with or without particles in saltation. The feedback of the particles on the flow decreases the streamwise wind velocity, but does not modify the shear stress outside the saltation layer.

*L732-733: What do you mean perturbed? How was this set of trajectories filtered? It sounds like you may be intentionally neglecting the impact of wall-normal winds and trying to only focus on ballistic-like?*

In the analysis presented in the manuscript, we have neglected saltating particles whose trajectory exhibited at least one local minimum far away from the surface (above 1 cm height). The goal was indeed to only consider trajectories that did not deviate considerably from classic ballistic ones because the hop height is expected to be a stronger function of the vertical ejection velocity in these cases. However, taking into account that we compare the average hop height with the average vertical ejection velocity considering all particles in saltation (and not only those that follow ballistic trajectories), we now recognize that this is not ideal. Therefore, we have modified the post-processing of the results and updated Figure 8b. In the current version, we have defined a particle hop as a fraction of a particle trajectory limited by two consecutive local minimums located close to the surface. The hop height is given by the maximum height attained by the particle between these two local minimums. This is currently explained in the text (l.845-846). Higher values of average hop height are obtained with the new post-processing, but similar conclusions are obtained. Slight changes were made in the text regarding the interpretation of the results (l.852-873).

*L735-736: The particles never reach the surface? Does the wind field extend all the way to the surface?*

As mentioned in the manuscript (l.849-851), in the model, the surface processes of rebound and splash are computed as soon as the particles reach a height lower than 4 times the average grain diameter (0.14 cm in this case). Therefore, the particles are assumed to rebound or deposit before reaching z = 0 m.

Regarding the wind field, the first grid point of the Eulerian mesh is placed at 5 mm above the surface. Below this height, the wind velocity in the horizontal direction is assumed to follow a logarithmic profile. This is now specified in the text when discussing the wind velocity profile (l.790-791).

*Figure 9: In the splash-dominated saltation regime, it appears the saltation height is constant with variations in u\*, and with constant mean particle velocity, correct? Then, researchers that are studying blowing snow in highly-unsteady wind should always measure the same snow particle velocities near the surface, as long as the windspeeds stay above the threshold and are averaged for 100 seconds? This seems like an unphysical conclusion to come to as there is no upper limit on wind speeds at the snow surface above the splash-dominated threshold, but you are imposing a clear upper limit on particle speeds. As well, would the height of the saltation layer not actually decrease with increasing windspeed as near-surface wind fluctuations (and fast particles) lead to quick suspension? This seems like a highly relevant question, especially for polar (and even high mountain regions) where 100km windspeeds are easily attained and shape the landscape.*

Yes, in the splash-dominated saltation regime, both the near-surface particle velocity and the saltation layer height are expected to be invariant with respect to $u_*$. This result is not only based on the current numerical simulations, but also on the wind tunnel measurements of particle velocity of Creyssels et al. (2009) and Ho et al. (2011) developed with sand for $u_*$ varying from 0.24 to 0.67 m s$^{-1}$ and from 0.42 to 1 m s$^{-1}$, respectively, the field measurements of Aksamit and Pomeroy (2016) of particle velocity over fresh snow ($u_*$=0.21-0.35 m s$^{-1}$) and old snow ($u_*$=0.24-0.54 m s$^{-1}$), and the numerical results of Ungar and Haff (1987), Kok and Renno (2009), and Durán et al. (2011). As long as the wind profiles form a focus point, there will be an upper limit for the wind speed close to the surface as $u_*$ increases. However, we agree that the presented dynamics might not represent high wind velocity conditions in the field, where the size of the turbulent eddies is larger than the size of those obtained in wind tunnels with a height of O(1) m and simulated in a half-channel with a height of O(1-10) m. These large eddies will enhance the entrainment of particles in suspension and,

perhaps, limit or suppress the motion of particles in saltation. The situation described goes beyond what the current model is able to simulate and what the existing wind tunnel and field measurements were able to assess. This is now acknowledged in the text (l.905-910).

**Reply to Reviewer #2**

**Specific comments:**

*1. The authors go through great lengths to distinguish the different types of shear stresses within the surface layer (e.g., fluid, surface, impact, etc). Given that this paper has a strong focus on the atmospheric aspects, I think it's worth discussing how u_star is determined through atmospheric models and how that value can relate to the different u_star values that comprise the different surface stresses. Further, I think it is also worth a brief discussion of drag partitioning (i.e., the loss of surface momentum to non-erodible elements) (e,g., Marsh et al., 2020, as an example of a parameterization with drag-partitioning). While maybe not critically important for much of the work performed in this paper, since much of this paper reviews the details of snow saltation at very small scales, it may be worth some brief discussion to tie in with other parameterzations used at regional atmospheric modeling scales.*

In atmospheric models, the near-surface horizontal wind velocity is assumed to follow a logarithmic profile (modified according to stability corrections). Therefore, the friction velocity, $u_*$, is a function of the horizontal wind velocity at the first grid point above the surface, the assumed roughness length, which varies according to the land-use, and the stability parameter. This is now summarized in the text (l.153-157). This brief clarification is part of a longer exposition about the structure of boundary layer, suggested by the first reviewer (l.140-159).

In addition, we currently refer in the text that equation 1 does not take into account the effect of non-erodible roughness elements (l.214-216). The atmospheric models that we analyze in this work (RACMO, MAR, Meso-NH, and CRYOWRF) do not include a parameterization for drag partition. Therefore, we have decided not to discuss it in detail, but to simply cite the works of Pomeroy and Gray (1990) and Raupach et al. (1993).

*2. As a sub-point, I think it may improve readability to have a table somewhere in section 2.1.1 listing the different surface stresses, though I will leave that choice to the authors. It can be hard to keep all of these straight, especially when many models use only one friction speed value.*

Thank you for this suggestion. We have added a table in section 2.2.1 (Table 1) (please note that the sections numbering changed). As suggested we present the different shear stresses defined in the manuscript. We also highlight in the table footnote that an alternative definition of the impact

threshold is considered in section 3.3. The explanation of this alternative definition was slightly improved (l.875-876, l.880-882) and is now clearly stated when describing Fig. 10 (l.898-899).

*3. The authors several times make reference to the fact that "snow surface" characteristics are critically important for accurately simulating the different variables that contribute the particle mass flux at the top of the saltation layer, however there is very little comment on the research that has been performed characterizing the threshold friction speeds for blowing snow algorithms, or any of the existing parameterizations in the literature. A brief discussion here may be warranted, especially an explanation on how values for u_star_threshold tie into the fluid stress and impact stress thresholds discussed here. A good "first principles" reference is in Schmidt, 1980.*

Thank you for this comment. We now mention in the text that the fluid threshold used in RACMO, MAR, Meso-NH and CRYOWRF is a function of the snow surface characteristics and cite the works where these parameterizations can be found (l.265-268). Taking into account that our goal is to discuss the surface shear stress and not the fluid threshold, we have decided not to expand on the parameterizations themselves and the theory behind them. In the conclusions (l.977-981), we state that some aspects of the drifting snow models were not covered in this work (e.g., the fluid threshold) and that they need to be assessed for a complete evaluation of the models. A slight modification was made in the conclusions to make our critic regarding the effect of the snow surface characteristics on the snow saltation parameterizations more clear (l.966).

The fluid threshold computed by atmospheric/snow models can be directly compared with the fluid threshold defined in this work. However, the equality between fluid threshold and surface shear stress must be regarded only as a approximation. We believe that Table 1 and l.262-269 make this point more clear.

*4. The authors claim on lines 317 – 322 that the linear relationship between u\* and Q suggested by Pomeroy and Gray is not in agreement with most saltation models. I think this may be, at least partially, a consequence of that fact that many of the snow a majority saltation parameterizations referenced throughout (e.g., Vionnet et al., 2014) are based on parameterizations originally configured for soils. In consideration of the fact that Pomeroy and Gray, specifically claim that the linear relationship is a consequence of snow vs. soil, and that soil parameterizations are not "directly applicable to snow," I think this is worth elaborating on here. While understanding this is an older paper, and that more recent work (e.g., Melo et al. 2022) provides some evidence that PG*

*is incomplete even when taking into reasonable account interparticle forces, it's worth making the distinction between PG and other parameterizations here, since PG was based on measurements of natural snow in the field instead of wind-tunnels. I think this is important, especially in light of the fact that the LES simulations in this paper specifically ignore the inter-particle forces of the snow surface in an effort to replicate wind-tunnel measurements.*

Indeed, the model of Pomeroy and Gray (1990) is one of the few that take into account field measurements of snow saltation. This is now acknowledged in the text (l.400-402). However, we would like to highlight that these measurements are limited: they correspond to a series of mass flux measurements at a single height above the surface (~2 cm). This experimental set-up has two issues: 1) it is extremely challenging to keep a sensor at exactly 2 cm above the snow surface during a drifting snow event; 2) the mass flux is expected to decrease exponentially with height in the saltation layer. The exponential nature of the mass flux profile of snow particles in saltation was not only observed in wind tunnels, but also in the field (e.g., Nishimura and Nemoto, 2005). This means that small changes in height can lead to significant changes in the measured mass flux, which are not related to modifications in $u_*$ or snow surface characteristics. Therefore, we cannot neglect the errors introduced by considering a single point measurement and the assumption of a constant mass flux in the saltation layer. In addition, Pomeroy and Gray (1990) assumed that the particle velocity is constant along the saltation layer. From field measurements of snow saltation (Aksamit and Pomeroy, 2016), this approximation does not seem to be accurate. The particle velocity is expected to increase with height and $u_*$. This result is confirmed by wind tunnel measurements performed with snow, as well as numerical simulations (see section 2.3.2). The last assumption made by Pomeroy and Gray (1990) was that the height of the saltation layer scales with $u_*^2$. This scaling is discussed in this work. According to the findings of section 3.3, this result is expected if saltation is dominated by aerodynamic entrainment. Differently from sand or soils, this can indeed be the case for some snow surfaces. With all of this in mind, we find it difficult to claim that the results of Pomeroy and Gray (1990) are related to the particular nature of snow. By using the same dataset, but considering different assumptions for the particle mass flux and velocity profiles, a different scaling would be obtained. In our view, the fact that the model of Pomeory and Gray (1990) is one of the few that uses field measurements of snow saltation is the main reason for its popularity in the snow community. However, we would argue that that alone is not enough. More experimental measurements, particularly in the field, are needed to fully assess the validity of the model of Pomeory and Gray (1990). However, at the moment, the existing field, wind tunnel, and numerical studies performed with snow do not support its conclusions. We would like to keep this clear in the text.

*5. Equation 5 and Lines 595 – 602: The value of 0.1 for A in equation 5 for the aerodynamic fluid threshold, was that tuned to match the "disintegrated individual particles?" How were the equations for splash and rebound tuned to match this assumption? Were any other roughness lengths tried, I've often seen snow reported as 2x10-4 meters (double that of the value used here), and even higher for some land surface models. Presumably, the roughness length of the surface is used to generate the log-wind profile in the LES?*

The value of A=0.1 was proposed by Bagnold (1941) for saltating sand. Taking into account that we are neglecting interparticle cohesion, we have decided to use the value proposed by Bagnold (1941). This yielded a fluid threshold friction velocity of 0.154 m s$^{-1}$. As mentioned in l.716-717, Sugiura et al. (1998) have reported that at $u_* = 0.15$ m s$^{-1}$ saltation was only sustained with the addition of seeding particles. Therefore, this proved to be a good assumption that correctly reproduces the experiments.

Regarding the remaining rebound and splash parameters, they were not adjusted to better reproduce the experiments. The same values considered in Melo et al. (2022) were considered. This is now mentioned in the text (l.687-688). In l.728-732, we acknowledge that this fact is one of the potential reasons for the quantitative mismatch between simulations and measurements.

Regarding the roughness length, we have tried a lower value ($z_o=10^{-5}$ m), which is more in agreement with previous wind tunnel experiments developed with snow (Nishimura et al., 2014). However, negligible improvements were obtained. This is mentioned in the text (l.724-726). The value of $z_o=2x10^{-4}$ m is more in agreement with field measurements, but it might not be appropriate to reproduce wind tunnel experiments. The assumed roughness length is indeed an important parameter of the lower boundary condition of the LES solver (logarithmic law). As currently stated in l.790-793, the surface friction velocity estimated by the model is a direct function of the horizontal wind velocity resolved by the model at the first grid point (~5 mm) and of the roughness length.

*6. Figure 4: "The decay height …. Follows approximately the trend proposed by NH (2002)". I'd disagree, to me it looks like there is simulations follow a relationship that looks more continuously proportional to the square-root of u_star, even below 0.3 m/s. It never appears to me that it is proportional to u^2. Though it would be helpful to see a value lower that 0.2 (if there is one). Given that this "agreement" is used to support a statement on Lines 653 – 654, it's worth a second look.*

Thank you for this comment. We agree with the reviewer that the decay height seems to follow a continuous evolution with respect to $u_*$. We have fitted the decay height to a square root function

and to a logarithmic function (see figure below, not shown in the manuscript). A better fit was obtained with the latter. As expected from a logarithmic function, the rate of increase of the decay height with $u_*$ is higher at low friction velocities. Therefore, we believe we can still relate this result to the increase of the vertical ejection velocity as $u_*$ rises from 0.15 to 0.3 m s$^{-1}$. This is now acknowledged in the text (l.743-745). In our view, it is fair to state that the simulation results do not deviate considerably from the parameterization of Nishimura and Hunt (2000) at this low friction velocities. This statement was added to the text as a replacement to the original one (l.737-738). The conclusions were also slightly modified to better reflect the updated analysis of the results (l.951-953). The lowest value of $u_*$ considered in the simulations is equal to 0.15 m s$^{-1}$, which is slightly lower than the assumed fluid threshold friction velocity ($u_{*,ft}$ = 0.154 m s$^{-1}$). For $u_*$<0.15 m s$^{-1}$, the mass flux would be highly intermittent.

[Figure]

Figure VIII: Decay height as a function of the friction velocity. Fit to a square root function as well as a logarithmic function.

*7. Figure 9: Great figure!*

Thank you.

*8. Line 847: "particle mas flux" should be "particle mass flux"*

This was corrected in the text.

**Additional changes made in the manuscript**

1) Assumption of a characteristic path:

When deducing Eq. 1, it is assumed that the ensemble of particle trajectories can be represented by a characteristic path. This is now acknowledged in the text (l.207-209). This assumption allows us to compute Q, $\langle l_p \rangle$ and $\langle a_p \rangle$ as a function of the mean quantities only (l.332-333, l.368). Even though this methodology is explicitly or implicitly assumed in most models available in the literature, its limitations are rarely discussed. This is also acknowledged in the text (l.217-220).

**References not included in the revised manuscript:**

Bintanja, R. (2000), Snowdrift suspension and atmospheric turbulence. Part I: Theoretical background and model description, *Boundary-layer Meteorology*, 95, 343–368.

Dyunin, A. K., Anfilofiyev, B. A., Istrapilovich, M. G., Mamayeva, N. T. and Kvon, Y. D. (1977), Strong Snow-Storms, their Effect on Snow Cover and Snow Accumulation, *Journal of Glaciology*, 19(81), 441–449.

Gauer, P. (2001), Numerical modeling of blowing and drifting snow in Alpine terrain, *Journal of Glaciology*, 47 (156), 97-110.

Lee, L. W., and L. Wah (1975), *Sublimation of Snow In Turbulent Atmosphere*, PhD Dissertation, Department of Mechanical Engineering, University of Wyoming.

Lin, Y. L., Farley, R. D. and Orville, H. D. (1983), Bulk Parameterization of the Snow Field in a Cloud Model, *Journal of Applied Meteorology and Climatology*, 22(6), 1065-1092.

Marsh, C.B., Pomeroy, J.W., Spiteri, R.J. and Wheater, H.S. (2020), A finite volume blowing snow model for use with variable resolution meshes, *Water Resources Research*, *56*(2), e2019WR025307.

Maxey, M., and J. Riley (1983), Equations of motion for a small rigid sphere in a nonuniform flow. *The Physics of Fluids*, 26, 883–889.

Noël, B., van de Berg, W. J., van Wessem, J. M., van Meijgaard, E., van As, D., Lenaerts, J. T. M., Lhermitte, S., Kuipers Munneke, P., Smeets, C. J. P. P., van Ulft, L. H., van de Wal, R. S. W., and van den Broeke, M. R. (2018), Modelling the climate and surface mass balance of polar ice sheets using RACMO2 – Part 1: Greenland (1958–2016), *The Cryosphere*, 12(3), 811-831.

Talaei, A. and Garrett, T. J. (2020), On the Maxey-Riley equation of motion and its extension to high Reynolds numbers, 1–19 [online] Available from: http://arxiv.org/abs/2006.16577, 2020.

Schlichting, H., and Gersten, K. (2017), *Boundary-Layer Theory*, Springer (9th edition).

Schmidt, R.A. (1980), Threshold wind-speeds and elastic impact in snow transport. *Journal of Glaciology*, 26(94), 453-467.

Shiotani, M., and Arai, H. (1953), A short note on the snow storm, *Cong. Appl. Mech.*, Science Council of Japan.